MPP-2023-150
DESY-24-006

# The Geometry of Cosmological Correlators

Paolo Benincasa[†], Gabriele Dian[‡]

[†] Max-Planck-Institut für Physik, Werner-Heisenberg-Institut, D–80805 München, Germany
[‡] Deutsches Elektronen-Synchrotron DESY, Notkestr. 85, 22607 Hamburg, Germany

pablowellinhouse@anche.no, gabriele.dian@desy.de

## Abstract

We provide a first principle definition of cosmological correlation functions for a large class of scalar toy models in arbitrary FRW cosmologies, in terms of novel geometries we name *weighted cosmological polytopes*. Each of these geometries encodes a universal rational integrand associated to a given Feynman graph. In this picture, all the possible ways of organising, and computing, cosmological correlators correspond to triangulations and subdivisions of the geometry, containing the in-in representation, the one in terms of wavefunction coefficients and many others. We also provide two novel contour integral representations, one connecting higher and lower loop correlators and the other one expressing any of them in terms of a building block. We study the boundary structure of these geometries allowing us to prove factorisation properties and Steinmann-like relations when single and sequential discontinuities are approached. We also show that correlators must satisfy novel vanishing conditions. As the weighted cosmological polytopes can be obtained as an orientation-changing operation onto a certain subdivision of the cosmological polytopes encoding the wavefunction of the universe, this picture allows us to sharpen how the properties of cosmological correlators are inherited from the ones of the wavefunction. From a mathematical perspective, we also provide an in-depth characterisation of their adjoint surface.

January 2024

## 1 Introduction

All the structures we can observe in our universe are the result of an evolution from seeds which are assumed to be planted by quantum fluctuations during a period of phase of accelerated expansion, the inflationary phase, that happened in the early stage of the life of our universe and suddenly stopped. This implies that understanding (and computing) correlation functions at the end of inflation, on the one hand, provides us with insights into the patterns we can observe, for example, in the large-scale structures in our universe, while on the other hand, they constitute a window on the physics of inflation, which remains still a mystery. Said differently, a deep understanding of the mathematical structure of inflationary correlation functions can allow us to address both observational and purely theoretical questions.

In recent years, novel efforts have been put into unveiling how physics is encoded in such quantities and how to more effectively compute them. More precisely, most of the efforts focused on the so-called wavefunctional of the universe[1], whose squared modulus return the probability distribution through which a cosmological observable $\langle f \rangle$ is determined

$$\langle f[\Phi] \rangle = \mathcal{N} \int \mathcal{D}\Phi \, |\Psi[\Phi]|^2 f[\Phi], \qquad \mathcal{N}^{-1} := \int \mathcal{D}\Phi \, |\Psi[\Phi]|^2 \qquad (1.1)$$

---

[1]See [1, 2] and references therein.

where $\boldsymbol{\Phi}$ collectively identifies the states at the space-like boundary where inflation ends, $\boldsymbol{\Psi}[\boldsymbol{\Phi}]$ is the wavefunctional of the boundary state configuration $\boldsymbol{\Phi}$, $f[\boldsymbol{\Phi}]$ is the quantity which needs to be spatially averaged, and $\mathcal{N}$ is the normalisation. Despite in and of itself the wavefunctional is not an observable quantity, it enjoys several sufficiently physical properties, such as gauge invariance. Furthermore, its structure determines (at least part of the) structure of an actual observable via (1.1). Hence, a number of theoretical questions can be asked directly at the level of the wavefunctional, for example about the imprint of causality [3, 4] and unitarity [5–15].

Perturbatively, the wavefunctional for a large class of scalar toy models enjoys a first principle description in terms of *cosmological polytopes* [16, 17]: they are defined in projective space independently of the physics, and the relation of each one of them with the latter is established in a certain local patch of projective space – which provides a parametrisation of the kinematic space of a given process through the *energies* [2] involved – via a differential form uniquely associated to a given polytope. Such a differential form, named *canonical form* [18], encodes the contribution of a given process to the wavefunctional into its *canonical function, i.e.* the rational function obtained by stripping off the standard measure of projective space. In particular, the canonical form is characterised by having logarithmic singularities on, and only on, the boundaries of the associated cosmological polytope. Each boundary of a cosmological polytope is still a (lower-dimensional) polytope whose canonical form is given by the residue along the given boundary of the canonical form associated to the original cosmological polytope. Because of the relation between canonical form and wavefunction, the boundary structure of the cosmological polytopes encodes the residues of the latter: the organisation of the vertices in each facet encodes the factorisation properties of the wavefunction [16, 19], while the one for codimension-2 and higher faces encodes the compatibility among different channels, generalising the flat-space Steinmann-relations [20–28] to cosmology [3] and to multiple discontinuities in both flat and expanding universes [29] [3].

As the wavefunction of the universe is related to the probability distribution which serves to compute any correlations at future infinity, a general question is how much of the structure of these correlations determines and how. The combinatorial picture of the cosmological polytopes provides a framework to make concrete and address this question. The first principle, mathematical, definition of a cosmological polytope is in terms of the intersection among originally disconnected triangles in the midpoints of at most two of their sides [16]. Such a definition identifies a set of special points, *i.e.* the midpoints where the triangles can get intersected, as well as a set of special 2-planes, *i.e.* the ones containing the triangles. They provide a specific polytope subdivision of the cosmological polytope. As we will show, a change in the mutual orientation among the elements of this subdivision along the facets that are internal to the original cosmological polytope, generates a novel object which turns out to encode the cosmological correlators. Such a novel object, that we name *weighted cosmological polytope*, is no longer a polytope, and more generally a positive geometry, in the standard sense as now contains *internal boundaries* corresponding to the facets introduced by the subdivision, and such that

---

[2]The term *energy* is used here with little bit abuse of language as in expanding universes there is no of time-translation and identifies the modulus of a momentum.

[3]These results have been proven graph-by-graph and, hence, they hold for any theory involving states with flat-space counter-part as the singularity structure of the wavefunction for these scalar toy models includes the one of the wavefunction for such other states.

not all the maximal residues have the same normalisation (up to a sign). The weighted cosmological polytopes fall into a class of objects, the *weighted positive geometries*, firstly introduced in [30]. As in the case of polytopes, they are also characterised by a differential form, the *weighted canonical form*, and a weight function which encodes the difference in the relative normalisation of the maximal residues, or, that is the same, the orientation along the boundaries of the geometry.

In this paper, we provide a first principle definition of the cosmological correlators in terms of these weighted cosmological polytopes. The weighed cosmological polytopes can be defined either as an orientation-changing operation on the elements of a subdivision of a cosmological polytope, as we have mentioned earlier, or by characterising their boundary structure at all codimensions as well as how it reflects into their canonical form and its residues. Such definitions allow us to find novel representations of the cosmological correlators, which correspond to triangulations of the associated weighted cosmological polytope.

The knowledge of the boundary structure of the weighted cosmological polytopes translates into constraints on arbitrary codimension singularities, showing the existence of both factorisation theorems and Steinmann-like relations for the correlators. Interestingly, if on one side this information allows us to fix the canonical form associated to the weighted cosmological polytopes, for example via its triangulations, on the other side the compatibility conditions on its facets are not enough to fix its numerator, as it is customary in the usual cosmological polytope (and more generally in any polytope). The compatibility conditions determine which hyperplanes containing a facet of a geometry intersect each other outside of the geometry in a given codimension and, hence, the associated multiple residue of the canonical form is zero. Such intersections are subspaces which do not intersect the original geometry in its interior, and their collection identifies a hypersurface, named adjoint surface [31], whose associated polynomial corresponds to the numerator of the canonical form. In the case of the weighted cosmological polytopes, the compatibility conditions are not enough to uniquely fix the adjoint surface [4] and, hence, these constraints do not fix uniquely the numerator of the canonical form of the weighted cosmological polytopes. This implies that new conditions need to be imposed to have the adjoint surface fixed and, thus, the definition itself of the adjoint surface would need to be modified. Nevertheless, the canonical form is uniquely fixed by its residues: being a rational function, the knowledge of its singularities and of its behaviour as the singularities are approached completely determines it [5], which is precisely what the triangulation does. Then, why should we worry about the adjoint surface at all? From a physics perspective, it is revealing that the locus of the zeroes of the cosmological correlators is determined by conditions beyond the Steinmann-like relations found in [3, 29] and it is interesting to understand not only what they are but also what is their origin.

The paper is organised as follows. In Section 2, we review the path integral definition of the Bunch-Davies wavefunction and its perturbative realisation via Feynman graphs. We also discuss the relation between the wavefunction, the probability distribution and the cosmological correlators, deriving the perturbative rules to compute the latter in terms of wavefunction perturbative Feynman

---

[4]The same phenomenon was first observed in another generalisation of the notion of polypols, which is characterised by having non-linear hypersurfaces as boundaries [32].

[5]The numerator of a canonical form is a polynomial whose degree is determined by the requirement that the canonical form is projective invariant. It turns out to be such that the canonical form vanishes as the energies are taken to infinity and, consequently, it can be reconstructed by just the knowledge of the poles at a finite location.

graphs. We also derive a Feynman tree theorem and its generalisation for the cosmological correlators, which is a direct consequence of an analogous theorem holding for the wavefunction [19] [6]. Section 3 is devoted to the discussion of the salient and relevant features of the combinatorial description of the perturbative wavefunction in terms of the cosmological polytopes. In section 4 we introduce the weighted cosmological polytopes as the result of an orientation-changing operation on the cosmological polytopes. We provide a characterization of it by studying its boundary structure as well as via compatibility conditions among the facets. We relate its weighted canonical form to the cosmological correlators by identifying one of the signed triangulations of the former with the result of the Keyldish-Schwinger computation of the latter. We provide novel representations for the cosmological correlators which are realised in terms of triangulations of the weighted cosmological polytopes. Also, the analysis of its boundary structure allows us to formulate factorisation theorems for the correlators and to prove that they satisfy the very same Steinman-like relations as the wavefunction. We formulate novel conditions that determine the zeroes of the cosmological correlators via the characterisation of the adjoint surface of the weighted cosmological polytopes. Section 5 is devoted to our conclusion and outlook. Finally, Appendix A contains further mathematical properties of the adjoint surface for the weighted cosmological polytopes.

### Summary of results

As the paper presents a number of technical aspects and for the sake of clarity, we present here our main results organised conceptually.

**Properties of the correlators from the wavefunctional** – The wavefunctional of the universe encodes the probability distribution for the state configurations at the space-like infinity and hence any observable, defined as an average over such probability distribution, inherits several features from it. In perturbation theory, correlators can be expressed in terms of wavefunction graphs, providing an alternative way of computing them and analysing their structure. We sharpen further this relation showing that the correlators enjoy the very same Steinmann-like relations as the wavefunction as well as the existence of a contour integral representation relating higher loop correlators to lower loop ones which is a direct consequence of a similar representation for the wavefunction graphs.

**A novel first principle formulation for correlators** – Correlation functions, and more generally any observable, play a crucial role in unveiling fundamental physics not only because they can be directly related to experiments or observations, but also because they provide us with the theoretical framework necessary for a deep understanding of physical phenomena. Here we provide a novel first principle definition for the cosmological correlators in terms of weighted cosmological polytopes, of which we provide an intrinsic formulation. Such a formulation relies on mathematical principles which do not make any explicit reference to physics, and allow for an independent characterisation. We show novel ways of computing the correlators which descend directly from the geometrical properties of the weighted cosmological polytopes, and to characterise them.

---

[6]A different formulation of a Feynman tree theorem in cosmology for the wavefunction was proposed in [33].

**Factorisation theorems, Steinmann-like relations and novel zeroes** – Using the formulation of the cosmological correlators in terms of the weighted cosmological polytopes, we show that as the singularities they share with the wavefunction are approached, a correlator (at any loop order) factorises into a lower-point/lower-level flat-space amplitude and a linear combination of correlators. We also show that as the other of its singularities are approached, a correlator can either reduce to a lower-loop correlator or factorise into two lower-point correlators. This construction allows also to study their singularity structure at all co-dimensions, proving the validity of the same Steinmann-like relations as for the wavefunction, and novel vanishing conditions which are specific to the correlators.

**The mathematical side: weighted polytopes** – We have defined a novel class of objects which falls into the weighted positive geometries. However, its peculiar structure, and its $1-1$ correspondence with graphs, allow for a complete and general characterisation of the whole class. In particular, we can fully characterise the structure of the adjoint surface which is no longer fixed by the compatibility conditions among its facets only, as is the case for convex polytopes. For weighted polytopes, such conditions identify a multi-parameter family of adjoint surfaces, with these parameters directly related to the weight function. In the specific case of weighted cosmological polytopes, the additional constraints which allow to fix completely the adjoint surface, are given by the requirement that elements of a certain subdivision of each facet which is still a weighted polytope, have the same residue along a given special hyperplane. This requirement is a consequence of the first principle definition of the geometry and is reflected into the $1-1$ correspondence with the graphs. Some of these conditions can be phrased as vanishing conditions when a facet is covariantly restricted to other special hyperplanes.

## 2 From the wavefunction to cosmological correlators

In this section, we provide the basic definition and review the fundamental properties of the observables of interest, *i.e.* the Bunch-Davies wavefunction, the probability distribution and the late-time correlation function. We will focus on the relation among them and the associated diagrammatics.

### 2.1 The Bunch-Davies wavefunction

Let us begin with a system whose time evolution is described by the Hamiltonian density operator $\hat{H}(\eta)$. The wavefunctional of the universe is then defined as the transition amplitude from a vacuum state $|0\rangle$ in the infinite past $\eta \longrightarrow -\infty$, and a certain field configuration $\langle\Phi|$ at the space-time boundary $\eta = 0$:

$$\Psi[\Phi] := \langle\Phi|\widehat{\mathcal{T}}\left\{\exp\left[-i\int_{-\infty}^{0} d\eta\,\hat{H}(\eta)\right]\right\}|0\rangle = \mathcal{N}\int_{\phi(-\infty)=0}^{\phi(0)=\Phi} \mathcal{D}\phi\,e^{iS[\phi]} \tag{2.1}$$

where $\widehat{\mathcal{T}}$ is the time-ordered operator, while the second equality provides the path integral representation with the Bunch-Davies vacuum condition in the infinite past, which selects the positive *energy* modes only

$$\phi(\eta) \xrightarrow{\eta\longrightarrow-\infty} f(\eta)\,e^{iE\eta}, \qquad E := |\vec{p}| > 0 \tag{2.2}$$

$f(\eta)$ being determined by the cosmology. As the modes infinitely oscillate in the infinite past, the wavefunction needs to be regularised as $\eta \longrightarrow -\infty$. This can be done by giving a small negative imaginary part to the energies [14]. In perturbation theory, the wavefunction can be formally written as

$$\Psi[\Phi] \;=\; \Psi_{\text{free}}[\Phi] \times \left\{ 1 + \sum_{n\geq 2} \int \prod_{j=1}^{n} \left[ \frac{d^d p_j}{(2\pi)^d} \Phi[\vec{p}_j] \right] \sum_{L\geq 0} \psi_n^{(L)}(\vec{p}_1,\ldots,\vec{p}_n) \right\} \tag{2.3}$$

where $\Psi_{\text{free}}[\Phi]$ is the free wavefunction, which is given by a Gaussian, $\Phi(\vec{p}_j) := \langle \Phi | \vec{p}_j \rangle$, and $\psi_n^{(L)}(\vec{p}_1,\ldots,\vec{p}_n)$ are the wavefunction coefficients with $n$ external states and $L$ loops. In turn, the connected part of wavefunction coefficients can be represented as a sum of graphs whose vertices represent the interactions, the edges connecting them the propagation of internal states, while the lines stretching from the vertices to the boundary represent the external states. Let $\mathcal{G}_n^{(L)}$ be the set of graphs with $n$ external states and $L$ loops, then each wavefunction coefficient $\psi_n^{(L)}$ is expressed as sums of graphs in $\mathcal{G}_n^{(L)}$

$$\psi_n^{(L)}(\vec{p}_1,\ldots,\vec{p}_n)\big|_{\text{connected}} \;=\; \sum_{\mathcal{G} \subset \mathcal{G}_n^{(L)}} \widetilde{\psi}_{\mathcal{G}}(\vec{p}_1,\ldots,\vec{p}_n), \tag{2.4}$$

where the wavefunction contribution $\widetilde{\psi}_{\mathcal{G}}$ from the graph $\mathcal{G}$ is explicitly given by

$$\widetilde{\psi}_{\mathcal{G}}(\vec{p}_1,\ldots,\vec{p}_n) \;=\; \delta^{(d)}\left( \sum_{j=1}^{n} \vec{p}_j \right) \int_{-\infty}^{0} \prod_{s\in\mathcal{V}} [d\eta_s \, i\lambda(\eta_s)\phi_{\circ}^{(s)}(\eta_s)] \prod_{e\in\mathcal{E}} \widetilde{G}(y_e; \eta_{s_e}, \eta_{s'_e}), \tag{2.5}$$

with $\lambda(\eta_s)$ being the vertex operator [7] which encodes the information of the cosmology and the interactions at the site $s$, $\phi_{\circ}^{(s)}$ being the product of external states at the site $s$, and $\widetilde{G}(y_e; \eta_{s_e}, \eta_{s'_e})$ being the propagator for the internal state associated to the edge $e$.

## 2.2 The probability distribution

The importance of the wavefunction discussed above relies on the fact that it encodes the probability distribution $\mathfrak{P}[\Phi]$ for a certain field configuration $\Phi$ at the boundary $\eta = 0$, via its squared modulus

$$\mathfrak{P}[\Phi] \;:=\; \mathcal{N} \, |\Psi[\Phi]|^2, \qquad \mathcal{N}^{-1} \;:=\; \int \mathcal{D}\Phi \, |\Psi[\Phi]|^2. \tag{2.6}$$

It is possible to express $\mathfrak{P}[\Phi]$ perturbatively, via (2.3). In particular, it can also be organised in terms of the number of external boundary states [2]

$$\mathcal{N}\,\mathfrak{P}[\Phi] \;=\; e^{-2\mathrm{Im}\{S_2[\Phi]\}} \left\{ 1 + \int \left[ \prod_{j=1}^{2} \frac{d^d p_j}{(2\pi)^d} \Phi(\vec{p}_j) \right] \sum_{L=1}^{\infty} \left[ \psi_2^{(L)} + \psi_2^{\dagger(L)} \right] + \right.$$
$$\left. + \sum_{k\geq 3} \int \left[ \prod_{j=1}^{k} \frac{d^d p_j}{(2\pi)^d} \Phi[\vec{p}_j] \right] \sum_{L\geq 0} \left[ \psi_k^{(L)} + \psi_k^{\dagger(L)} + \sum_{M=0}^{L} \sum_{r=2}^{k-2} \left( \psi_r^{(M)} \psi_{k-r}^{\dagger(L-M)} + \psi_{k-r}^{(L-M)} \psi_r^{\dagger(M)} \right) \right] \right\} \tag{2.7}$$

---

[7]The vertex operator $\lambda(\eta_s)$ is defined as a derivative operator in $\eta_s$ acting on $\phi_{\circ}^{(s)}$ and $\check{G}$. When the number of time derivatives is zero, it acts multiplicatively as a function of $\eta$ and a polynomial of rotational invariant combination of the external momenta.

The knowledge of the probability distribution $\mathfrak{P}[\Phi]$ allows to compute the spatial average of any operator $\widehat{\mathcal{O}}[\Phi]$ build out of the boundary configurations $\Phi$:

$$\langle\widehat{\mathcal{O}}[\Phi]\rangle = \int \mathcal{D}\Phi\,\mathfrak{P}[\Phi]\,\widehat{\mathcal{O}}[\Phi] = \mathcal{N}\int \mathcal{D}\Phi\,|\Psi[\Phi]|^2\,\widehat{\mathcal{O}}[\Phi]. \tag{2.8}$$

In particular, its perturbative expression (2.7) allows to obtain Feynman rules for the correlator $\langle\widehat{\mathcal{O}}[\Phi]\rangle$ in terms of wavefunction coefficients via (2.8). In the next subsection, we will describe such rules when the operators under consideration are given by

$$\widehat{\mathcal{O}}[\Phi] = \prod_{j=1}^{n} \Phi(\vec{p}_j) \tag{2.9}$$

The correlation functions $\langle\Phi(\vec{p}_1)\ldots\Phi(\vec{p}_n)\rangle$ in a fixed and expanding background are referred to as *cosmological correlators*. The most common way of computing such correlation functions and via the Keyldish-Schwinger formalism – see, for example, [34]. However, in this paper we are taking the point of view of the wavefunction via (2.8). This formula relates the correlations $\langle\widehat{\mathcal{O}}[\Phi]\rangle$ to the probability distribution $\mathfrak{P}[\Phi]$ of a field configuration $\Phi$. This implies that a number of feature of $\langle\widehat{\mathcal{O}}[\Phi]\rangle$ are determined by $\mathfrak{P}[\Phi]$ and, in turn, by wavefunction. A general question that we would like to address is which features of a correlation function can be directly read off from the wavefunction.

### 2.3 Cosmological correlators

Let us consider the correlation function of $n$ fields $\{\Phi(\vec{p}_j),\, j = 1,\ldots,n\}$ defined via (2.8)

$$\langle\prod_{j=1}^{n}\Phi(\vec{p}_j)\rangle = \mathcal{N}\int\mathcal{D}\Phi\,|\Psi[\Phi]|^2\,\prod_{j=1}^{n}\Phi(\vec{p}_j), \tag{2.10}$$

as well as the expression written for the probability distribution in terms of the wavefunction coefficients (2.7). The path integral (2.10) can be straightforwardly performed. The non-vanishing contribution to the $n$-point correlator from the path integral (2.10) can be written in the following schematic form

$$\langle\prod_{j=1}^{n}\Phi(\vec{p}_j)\rangle = \prod_{j=1}^{n}\frac{1}{2\mathrm{Re}\{\psi_2(E_j)\}}\sum_{L\geq 0}\sum_{m\geq 0}\prod_{\substack{a,b=1\\a<b}}^{m}\left[\frac{1}{2\mathrm{Re}\{\psi_2(y_{ab})\}}\right]\times$$
$$\times\left\{\psi_{n+2m}^{(L-m)} + \psi_{n+2m}^{\dagger\,(L-m)}\sum_{r=2}^{n+2m}\sum_{M=0}^{L}\left[\psi_{r+m}^{(M)}\psi_{n+m-r}^{\dagger\,(L-M)} + \psi_{r+m}^{\dagger\,(M)}\psi_{n+m-r}^{(L-M)}\right]\right\} \tag{2.11}$$

where $E_j := |\vec{p}_j|$ is the energy of the external $j$-th states, while the $y_{ab}$'s are the internal energies – when performing the path integral (2.10), there are just two classes of contribution surviving which return the two terms in the curly brackets, which are respectively connected and disconnected wavefunction contributions, and the factors $(2\mathrm{Re}\{\psi_2(E)\})^{-1}$ come from the Gaussian integration. As the correlation functions have an overall factor given by the product of the inverse of the two-point wavefunction for each external state, it is convenient to strip them off

$$\langle\prod_{j=1}^{n}\Phi(\vec{p}_j)\rangle = \prod_{j=1}^{n}\frac{1}{2\mathrm{Re}\{\psi_2(E_j)\}}\langle\prod_{j=1}^{n}\Phi(\vec{p}_j)\rangle' \tag{2.12}$$

With a little abuse of language, we will refer to these stripped, primed, correlators simply as *cosmological correlators*. The formula (2.11) can be conveniently translated into operations on the graphs associated to the wavefunctions. Given a graph $\mathcal{G}$ and the related wavefunction coefficient $\psi_\mathcal{G}$, the associated correlator (times the product of the real part of the two-point correlation function for each external state) can be computed from the wavefunction coefficient $\psi_\mathcal{G}$ and all those contributions obtained by eliminating in turn an edge at the time of the graph replacing it with the inverse of the real part of the two-point wavefunction. This latter operation can be graphically represented by marking with a dash the relevant edges ⎯⎯┤⎯⎯ – see Section 2.5 for explicit examples. Therefore, a cosmological correlator with $n$ external states and at $L$-loops, can be written as a sum over all the wavefunction graphs topologies contributing at $n$-points and at $L$-loop, with each term obtainable from the corresponding wavefunction graph and all the possible ways of dashing its edges

$$\langle \prod_{j=1}^{n} \Phi(\vec{p}_j) \rangle' = \sum_{\mathcal{G} \subset \mathcal{G}_n^{(L)}} \mathcal{C}_\mathcal{G} \equiv \sum_{\mathcal{G} \subset \mathcal{G}_n^{(L)}} \langle \prod_{j=1}^{n} \Phi(\vec{p}_j) \rangle_\mathcal{G}, \tag{2.13}$$

where $\mathcal{G}_n^{(L)}$ is the set of graphs with $n$ external states at $L$-loop order in perturbation theory. Let $\mathcal{E}_j \subseteq \mathcal{E}$ be the set of $j \in [0, n_e]$ fixed edges and $\{\mathcal{E}_j\}$ the set of all inequivalent ways of choosing $\mathcal{E}_j$. Let also $\mathcal{G}_j$ be the graph obtained from $\mathcal{G}$ by erasing all the edges in $\mathcal{E}_j$ – it is a graph which can be either connected or disconnected with the same set of sites $\mathcal{V}$ as $\mathcal{G}$ and edges given by the elements of $\mathcal{E} \setminus \mathcal{E}_j$. Then, the correlator $\mathcal{C}_\mathcal{G}$ can be written as

$$\mathcal{C}_\mathcal{G} = \sum_{j=0}^{n_e} \sum_{\mathcal{G}_j \in \{\mathcal{G}_j\}} \psi_{\mathcal{G}_j} \tag{2.14}$$

where $\mathcal{E}_0 = \varnothing$ and $\mathcal{E}_{n_e} = \mathcal{E}$, and the erased edge is replaced by the inverse of the two-point wavefunction with the same energy as the erased edge. Note that the set $\{\mathcal{G}_j\}$ has $\binom{n_e}{j}$ elements, and for $j = n_e$ the only term corresponds to a product of contact graphs. Finally, it is straightforward to count the number of wavefunction terms contributing to a correlator $\mathcal{C}_\mathcal{G}$. Let $n_e$ be the number of edges of a given graph $\mathcal{G}$, then the total number $\mathcal{N}_{\mathcal{C}_\mathcal{G}}$ of terms in the representation (2.14) is given by

$$\mathcal{N}_{\mathcal{C}_\mathcal{G}} = \sum_{j=0}^{n_e} \binom{n_e}{j} = 2^{n_e}. \tag{2.15}$$

In the next subsection, we will spell out these diagrammatic rules for a large class of scalar toy models.

## 2.4 Observables and universal integrands

Let us consider now the following class of scalar toy models with time-dependent mass and polynomial interactions with time-depending couplings

$$S[\phi] = -\int_{-\infty}^{0} d\eta \int d^d x \left[ \frac{1}{2} (\partial \phi)^2 - \frac{1}{2} m^2(\eta) \phi^2 - \sum_{k \geq 3} \frac{\lambda_k(\eta)}{k!} \phi^k \right]. \tag{2.16}$$

This action describes also scalar states with arbitrary masses in FRW cosmologies, provided that the functions $m^2(\eta)$ and $\lambda_k(\eta)$ are related to the metric

$$ds^2 = a^2(\eta) \left[ -d\eta^2 + d\vec{x} \cdot d\vec{x} \right] \tag{2.17}$$

via [17]

$$m^2(\eta) \;=\; m^2 a^2(\eta) + 2d\left(\xi - \frac{d-1}{4d}\right)\left[\partial_\eta\left(\frac{\dot a}{a}\right) + \frac{d-1}{2}\left(\frac{\dot a}{a}\right)^2\right],$$

$$\lambda_k(\eta) \;=\; \lambda_k\,[a(\eta)]^{2-\frac{(d-1)(k-2)}{2}}$$

(2.18)

$m$ and $\lambda_k$ being the bare mass and $k$-point coupling respectively, $\xi$ is a parameter which can acquire the value 0 when the coupling is minimal and $(d-1)/4d$ when the states are conformally coupled, and " $\cdot$ " indicates the derivative with respect to the conformal time $\eta$.

We will restrict ourselves to the cases such that $m^2(\eta) = \mu^2/\eta^2$, corresponding to a scalar with generic mass in de Sitter space, *i.e.* for $a(\eta) = \ell/(-\eta)$, and massless scalars in FRW cosmologies of the type $a(\eta) = (\ell/(-\eta))^\gamma$. In all these cases, the solution of the equation of motion from (2.16) is given in terms of Hankel functions. The Bunch-Davies condition fixes the mode function $\phi_\circ^{(\nu)}$ to be

$$\phi_\circ^{(\nu)}(-E\eta) \;=\; \sqrt{-E\eta}\,H_\nu^{(2)}(-E\eta)$$

(2.19)

with the order parameter $\nu$ of the Hankel function being related to the mass of the state and the spatial dimensions: $\nu = \sqrt{1/4 - (m\ell)^2}$ for conformal couplings, $\nu = \sqrt{d^2/4 - (m\ell)^2}$ for minimal couplings in de Sitter, an $\nu = 1/2 + (d-1)\gamma/2$ for a massless state in FRW cosmologies. The contribution of a graph $\mathcal{G}$ to the wavefunction for this class of models can be written from (2.5) as

$$\widetilde\psi_{\mathcal{G}} \;=\; \int_{-\infty}^0 \left[\prod_{s\in\mathcal{V}} d\eta_s\, i\lambda_{k_s}(\eta_s)\phi_\circ^{(s,\,\nu)}(\eta_s)\right]\prod_{e\in\mathcal{E}}\widetilde G(y_e;\eta_{s_e},\eta_{s'_e})$$

(2.20)

where the vertex function is just the time-dependent coupling. For states identified by half-integer order parameter $\nu = l + 1/2$, $l \in \mathbb{Z}_+ \cup \{0\}$, the mode functions can be written in terms of a differential operator $\widehat{\mathcal{O}}_\nu(E)$ acting on the mode function for $\nu = 1/2$ which is a simple exponential [17]. Consequently, $\widetilde\psi_{\mathcal{G}}$ can be obtained by acting with such differential operators on the wavefunction computed with external conformally coupled scalars. Furthermore, this latter object can still be computed from the purely conformally coupled wavefunction coefficients via a recursion relation which involves also certain differential operators – for further details, see [17]. Finally, a further simplification is provided by using the following integral representation for the time-dependent couplings $\lambda_k(\eta)$

$$\lambda_k(\eta) \;=\; \int_{-\infty}^{+\infty} dz\, e^{iz\eta}\,\widetilde\lambda_k(z)$$

(2.21)

and the contribution $\widetilde\psi_{\mathcal{G}}$ can be obtained acting with an integro-differential operators on the *universal integrand* $\psi_{\mathcal{G}}(x,y)$ defined as [17]

$$\psi_{\mathcal{G}}(x_s,y_e) \;=\; \int_{-\infty}^0 \prod_{s\in\mathcal{V}}\left[d\eta_s\, i\, e^{ix_s\eta_s}\right]\prod_{e\in\mathcal{G}} G(y_e;\eta_{s_e},\eta_{s'_e})$$

(2.22)

where $x_s$ is the sum of the energies of the external states at the site $s$, $x$ and $y$ appearing on the left-hand-side being the sets $x := \{x_s,\, s \in \mathcal{V}\}$ and $y := \{y_e,\, e \in \mathcal{E}\}$ of all the $x_s$'s and all the $y_e$'s respectively, and $G(y_e;\eta_{s_e},\eta_{s'_e})$ is the internal propagator given by

$$G(y_e;\eta_{s_e},\eta_{s'_e}) \;=\; \frac{1}{2y_e}\left[e^{-iy_e(\eta_{s_e}-\eta_{s'_e})}\vartheta(\eta_{s_e}-\eta_{s'_e}) + e^{+iy_e(\eta_{s_e}-\eta_{s'_e})}\vartheta(\eta_{s'_e}-\eta_{s_e}) - e^{iy_e(\eta_{s_e}+\eta_{s'_e})}\right].$$

(2.23)

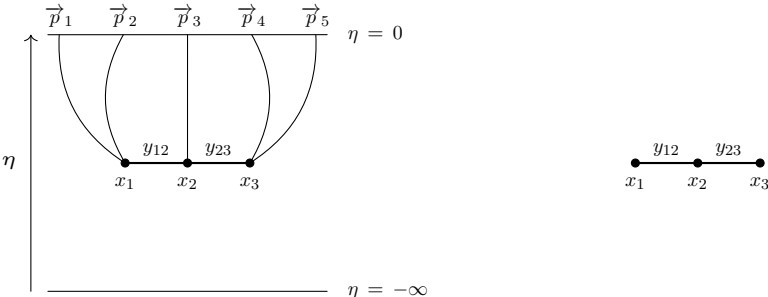

Figure 1: Reduced graphs. A Feynman graph (on the left) can be represented as a reduced graph (on the right) by suppressing the lines associated to the external states as well as the line representing the late-time boundary. It is a weighted graph whose site weights are the sums of the energies of the external states on the given site, while the edge weights are the energies of the internal states.

A distinctive feature of the internal propagator (2.23) for the wavefunction, is its three-term structure, with the first two constituting the time-ordered propagation and last one being required by the condition that the fluctuations vanish at the boundary.

The integrand $\psi_{\mathcal{G}}(x, y)$ in (2.22) is common to all cosmologies, which are instead specified by the function $\tilde{\lambda}(z)$ that acts as the integration measure in the space of external energies:

$$\widetilde{\psi}_{\mathcal{G}}(X, y) \; = \; \prod_{s \in \mathcal{V}} \left[ \int_{-\infty}^{+\infty} dx_s \, \tilde{\lambda}(x_s - X_s) \right] \int_{\Gamma} \prod_{e \in \mathcal{E}^{(L)}} [dy_e \, y_e] \, \mu\,(y_e) \, \psi_{\mathcal{G}}(x, y) \tag{2.24}$$

where the weights $\{y_e, \, e \in \mathcal{E}^{(L)}\}$ associated to the set $\mathcal{E}^{(L)} \subseteq \mathcal{E}$ of edges in the graph loops, parametrise the loop momentum space and $\mu(y_e)$ is its measure so that the integral of these edge-weights is just the loop integration, while the integrals over $\{x_s, \, s \in \mathcal{V}\}$ with measure $\{\lambda(x_s - X_s), \, s \in \mathcal{V}\}$ maps the integrand, whose kinematic space is parameterised by $\{x_s, \, s \in \mathcal{V}\}$ and $\{y_e, \, e \in \mathcal{E} \setminus \mathcal{E}^{(L)}\}$, into a function – the actual cosmological wavefunction – that is defined in the kinematic space parameterised by $X := \{X_s, \, s \in \mathcal{V}\}$ and $y := \{y_e, \, e \in \mathcal{E} \setminus \mathcal{E}^{(L)}\}$.

Going back to the wavefunction universal integrand $\psi_{\mathcal{G}}(x_s, y_e)$, note that it depends on the external energies in terms of sums $\{x_s, \, s \in \mathcal{V}\}$ of the energies of the states incident at each site $s$. A graph $\mathcal{G}$ can then be represented as a reduced weighted graph, suppressing the legs associated to the external states, and attaching the weights $x_s$ and $y_e$ to the site $s$ and the edge $e$ respectively. The structure of a given reduced graph $\mathcal{G}$ allows to read off the codimension-1 singularities of the associated universal integrand $\psi_{\mathcal{G}}(x_s, y_e)$. They are given by the total energy of each subprocess, which is identified by a subgraph $\mathfrak{g} \subseteq \mathcal{G}$ [16]:

$$E_{\mathfrak{g}} \; = \; \sum_{s \in \mathcal{V}_{\mathfrak{g}}} x_s + \sum_{e \in \mathcal{E}_{\mathfrak{g}}^{\text{ext}}} y_e \tag{2.25}$$

$\mathcal{V}_{\mathfrak{g}}$ and $\mathcal{E}_{\mathfrak{g}}^{\text{ext}}$ being respectively the set of vertices in $\mathfrak{g} \subseteq \mathcal{G}$ and the set of edges departing from $\mathfrak{g}$.

## 2.5 Universal integrands for cosmological correlators

For the class of models specified by (2.16), also the correlators enjoy an integral representation similar to the one for the wavefunction given in (2.24). Schematically, it can be written as

$$\widetilde{\mathcal{C}}_{\mathcal{G}} = \prod_{s \in \mathcal{V}} \left[ \int_{-\infty}^{+\infty} dx_s \, \tilde{\lambda}(x_s - X_s) \right] \int_{\Gamma} \prod_{e \in \mathcal{E}^{(L)}} [dy_e \, y_e] \, \mu(y_e) \, \mathcal{C}_{\mathcal{G}}(x, y) \tag{2.26}$$

where again the cosmology is encoded by the integration measure in the space of the site weights of $\mathcal{G}$

The general rules for computing the correlators from the knowledge of the wavefunction discussed in Section 2.3, directly translate to the integrand $\mathcal{C}_{\mathcal{G}}$, via the very same expression (2.14), where now all the terms ought to be interpreted as integrands.

Diagrammatically, one can compute all the terms contributing to the universal correlator integrands by starting with the associated wavefunction graph, and summing over all the possible ways of marking with a dash ——|—— with the dash operation representing the deletion of the edge and its replacement with the inverse of the two-point wavefunction, *i.e.* $1/y_e$ and shifting the weight of the sites at the endpoints of the dashed edge by $y_e$. It is instructive to work out explicitly some examples.

**Two-site line graph** – Let us begin with the simplest non-trivial example which is constituted by the process associated to a two-site line graph. Then, the universal correlator integrand – which, from now on, we will refer to simply as *correlator* – can be expressed as a sum of the connected wavefunction two-site line graph and the associated disconnected graph

$$\mathcal{C}_{\mathcal{G}} = \underset{x_1 \quad\quad x_2}{\bullet\!-\!\!\!-\!\!\!\overset{y}{-}\!\!\!-\!\!\!-\!\bullet} \quad + \quad \underset{x_1 + y \quad\quad y + x_2}{\bullet\!-\!\!\!-\!|\!-\!\!\!-\!\bullet} \tag{2.27}$$

which explicitly can be written as

$$\mathcal{C}_{\mathcal{G}} = \frac{2}{(x_1 + x_2)(x_2 + y)(y + x_2)} + \frac{2}{y(x_1 + y)(y + x_2)} \tag{2.28}$$

Note that the result (2.28) matches with the in-in computation in [35].

**Two-site bubble graph** – Let us move on to the simplest one-loop case. The dashed operation returns a representation of the two-site one-loop graph as a sum of $2^{n_e=2} = 4$ terms

$$\tag{2.29}$$

The diagrams (2.29) yield the functional expression

$$\mathcal{C}_{\mathcal{G}} = \frac{2(x_1 + y_a + y_b + x_2)}{(x_1 + x_2)(x_1 + x_2 + 2y_a)(x_1 + x_2 + 2y_b)(x_1 + y_a + y_b)(x_2 + y_a + y_b)}$$

$$+ \frac{1}{y_a(x_1 + x_2 + 2y_a)(x_1 + y_a + y_b)(x_2 + y_a + y_b)} + \frac{1}{y_b(x_1 + x_2 + 2y_b)(x_1 + y_a + y_b)(x_2 + y_a + y_b)}$$

$$+ \frac{1}{y_a y_b(x_1 + y_a + y_b)(x_2 + y_a + y_b)}. \tag{2.30}$$

## 2.6 From trees to loops: a tree theorem and its generalisation

The path integral definition of the cosmological correlator relates a single perturbative contribution to a sum over wavefunction graphs. However, a natural question is whether it is possible to express correlators at a certain order in perturbation theory in terms of lower-order ones. In this section, we show that the answer to this question is affirmative and that it is a direct consequence of the same type of relation existing for the wavefunction coefficients [19].

Let us consider the contribution to a cosmological correlator $\mathcal{C}_{\mathcal{G}}(x, y)$ associated to a graph $\mathcal{G}$. The graph $\mathcal{G}$ can be mapped into a graph with the same number of edges and one more loop by merging two of its sites:

This graphical operation can be analytically implemented on $\mathcal{C}_{\mathcal{G}}(x_s, y_e)$ by introducing a one-parameter deformation in the space of site-weights

$$x_i(z) := x_i + z, \quad x_j(z) := x_j - z, \quad x_k(z) := x_k, \ \forall\, k \neq i, j \tag{2.31}$$

in such a way that the poles of $\mathcal{C}_{\mathcal{G}}(x, y)$ which depend on both the shifted site weights do not depend on the deformation parameter $z$. Such a deformation maps the original function into a 1-parameter family of functions $\mathcal{C}_{\mathcal{G}}(x, y; z)$, which can be analysed as function of the parameter $z$.

Let us consider the function $\mathcal{C}_{\mathcal{G}}^{(i, j)}$ obtained by integrating $\mathcal{C}_{\mathcal{G}}(x, y; z)$ along the imaginary axis:

$$\mathcal{C}_{\mathcal{G}}^{(i, j)} := \frac{1}{2\pi i} \int_{-i\infty}^{+i\infty} dz\, \mathcal{C}_{\mathcal{G}}(x, y; z). \tag{2.32}$$

In the $z$ plane, the poles lie either on the positive or negative real axis – see Figure 2

$$
\begin{aligned}
\mathcal{P}_- &:= \left\{ z_{\mathfrak{g}} = -\left( \sum_{s \in \mathcal{V}_{\mathfrak{g}}} x_s + \sum_{e \in \mathcal{E}_{\mathfrak{g}}^{\text{ext}}} y_e \right), \ \forall\, \mathfrak{g} \subset \mathcal{G} \,|\, s_i \in \mathcal{V}_{\mathfrak{g}},\, s_j \notin \mathcal{V}_{\mathfrak{g}} \right\} \\
\mathcal{P}_+ &:= \left\{ z_{\mathfrak{g}} = +\left( \sum_{s \in \mathcal{V}_{\mathfrak{g}}} x_s + \sum_{e \in \mathcal{E}_{\mathfrak{g}}^{\text{ext}}} y_e \right), \ \forall\, \mathfrak{g} \subset \mathcal{G} \,|\, s_i \notin \mathcal{V}_{\mathfrak{g}},\, s_j \in \mathcal{V}_{\mathfrak{g}} \right\}
\end{aligned} \tag{2.33}
$$

and the contour of integration can be equivalently closed in the positive or negative half-plane:

$$
\begin{aligned}
\mathcal{C}_{\mathcal{G}}^{(i, j)} :=\ & \pm \frac{1}{2\pi i} \oint_{\gamma_{\mp}} dz\, \mathcal{C}_{\mathcal{G}}(x, y; z) = \\
=\ & \pm \sum_{z_{\mathfrak{g}} \in \mathcal{P}_{\mp}} \mathrm{Res}_{z=z_{\mathfrak{g}}} \left\{ \mathcal{C}_{\mathcal{G}}(x_s, y_e; z) | \right\}
\end{aligned} \tag{2.34}
$$

where $\gamma_{\mp}$ indicates the contour closes in the negative half-plane ($\gamma_-$) and positive half-plane, ($\gamma_+$), with this choice carrying an overall sign ($+$ for $\gamma_-$ and $-$ for $\gamma_+$). The last equality follows from the Cauchy theorem, with the sum over the residues running just on the residues at finite location as the

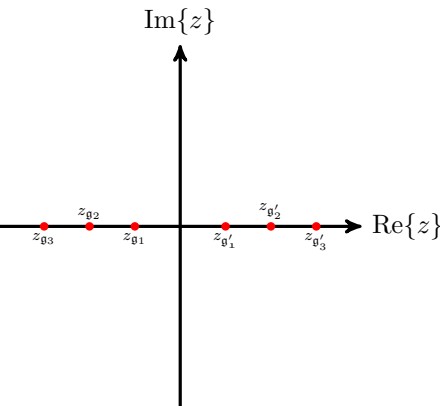

Figure 2: Poles location in the $z$-plane for a generic one-parameter family of correlators $\mathcal{C}_{\mathcal{G}}(x,y;z)$. The poles $z_{\mathfrak{g}_j}$ belong to the set $\mathcal{P}_-$, while $z_{\mathfrak{g}'_j}$ belong to $\mathcal{P}_+$. The contour can be closed either in the negative half-plane, enclosing with counter-clockwise orientation the poles which are function of $x_i$; or in the negative half-plane, enclosing the poles which are function of $x_j$ with clockwise orientation.

class of functions $\mathcal{C}_{\mathcal{G}}(x,y;z)$ vanishes as $z \longrightarrow \infty$. The two different contours provide two different representations for the function $\mathcal{C}_{\mathcal{G}}^{(i,j)}$: one in terms of the residues of the poles along the negative real axis and the other one in terms of the residues of the poles along the positive real axis.

Note that the poles of $\mathcal{C}_{\mathcal{G}}(x,y)$ which receive a $z$-dependence, and hence contribute in the sum (2.34), are all poles which are in common with the wavefunction. Furthermore, any of the poles in $\mathcal{P}_-$ corresponds to the total energy associated to a graph which can be either disconnected or partially overlapping with the graphs whose total energy is related to the poles in $\mathcal{P}_+$ (and vice versa). This implies that when the poles in $\mathcal{P}_+$ are computed at the location of the poles in $\mathcal{P}_-$ (and vice versa), they return polynomials which are positive sums only, they depend on $x_i$ and $x_j$ just through the combination $x_i + x_j$, and, were the energies associated to the two poles in $\mathcal{P}_-$ and $\mathcal{P}_+$ related to disconnected subgraphs, they correspond to the energies associated at the connected subgraphs of the graph obtained from $\mathcal{G}$ by merging the sites $s_i$ and $s_j$ into a single one with weight $x_i + x_j$. This last statement is straightforward to see as follows. Without loss of generality, let us consider a pole $z_{\mathfrak{g}} \in \mathcal{P}_-$. Then, any linear polynomial which gives rise to a pole in the $z$-plane in $\mathcal{P}_+$ acquires the form:

$$
\begin{aligned}
E_{\mathfrak{g}'}(z_{\mathfrak{g}}) \;=\; & x_j + \sum_{s' \in \mathcal{V}_{\mathfrak{g}'} \backslash \{s_j\}} x_{s'} + \sum_{e' \in \mathcal{E}_{\mathfrak{g}'}^{\text{ext}}} y_{e'} - z_{\mathfrak{g}} \;= \\
=\; & x_i + x_j + \sum_{s \in \mathcal{V}_{\mathfrak{g}} \backslash \{s_i\}} x_s + \sum_{e \in \mathcal{E}_{\mathfrak{g}}^{\text{ext}}} y_e + \sum_{s' \in \mathcal{V}_{\mathfrak{g}'} \backslash \{s_j\}} x_{s'} + \sum_{e' \in \mathcal{E}_{\mathfrak{g}'}^{\text{ext}}} y_{e'} \;= \\
=\; & \sum_{s \in \mathcal{V}_{\mathfrak{g}_{ij}}} x_s + \sum_{e \in \mathcal{E}_{\mathfrak{g}_{ij}}^{\text{ext}}} y_e
\end{aligned}
\tag{2.35}
$$

where $s_{ij}$ labels the site with weight $x_i + x_j$ obtained by merging the two sites $s_i$ and $s_j$, and $\mathfrak{g}_{ij}$ is a subgraph identified by the set of sites $\mathcal{V}_{\mathfrak{g}_{ij}} := \{\mathcal{V}_{\mathfrak{g}} \backslash \{s_i\}\} \cup \{\mathcal{V}_{\mathfrak{g}'} \backslash \{s_j\}\} \cup \{s_{ij}\}$ and edges $\mathcal{E}_{\mathfrak{g}_{ij}} := \mathcal{E}_{\mathfrak{g}} \cup \mathcal{E}_{\mathfrak{g}'}$.

If instead $\mathfrak{g}$ and $\mathfrak{g}'$ are partially overlapping, $E_{\mathfrak{g}'}(z_{\mathfrak{g}})$ turns out to be a spurious pole. Recall that that all the $z$-dependent denominators are all also wavefunction denominators. Recall also $\mathcal{C}_{\mathcal{G}}$ can be written in terms of wavefunction graphs, and hence $\mathcal{C}_{\mathcal{G}}(z)$ can be also expressed in terms of the deformed wavefunction coefficients associated to the same graphs. Individually, each term of such a sum satisfies the Steinmann-like relations [3]: they simply state that the double residues along partially overlapping channels vanish. Hence, if $\mathfrak{g}$ and $\mathfrak{g}'$ are partially overlaping graph, $E_{\mathfrak{g}'}(z_{\mathfrak{g}}) = 0$ is not a pole of $\mathrm{Res}_{z=z_{\mathfrak{g}}} \mathcal{C}_{\mathcal{G}}(z)$.

Finally, we are left to examine the singularities when either $\mathfrak{g}' \subset \mathfrak{g}$ or $\mathfrak{g} \subset \mathfrak{g}'$ and, thus, the associated poles both belong to the same set $\mathcal{P}_{\mp}$. Importantly, $E_{\mathfrak{g}'}(z_{\mathfrak{g}})$ and $E_{\mathfrak{g}}(z_{\mathfrak{g}'})$ would correspond to folded singularities. However, it is straightforward to show that $E_{\mathfrak{g}}(z_{\mathfrak{g}'}) = -E_{\mathfrak{g}'}(z_{\mathfrak{g}})$, and that the residues with respect to $E_{\mathfrak{g}'}(z_{\mathfrak{g}})$ and $E_{\mathfrak{g}}(z_{\mathfrak{g}'})$ are the same up to a sign: when summing over all terms in (2.34), the poles $E_{\mathfrak{g}}(z_{\mathfrak{g}'}) = -E_{\mathfrak{g}'}(z_{\mathfrak{g}})$ cancel.

Thus, $\mathcal{C}_{\mathcal{G}}^{(i,j)}$ has the very same singularities associated to a graph $\mathcal{G}_{ij}$ obtained from $\mathcal{G}$ by merging its sites $s_i$ and $s_j$ into a single site $s_{ij}$ with weight $x_i + x_j$.

Lastly, consider the representation of $\mathcal{C}_{\mathcal{G}}$ in terms of wavefunction graphs. Then $\mathcal{C}_{\mathcal{G}}^{(i,j)}$ can be seen as a sum of contour integrals around $\gamma_{\mp}$ of the $z$-deformed wavefunctions associated to such graphs. It was shown in [19] that precisely such a contour integration maps a wavefunction coefficient associated to a graph $\mathcal{G}$ into the wavefunction coefficient associated to the graph $\mathcal{G}_{ij}$ obtained by merging the sites $s_i$ and $s_j$, which has one site less but the same number of edges. Furthermore, given a graph with $n_e$ edges, the associated correlator can be written in terms of $2^{n_e}$ wavefunction graphs obtained from all the inequivalent ways of erasing an edge (from 0 to $n_e$). Since at the level of the wavefunction graphs, the contour integral maps the associated wavefunction into a wavefunction associated to $\mathcal{G}_{ij}$, all the terms appearing on the right-hand-side of $\mathcal{C}_{\mathcal{G}}^{(i,j)}$ when $\mathcal{C}_{\mathcal{G}}(z)$ is expressed in terms of wavefunction graphs, are precisely all the graphs that can be obtained from all the inequivalent ways of erasing an edge in $\mathcal{G}_{ij}$. Hence $\mathcal{C}_{\mathcal{G}}^{(i,j)} = \mathcal{C}_{\mathcal{G}_{ij}}$. Explicitly:

$$\mathcal{C}_{\mathcal{G}_{ij}}\left(x_i + x_j, \{x_k\}, \{y_e\}\right) \;=\; \frac{1}{2\pi i} \int\limits_{-i\infty}^{+i\infty} dz\, \mathcal{C}_{\mathcal{G}}\left(x_i(z), x_j(z), \{x_k\}, \{y_e\}\right), \qquad \left\{ \begin{array}{l} x_i(z) := x_i + z, \\ x_j(z) := x_j - z, \\ x_k(z) := x_k,\ \forall\, k \neq i,j. \end{array} \right.$$
$$(2.36)$$

The formula (2.36) establishes a relation between correlation functions at higher perturbative order with correlation functions at lower.

## 3 Cosmological polytopes and the wavefunction of the universe

Let us now consider a reduced graph $\mathcal{G}$ with $n_s$ sites and $n_e$ edges. Irrespectively of the topology of $\mathcal{G}$, it can be thought of as obtained from a collection of $n_e$ 2-site line graphs whose sites have been suitably identified: if the graph $\mathcal{G}$ has $n_s$ sites, then $r = 2n_e - n_s$ identifications need to be imposed – see Figure 3. The identification of the sites implies the identification of the corresponding weights.

Let us now consider the collection of $n_e$ 2-site line graphs from which $\mathcal{G}$ can be obtained. Each of them is endowed with a triples of weights $\{x_{s_e}, y_e, x_{s'_e}\}$. The set of these $n_e$ triples can be taken as local coordinates of a projective space $\mathbb{P}^{3n_e - 1}$. Then, each 2-site line graph can be associated to

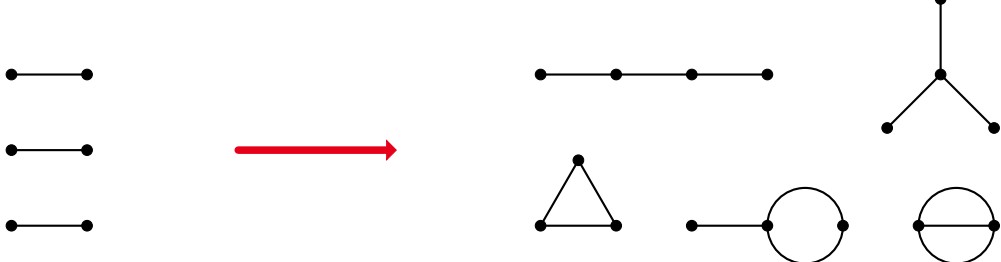

Figure 3: From a collection of 2-site line graphs to connected graphs. Given a collection of 2-site line graphs, it is possible to generate different topologies of connected graphs by merging them in their sites. Here we depict a collection of 3 2-site line graphs (on the left) and the possible connected topologies that can be generated from them (on the right).

a triangle identified via its midpoints given by the triple $\{\mathbf{x}_{s_e}, \mathbf{y}_e, \mathbf{x}_{s'_e}\}$, with the collection of triples $\{\mathbf{x}_{s_e}, \mathbf{y}_e, \mathbf{x}_{s'_e}\}_{e \in \mathcal{E}}$, associated to all triangles, forming a canonical basis for $\mathbb{P}^{3n_e-1}$. Each of these triangles have vertices $\{\mathbf{x}_{s_e} - \mathbf{y}_e + \mathbf{x}_{s'_e}, \mathbf{x}_{s_e} + \mathbf{y}_e - \mathbf{x}_{s'_e}, -\mathbf{x}_{s_e} + \mathbf{y}_e + \mathbf{x}_{s'_e}\}$ which can be equivalently used to define them. The identification which maps the collection of $n_e$ 2-site line graphs into the graph $\mathcal{G}$ with $n_s$ sites and $n_e$ edges, translates into the intersection of the associated triangles into their midpoints $\mathbf{x}_{s_e}, \mathbf{x}_{s_{e'}}, \ldots$, projecting it down to $\mathbb{P}^{3n_e-r-1} \equiv \mathbb{P}^{n_s+n_e-1}$. The convex hull of the vertices $\{\mathbf{x}_{s_e} - \mathbf{y}_e + \mathbf{x}_{s'_e}, \mathbf{x}_{s_e} + \mathbf{y}_e - \mathbf{x}_{s'_e}, -\mathbf{x}_{s_e} + \mathbf{y}_e + \mathbf{x}_{s'_e}\}_{e \in \mathcal{E}}$ after such an intersection defines the cosmological polytope $\mathcal{P}_{\mathcal{G}} \subset \mathbb{P}^{n_s+n_e-1}$. This definition establishes a $1-1$ correspondence between a graph $\mathcal{G}$ and a cosmological polytope $\mathcal{P}_{\mathcal{G}}$.

Given a cosmological polytope $\mathcal{P}_{\mathcal{G}} \subset \mathbb{P}^{n_s+n_e-1}$, there is a unique, up to an overall normalisation, differential form associated to it, named *canonical form*

$$\omega(\mathcal{Y}, \mathcal{P}_{\mathcal{G}}) = \Omega(\mathcal{Y}, \mathcal{P}_{\mathcal{G}})\langle \mathcal{Y}d^{n_s+n_e-1}\mathcal{Y}\rangle \tag{3.1}$$

where $\mathcal{Y}$ is a generic point in $\mathbb{P}^{n_s+n_e-1}$, $\Omega(\mathcal{Y}, \mathcal{P}_{\mathcal{G}})$ is a rational function named *canonical function*; and $\langle \mathcal{Y}d^{n_s+n_e-1}\mathcal{Y}\rangle$ is the canonical measure in $\mathbb{P}^{n_s+n_e-1}$.

The canonical form (3.1) has the properties of: *i*) having only logarithmic singularities along the boundaries of $\mathcal{P}_{\mathcal{G}}$; *ii*) having its residue along any of the boundaries being the canonical form describing a codimension-1 polytope; *iii*) having all its highest codimension singularities with the same normalisation, up to a sign.

It turns out that the canonical function $\Omega(\mathcal{Y}, \mathcal{P}_{\mathcal{G}})$ is the wavefunction coefficient associated to the graph $\mathcal{G}$

$$\Omega(\mathcal{Y}, \mathcal{P}_{\mathcal{G}}) = \psi_{\mathcal{G}}(x_s, y_e) \tag{3.2}$$

where the graph weights $\{x_s, s \in \mathcal{V}\}$ and $\{y_e, e \in \mathcal{E}\}$ are a local coordinate system for $\mathbb{P}^{n_s+n_e-1}$ [16]. The singularities of $\Omega(\mathcal{Y}, \mathcal{P}_{\mathcal{G}})$ along the boundaries of $\mathcal{P}_{\mathcal{G}}$ correspond to the singularities of $\psi_{\mathcal{G}}$, and the codimension-1 polytopes on them encode the residue of $\psi_{\mathcal{G}}$ with respect to the associated singularity. As the singularities of $\psi_{\mathcal{G}}$ are associated to subgraphs $\mathfrak{g} \subseteq \mathcal{G}$, there is a $1-1$ correspondence between subgraphs of $\mathcal{G}$ and codimension-1 boundaries of $\mathcal{P}_{\mathcal{G}}$, which are called *facets*. The latter is identified by the intersection $\mathcal{P}_{\mathcal{G}} \cap \mathcal{W}^{(\mathfrak{g})}$ of the cosmological polytope $\mathcal{P}_{\mathcal{G}}$ with the hyperplanes $\{\mathcal{W}^{(\mathfrak{g})}, \forall \mathfrak{g} \subseteq \mathcal{G}\}$

such that a number of vertices $\{\mathcal{Z}_e^{(j)}, \forall\, e \in \mathcal{E},\, j = 1, 2, 3\}$ of $\mathcal{P}_\mathcal{G}$ greater or equal to the dimension of the associated affine space satisfies the condition $\mathcal{Z}_e^{(j)} \cdot \mathcal{W}^{(\mathfrak{g})} = 0$ (the vertex $Z_e^{(j)}$ is on the hyperplane $\mathcal{W}^{(\mathfrak{g})}$) while the other vertices satisfy $\mathcal{Z}_e^{(j)} \cdot \mathcal{W}^{(\mathfrak{g})} > 0$ (the vertex is on the positive half-space identified by the hyperplane $\mathcal{W}^{(\mathfrak{g})}$). The hyperplane $\mathcal{W}^{(\mathfrak{g})}$ can be represented via the dual vector

$$\mathcal{W}^{(\mathfrak{g})} = \sum_{s \in \mathcal{V}_\mathfrak{g}} \tilde{\mathbf{x}}_s + \sum_{e \in \mathcal{E}_\mathfrak{g}^{\mathrm{ext}}} \tilde{\mathbf{y}}_e \tag{3.3}$$

where $\{\tilde{\mathbf{x}}_s,\, s \in \mathcal{V}\}$ and $\{\tilde{\mathbf{y}}_e,\, e \in \mathcal{E}\}$ constitutes a basis of the dual space of $\mathbb{P}^{n_s + n_e - 1}$ and satisfy

$$\mathbf{x}_s \cdot \tilde{\mathbf{x}}_{s'} = \delta_{ss'}, \qquad \mathbf{y}_e \cdot \tilde{\mathbf{y}}_{e'} = \delta_{ee'}, \qquad \mathbf{x}_s \cdot \tilde{\mathbf{y}}_e = 0 = \mathbf{y}_e \cdot \tilde{\mathbf{x}}_s \tag{3.4}$$

Consequently, given a generic point $\mathcal{Y} \in \mathbb{P}^{n_s + n_e - 1}$, the quantity $\mathcal{Y} \cdot \mathcal{W}^{(\mathfrak{g})}$ is just the total energy $E_\mathfrak{g}$ of the subprocess associated to the subgraph $\mathfrak{g} \subseteq \mathcal{G}$

$$\mathcal{Y} \cdot \mathcal{W}^{(\mathfrak{g})} = \sum_{s \in \mathcal{V}_\mathfrak{g}} x_s + \sum_{e \in \mathcal{E}_\mathfrak{g}^{\mathrm{ext}}} y_e \equiv E_\mathfrak{g} \tag{3.5}$$

and $\mathcal{Y}$ is a point of $\mathcal{P}_\mathcal{G}$ if $\mathcal{Y} \cdot \mathcal{W}^{(\mathfrak{g})} \geq 0\, \forall\, \mathfrak{g} \subseteq \mathcal{G}$. The relation (3.5) makes manifest that the boundary $\mathcal{P}_\mathcal{G} \cap \mathcal{W}^{(\mathfrak{g})}$ is approached as $E_\mathfrak{g} \longrightarrow 0$. For $\mathfrak{g} = \mathcal{G}$, $\mathcal{Y} \cdot \mathcal{W}^{(\mathcal{G})} = \sum_{s \in \mathcal{V}} x_s \equiv E_{\mathrm{TOT}}$ is the total energy of the process associated to $\mathcal{G}$, and the boundary $\mathcal{S}_\mathcal{G} := \mathcal{P}_\mathcal{G} \cap \mathcal{W}^{(\mathcal{G})}$ encodes the residue of the wavefunction as its total energy vanishes, which corresponds to (the high energy limit of) a flat-space scattering amplitude [35–37]. The facet $\mathcal{S}_\mathcal{G}$ of $\mathcal{P}_\mathcal{G}$ is named *scattering facet* [16].

As just mentioned, the facets $\{\mathcal{P}_\mathcal{G} \cap \mathcal{W}^{(\mathfrak{g})},\, \forall\, \mathfrak{g} \subseteq \mathcal{G}\}$ encode the residues of $\psi_\mathcal{G}$. They can be characterised by determining which vertices of $\mathcal{P}_\mathcal{G}$ are on each given facet $\mathcal{P}_\mathcal{G} \cap \mathcal{W}^{(\mathfrak{g})}$, *i.e.* which subset of them satisfies the condition $\mathcal{Z}_e^{(j)} \cdot \mathcal{W}^{(\mathfrak{g})} = 0$. This question can be answered straightforwardly via the $1-1$ correspondence between subgraphs and facets, and via the introduction of a marking identifying the vertices *which are not* on the facet of interest

Given a graph $\mathfrak{g} \subseteq \mathcal{G}$, the corresponding facet $\mathcal{P}_\mathcal{G} \cap \mathcal{W}^{(\mathfrak{g})}$ is obtaining by marking all the edges of $\mathfrak{g}$ midway as well as the edges departing from $\mathfrak{g}$ close to theirs sites in $\mathfrak{g}$ [16].

The marking introduced above allows to characterise the full face structure of $\mathcal{P}_\mathcal{G}$. A codimension-$k$ face is given by the intersection $\mathcal{P}_\mathcal{G} \cap \mathcal{W}^{(\mathfrak{g}_1 \cdots \mathfrak{g}_k)} \neq \varnothing$ in codimension-$k$, where $\mathcal{W}^{(\mathfrak{g}_1 \cdots \mathfrak{g}_k)} := \bigcap_{j=1}^{k} \mathcal{W}^{(\mathfrak{g}_j)}$. The order in which the sequential intersections are taken to reach a codimension-$k$ face is important: all non-trivial ordered intersections differ only by their orientation. Furthermore, each intersection removes regions that are not top-dimensional. Therefore when considering ordered intersections of hyperplanes containing facets of $\mathcal{P}_\mathcal{G}$, they are non-vanishing if and only if

$$\text{co-dim}\big(\mathcal{P}_\mathcal{G} \cap \mathcal{W}^{(\mathfrak{g}_1 \cdots \mathfrak{g}_m)}\big) = m \qquad \forall\, m \leq k. \tag{3.6}$$

At the level of the canonical forms, these sequential intersections correspond to sequential residues on it, and their ordering to the order in which the residues are taken: non-zero sequential residues differing from the ordering return the same lower-dimensional canonical form up to a sign which

reflect the orientation – because the induced orientation on a face changes by a sign if the order of two consecutive intersection is changed, the canonical form gains a $\pm 1$ factor upon such change. From now on $\mathcal{P}_\mathcal{G} \cap \mathcal{W}^{(\mathfrak{g}_1 \cdots \mathfrak{g}_k)}$ will indicate the unordered intersection unless specified. As it was shown in [29], the intersection $\mathcal{P}_\mathcal{G} \cap \mathcal{W}^{(\mathfrak{g}_1 \cdots \mathfrak{g}_k)}$ factorises into subspaces in such a way that the vertices of $\mathcal{P}_\mathcal{G}$ organise in them, and subgroups of them form lower dimensional scattering facets. In order for $\mathcal{P}_\mathcal{G} \cap \mathcal{W}^{(\mathfrak{g}_1 \cdots \mathfrak{g}_k)}$ to be non-empty in codimension-$k$, the sum of the dimensionality of all these subspaces has to match the dimension of a codimension-$k$ face. The characterisation of the full face structure can be expressed in terms of compatibility conditions among the facets which arise from such an analysis [3, 29]:

$$\not{h} + \sum_{\{\mathcal{S}_\mathfrak{g}\}} 1 \; = \; k \tag{3.7}$$

where $\mathcal{S}_\mathfrak{g}$ is a lower-dimensional scattering facet, the sum runs over all the lower-dimension scattering facet in which the intersection among the hyperplanes containing the facets and $\mathcal{P}_\mathcal{G}$ factorises, $\not{h}$ is the number of edges of $\mathcal{G}$ whose the associated polytope vertices are all not on the intersection under consideration, and $k$ is the codimension, and hence the number of hyperplanes, of the intersection. Such compatibility conditions determine the adjoint surface, *i.e.* the geometrical locus of the intersections of the hyperplanes containing the facets *outside* $\mathcal{P}_\mathcal{G}$, which in turn determines the numerator of the canonical form [3, 29]. Also, the knowledge of the compatibility conditions allows for computing the canonical form via a *canonical form triangulation*

$$\Omega(\mathcal{Y}, \mathcal{P}_\mathcal{G}) \; = \; \sum_{j=1}^{n} \Omega(\mathcal{Y}, \mathcal{P}_\mathcal{G}^{(j)}) \tag{3.8}$$

which uses subspaces of the adjoint surface, *i.e.* using only the hyperplanes containing the facets and, consequently, avoiding introducing spurious singularities in the triangulation [29].

## 4   Cosmological correlators from cosmological polytopes

The first principle definition of the cosmological polytopes as the result of the intersection of a collection of $n_e$ originally disconnected triangles in the midpoints of at most two of their three sides, identifies a special 2-plane for each triangle. Such a collection of $n_e \geq 2$ 2-planes provides a specific polytope subdivision of the cosmological polytope. For $n_e = 1$, the 2-plane containing the triangle is just the projective space $\mathbb{P}^2$. This triangle, which is the cosmological polytope associated to the 2-site line graph, is characterised by two special points, $\{\mathbf{x}_{s_e}, \mathbf{x}_{s'_e}\}$: they univocally determine a line identified by the co-vector $\widetilde{\mathcal{W}}_I^{(\mathcal{G})} = \epsilon_{IJK} \mathbf{x}_{s_e}^J \mathbf{x}_{s'_e}^K \sim \tilde{\mathbf{y}}_I^{(e)}$. We will use these special $k$-planes $(k = 1, 2)$ to define an operation on the cosmological polytopes which maps it into a *weighted cosmological polytope* to which a *weighted canonical function* is associated, providing an invariant definition for the cosmological correlators.

In order to fix the ideas, let us consider the simple case of the triangle defined as the convex hull of the vertices

$$\{\mathcal{Z}_1, \mathcal{Z}_2, \mathcal{Z}_3\} \; := \; \{\mathbf{x}_1 - \mathbf{y}_{12} + \mathbf{x}_2, \; \mathbf{x}_1 + \mathbf{y}_{12} - \mathbf{x}_2, \; -\mathbf{x}_1 + \mathbf{y}_{12} + \mathbf{x}_2, \}$$

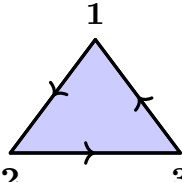 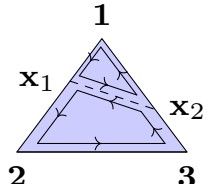 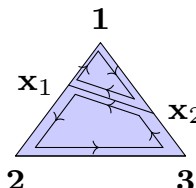

Figure 4: Cosmological and weighted cosmological polytopes. A cosmological polytope is endowed with an orientation determined by positivity conditions among the vertices, in the case above by $\langle 123 \rangle > 0$ (on the left). When considering a polytope subdivision, its elements inherit the orientation from the full polytope in such a way that the codimension-1 boundaries belonging to the elements of the subdivision but not to the polytope are un-oriented and hence spurious (center). Reversing the mutual orientation between the elements of the subdivision provides an orientation to these boundaries turning them into *internal boundaries* (on the right).

associated to the 2-site line graph, together with the line passing through the two points $\mathbf{x}_1$ and $\mathbf{x}_2$. Such a line identifies a polytope subdivision into the quadrilateral $(\mathbf{x}_1 \mathcal{Z}_2 \mathcal{Z}_3 \mathbf{x}_2)$ and the triangle $(\mathbf{x}_2 \mathcal{Z}_1 \mathbf{x}_1)$ which are endowed with an orientation induced from the orientation of the triangle $(\mathcal{Z}_1 \mathcal{Z}_2 \mathcal{Z}_3)$ – note that the round brackets provide an ordered list of vertices. As an operation, it is possible to change the mutual orientation between the two elements of this subdivision: the line $(\mathbf{x}_2 \mathbf{x}_1)$ becomes an *internal boundary* as now both elements of the subdivision have the same orientation on it – see Figure 4. This operation maps the cosmological polytope $\mathcal{P}_\mathcal{G}$ into a *weighted cosmological polytope* $\mathcal{P}_\mathcal{G}^{(w)}$, a special class of the weighted positive geometries introduced in [30].

The weighted cosmological polytope has a *weighted canonical form*, with logarithmic singularities along *all* the boundaries, including the internal ones [8]:

$$\omega\left(\mathcal{Y}, \mathcal{P}_\mathcal{G}^{(w)}\right) = \frac{\mathfrak{n}_1(\mathcal{Y})\langle \mathcal{Y} d^2 \mathcal{Y}\rangle}{\langle \mathcal{Y} 12\rangle\langle \mathcal{Y} 23\rangle\langle \mathcal{Y} 31\rangle\langle \mathcal{Y}\mathbf{x}_2\mathbf{x}_1\rangle} \tag{4.1}$$

where $\mathfrak{n}_1(\mathcal{Y})$ is a polynomial whose degree is fixed to 1 by the requirement of projective invariance and it is the information provided by the subscript.

Note that the presence of a single internal boundary provides a natural polytope subdivision for $\mathcal{P}_\mathcal{G}^{(w)}$ in terms of a quadrilateral and a triangle

$$\mathcal{P}_\mathcal{G}^{(w)} = (\mathbf{x}_1 \mathcal{Z}_2 \mathcal{Z}_3 \mathbf{x}_2) + (\mathbf{x}_1 \mathcal{Z}_1 \mathbf{x}_2) \tag{4.2}$$

where the vertex sequence in each bracket represents the elements of the subdivision, while their order represents the orientation. Then, the weighted canonical form (4.1) can be written in terms of the canonical forms of the elements of the subdivision, which are now polytopes in the usual sense

$$\omega\left(\mathcal{Y}, \mathcal{P}_\mathcal{G}^{(w)}\right) = \frac{\langle \mathcal{Y} A1\rangle\langle \mathcal{Y} d^2 \mathcal{Y}\rangle}{\langle \mathcal{Y}\mathbf{x}_1 2\rangle\langle \mathcal{Y} 23\rangle\langle \mathcal{Y} 3\mathbf{x}_1\rangle\langle \mathcal{Y}\mathbf{x}_2\mathbf{x}_1\rangle} + \frac{\langle \mathbf{x}_1 1\mathbf{x}_2\rangle^2\langle \mathcal{Y} d^2 \mathcal{Y}\rangle}{\langle \mathcal{Y}\mathbf{x}_2\mathbf{x}_1\rangle\langle \mathcal{Y}\mathbf{x}_1 1\rangle\langle \mathcal{Y} 1\mathbf{x}_2\rangle} \tag{4.3}$$

where the first term is the canonical form of the square $(\mathbf{x}_1 \mathcal{Z}_2 \mathcal{Z}_3 \mathbf{x}_2)$, while the second one is the canonical form of the triangle $(\mathbf{x_1} \mathcal{Z}_1 \mathbf{x}_2)$ – the label $A$ appearing in its numerator identifies the point $\mathcal{Z}_A^I := \epsilon^{IJK} \mathcal{W}_J^{(\mathcal{G})} \widetilde{\mathcal{W}}_K^{(\mathcal{G})}$ of the intersection between the lines $\mathcal{W}_I^{(\mathcal{G})} := \epsilon_{IJK} \mathcal{Z}_2^J \mathcal{Z}_3^K$ and $\widetilde{\mathcal{W}}_I^{(\mathcal{G})} := \epsilon_{IJK} \mathbf{x}_2^J \mathbf{x}_1^K$.

---

[8]See Section 4.1 for the formal definition of the canonical form of weighted polytopes.

In order to single out the peculiarities of the weighted cosmological polytope $\mathcal{P}_{\mathcal{G}}^{(w)}$, let us pause and analyse in detail the weighted canonical form both as expressed in (4.3) and later in its invariant form (4.1). General features of the usual polytopes are that: $i$) any boundary is a lower-dimensional polytope whose canonical form is given by the residue of the canonical form of the original polytope along the given boundary; $ii$) the non-vanishing maximal residues of the canonical form have all the same normalisation, up to a sign. Let us now turn to $\mathcal{P}_{\mathcal{G}}^{(w)}$ and consider the boundary $\mathcal{P}_{\mathcal{G}}^{(w)} \cap \mathcal{W}^{(\mathcal{G})}$ which is just a segment with vertices $\mathcal{Z}_2$ and $\mathcal{Z}_3$ [9]. The associated pole in the weighted cosmological polytope is given by $\mathcal{Y} \cdot \mathcal{W}^{(\mathcal{G})} = \langle \mathcal{Y}23 \rangle = 0$. From (4.3), it is straightforward to see that it is a boundary of just one of the two elements of the polytope subdivision, and it does not show internal boundaries. Then

$$\mathrm{Res}_{\mathcal{W}^{(\mathcal{G})}} \left\{ \omega(\mathcal{Y}, \mathcal{P}_{\mathcal{G}}^{(w)}) \right\} = \frac{\langle \mathcal{Z}_{\mathcal{G}}23 \rangle \langle \mathcal{Z}_{\mathcal{G}} \mathcal{Y}_{\mathcal{G}} d\mathcal{Y}_{\mathcal{G}} \rangle}{\langle \mathcal{Y}_{\mathcal{G}} \mathcal{Z}_{\mathcal{G}}2 \rangle \langle \mathcal{Y}_{\mathcal{G}}3\mathcal{Z}_{\mathcal{G}} \rangle} \tag{4.4}$$

where the vector $\mathcal{Z}_{\mathcal{G}}$ identifies the restriction on $\mathcal{W}^{(\mathcal{G})}$, and $\mathcal{Y}_{\mathcal{G}}$ is a generic point on the restriction. Note that the right-hand side of (4.4) is precisely the canonical form of the segment with boundaries $\mathcal{Z}_2$ and $\mathcal{Z}_3$. Let us now consider the internal boundary identified by $\widetilde{\mathcal{W}}^{(\mathcal{G})}$ and let us consider its residue. Again, the expression (4.3) shows that it is a boundary of both the elements of the subdivision, and the residue gets a contribution from both terms: for each term the intersection with the line $\widetilde{\mathcal{W}}^{(\mathcal{G})}$ is the segment $(\mathbf{x}_2\mathbf{x}_1)$

$$\begin{aligned}
\mathrm{Res}_{\widetilde{\mathcal{W}}^{(\mathcal{G})}} \left\{ \omega\left(\mathcal{Y}, \mathcal{P}_{\mathcal{G}}^{(w)}\right) \right\} &= \frac{\langle \widetilde{\mathcal{Z}}_{\mathcal{G}}\mathbf{x}_2\mathbf{x}_1 \rangle \langle \widetilde{\mathcal{Z}}_{\mathcal{G}} \widetilde{\mathcal{Y}}_{\mathcal{G}} d\widetilde{\mathcal{Y}}_{\mathcal{G}} \rangle}{\langle \widetilde{\mathcal{Y}}_{\mathcal{G}} \widetilde{\mathcal{Z}}_{\mathcal{G}}\mathbf{x}_2 \rangle \langle \widetilde{\mathcal{Y}}_{\mathcal{G}}\mathbf{x}_2\widetilde{\mathcal{Z}}_{\mathcal{G}} \rangle} + \frac{\langle \widetilde{\mathcal{Z}}_{\mathcal{G}}\mathbf{x}_2\mathbf{x}_1 \rangle \langle \widetilde{\mathcal{Z}}_{\mathcal{G}} \widetilde{\mathcal{Y}}_{\mathcal{G}} d\widetilde{\mathcal{Y}}_{\mathcal{G}} \rangle}{\langle \widetilde{\mathcal{Y}}_{\mathcal{G}} \widetilde{\mathcal{Z}}_{\mathcal{G}}\mathbf{x}_2 \rangle \langle \widetilde{\mathcal{Y}}_{\mathcal{G}}\mathbf{x}_1\widetilde{\mathcal{Z}}_{\mathcal{G}} \rangle} = \\
&= 2 \frac{\langle \widetilde{\mathcal{Z}}_{\mathcal{G}}\mathbf{x}_2\mathbf{x}_1 \rangle \langle \widetilde{\mathcal{Z}}_{\mathcal{G}} \widetilde{\mathcal{Y}}_{\mathcal{G}} d\widetilde{\mathcal{Y}}_{\mathcal{G}} \rangle}{\langle \widetilde{\mathcal{Y}}_{\mathcal{G}} \widetilde{\mathcal{Z}}_{\mathcal{G}}\mathbf{x}_2 \rangle \langle \widetilde{\mathcal{Y}}_{\mathcal{G}}\mathbf{x}_1\widetilde{\mathcal{Z}}_{\mathcal{G}} \rangle}
\end{aligned} \tag{4.5}$$

where $\widetilde{\mathcal{Z}}_{\mathcal{G}}$ identifies the restriction on $\widetilde{\mathcal{W}}^{(\mathcal{G})}$. Notice that each term in the first line in (4.5) is the canonical form of the segment $(\mathbf{x}_2\mathbf{x}_1)$, and the fact that their relative sign is the same is a reflection that they share the same orientation. Comparing the two formulas (4.5) and (4.4), it is straightforward to see that both of them provide the canonical form of a segment – all the boundaries are segments – but their relative normalisation is different which is the manifestation of the fact that $(\mathbf{x}_2\mathbf{x}_1)$ is an internal boundary. Also, the maximal residues which can be computed from (4.4) and the ones from (4.5) inherit the difference in the relative normalisation. The boundaries of the weighted cosmological polytopes are therefore characterised by *weights*, which codify the presence of internal boundaries.

Let us further consider the other two codimension-1 boundaries, identified by the co-vectors

$$\begin{aligned}
\mathcal{W}_{\mathrm{I}}^{(\mathfrak{g}_2)} &:= \epsilon_{\mathrm{IJK}} \mathcal{Z}_1^J \mathcal{Z}_2^K \sim \epsilon_{\mathrm{IJK}} \mathcal{Z}_1^J \mathbf{x}_1^K \sim \epsilon_{\mathrm{IJK}} \mathbf{x}_1^J \mathcal{Z}_2^K, \\
\mathcal{W}_{\mathrm{I}}^{(\mathfrak{g}_1)} &:= \epsilon_{\mathrm{IJK}} \mathcal{Z}_3^J \mathcal{Z}_1^K \sim \epsilon_{\mathrm{IJK}} \mathcal{Z}_3^J \mathbf{x}_2^K \sim \epsilon_{\mathrm{IJK}} \mathbf{x}_2^J \mathcal{Z}_1^K
\end{aligned} \tag{4.6}$$

where the $\sim$ indicate projective identities due to the linear dependence of $\mathbf{x}_1$ and $\mathbf{x}_2$ from the vertices $\{\mathcal{Z}_j, j=1,2,3\}$, *i.e.* $2\mathbf{x}_1 \sim \mathcal{Z}_1 + \mathcal{Z}_2$ and $2\mathbf{x}_2 \sim \mathcal{Z}_3 + \mathcal{Z}_1$. Both of these codimension-1 boundaries $\{\mathcal{P}_{\mathcal{G}} \cap \mathcal{W}^{(\mathfrak{g}_j)}, j=1,2\}$ show an internal boundary, given by the intersection of the lines $\mathcal{W}^{(\mathfrak{g}_j)}$ and the internal boundary of $\mathcal{P}_{\mathcal{G}}^{(w)}$ – they are the two vertices $\mathbf{x}_2$ and $\mathbf{x}_1$ respectively: these boundaries are

---

[9] We will discuss later how to determine in general which vertices are on a given face for the case of the weighted cosmological polytopes. For the time being, while we are discussing the simplest example, this can be seen directly from Figure 4.

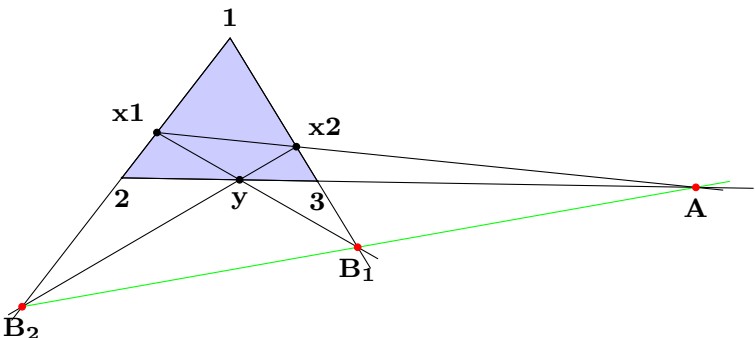

Figure 5: Adjoint surface for a weighted cosmological polytope. For the weighted cosmological polytope associated to a 2-site line graph, it is given by the green line $(AB_1B_2)$ with just one of its points, $\mathcal{Z}_A^I$, reflecting a straightforward generalisation of the usual definition of the adjoint for polytopes. Further conditions are needed, and they fix the points $\mathcal{Z}_{B_1}^I$ and $\mathcal{Z}_{B_2}^I$.

still weighted polytopes. From the perspective of the canonical form, the lines $\mathcal{W}^{(\mathfrak{g}_j)}$ intersects both the elements of the subdivision in (4.3): $\mathcal{Y} \cdot \mathcal{W}^{(\mathfrak{g}_1)} \sim \langle \mathcal{Y} 3 \mathbf{x}_2 \rangle \sim \langle \mathcal{Y} \mathbf{x}_2 1 \rangle$ and $\mathcal{Y} \cdot \mathcal{W}^{(\mathfrak{g}_2)} \sim \langle \mathcal{Y} 1 \mathbf{x}_1 \rangle \sim \langle \mathcal{Y} \mathbf{x}_1 2 \rangle$. Its residue along them turns out to be

$$
\begin{aligned}
\text{Res}_{\mathcal{W}^{(\mathfrak{g}_1)}}\{\omega(\mathcal{Y}, \mathcal{P}_\mathcal{G})\} &= \frac{\langle \mathcal{Z}_{\mathfrak{g}_1} 3 \mathbf{x}_2 \rangle \langle \mathcal{Z}_{\mathfrak{g}_1} \mathcal{Y}_{\mathfrak{g}_1} d\mathcal{Y}_{\mathfrak{g}_1} \rangle}{\langle \mathcal{Y}_{\mathfrak{g}_1} \mathcal{Z}_{\mathfrak{g}_1} 3 \rangle \langle \mathcal{Y}_{\mathfrak{g}_1} \mathbf{x}_2 \mathcal{Z}_{\mathfrak{g}_1} \rangle} + \frac{\langle \mathcal{Z}_{\mathfrak{g}_1} 1 \mathbf{x}_2 \rangle \langle \mathcal{Z}_{\mathfrak{g}_1} \mathcal{Y}_{\mathfrak{g}_1} d\mathcal{Y}_{\mathfrak{g}_1} \rangle}{\langle \mathcal{Y}_{\mathfrak{g}_1} \mathcal{Z}_{\mathfrak{g}_1} 1 \rangle \langle \mathcal{Y}_{\mathfrak{g}_1} \mathbf{x}_2 \mathcal{Z}_{\mathfrak{g}_1} \rangle}, \\
\text{Res}_{\mathcal{W}^{(\mathfrak{g}_2)}}\{\omega(\mathcal{Y}, \mathcal{P}_\mathcal{G})\} &= \frac{\langle \mathcal{Z}_{\mathfrak{g}_2} \mathbf{x}_1 2 \rangle \langle \mathcal{Z}_{\mathfrak{g}_2} \mathcal{Y}_{\mathfrak{g}_2} d\mathcal{Y}_{\mathfrak{g}_2} \rangle}{\langle \mathcal{Y}_{\mathfrak{g}_2} \mathcal{Z}_{\mathfrak{g}_2} 2 \rangle \langle \mathcal{Y}_{\mathfrak{g}_2} \mathbf{x}_1 \mathcal{Z}_{\mathfrak{g}_2} \rangle} + \frac{\langle \mathcal{Z}_{\mathfrak{g}_1} \mathbf{x}_1 2 1 \rangle \langle \mathcal{Z}_{\mathfrak{g}_2} \mathcal{Y}_{\mathfrak{g}_2} d\mathcal{Y}_{\mathfrak{g}_2} \rangle}{\langle \mathcal{Y}_{\mathfrak{g}_2} \mathcal{Z}_{\mathfrak{g}_2} \mathbf{x}_2 \rangle \langle \mathcal{Y}_{\mathfrak{g}_2} 1 \mathcal{Z}_{\mathfrak{g}_2} \rangle}
\end{aligned}
\tag{4.7}
$$

Note that, for each residue, each term is the canonical form of a segment that shares a boundary with the other. This manifests itself in the presence of a common pole, at $\langle \mathcal{Y}_{\mathfrak{g}_j} \mathbf{x}_k \mathcal{Z}_{\mathfrak{g}_j} \rangle = 0$ ($j, k = 1, 2, k \neq j$), whose residues are the same, including their signs – this is the same situation we encountered for the internal boundary $(\mathbf{x}_2 \mathbf{x}_1)$ of $\mathcal{P}_\mathcal{G}^{(w)}$, and the point $\mathbf{x}_k$ constitutes an internal boundary for these segments. This shows that the facets of $\mathcal{P}_\mathcal{G}^{(w)}$ which are intersected by its internal boundaries are still weighted polytopes, while those which are not – e.g. $\mathcal{P}_\mathcal{G}^{(w)} \cap \mathcal{W}^{(\mathcal{G})}$ – are ordinary polytopes

The analysis we have performed so far relied on a specific polytope subdivision and the way that the weighted canonical form decomposes under it. However, it is desirable to have an invariant way to determine it, i.e. a way that does not rely on a polytope subdivision. In the case of the usual polytopes, while the knowledge of its codimension-1 boundaries fixes the denominator of the canonical form, the numerator is fixed by the so-called adjoint surface, which is defined as the locus of the intersection of the hyperplanes containing the facets of the polytope outside the polytope itself. It fixes the higher-codimension boundary structure: if the intersection of $k$ hyperplanes with the polytope is non-empty in codimension-$k$, then such an intersection is a codimension-$k$ boundary of the polytope; otherwise, it belongs to the adjoint surface. From the perspective of the canonical form, this is equivalent to the statement that its multiple residue along the $k$ hyperplanes is non-zero (the intersection is non-empty in codimension-$k$) or zero (the intersection is empty in codimension-$k$) [3, 29, 31, 38].

Let us examine, for the specific case under consideration, the relation between the numerator $\mathfrak{n}_1(\mathcal{Y})$ of the weighted canonical form (4.1) and the geometry. As already mentioned, the degree

of the polynomial $\mathfrak{n}_1(\mathcal{Y})$ is fixed to 1 by the requirement of projective invariance. Being a linear polynomial, $\mathfrak{n}_1(\mathcal{Y})$ is identified by a co-vector $\mathcal{C}_I$ such that $\mathfrak{n}_1(\mathcal{Y}) = \mathcal{C}_I \mathcal{Y}^I$. If we were to follow the definition of the adjoint surface as for the usual polytopes, then just the lines $\mathcal{W}^{(\mathcal{G})}$ and $\widetilde{\mathcal{W}}^{(\mathcal{G})}$ intersects each other in codimension-1 outside of $\mathcal{P}_{\mathcal{G}}^{(w)}$ – see Figure 5. Such intersection is identified by the vector $\mathcal{Z}_A^I$

$$\mathcal{Z}_A^I \;=\; \epsilon^{IJK} \mathcal{W}_J^{(\mathcal{G})} \widetilde{\mathcal{W}}_K^{(\mathcal{G})}. \tag{4.8}$$

The knowledge of this point fixes one of the two degrees of freedom of the linear polynomial, and the co-vector $\mathcal{C}_I$ can be written as

$$\mathcal{C}_I \;:=\; \epsilon_{IJK} \mathcal{Z}_A^J \mathcal{Z}_B^K \tag{4.9}$$

with $\mathcal{Z}_A^J$ given by (4.8) and $\mathcal{Z}_B^J$ yet to be determined.

Recall that the weighted cosmological polytope is defined from the cosmological polytope by considering a line identified by the two special points $\mathbf{x}_1$ and $\mathbf{x}_2$, taking its polytope subdivision with respect to such a line and changing the relative orientation. Together with $\mathbf{x}_1$ and $\mathbf{x}_2$, there is a third special point, $\mathbf{y}$. The triple $\{\mathbf{x}_1, \mathbf{y}, \mathbf{x}_2\}$ define a triple of lines $\{(\mathbf{x}_2\mathbf{x}_1), (\mathbf{x}_1\mathbf{y}), (\mathbf{y}\mathbf{x}_2)\}$. The first of them identifies the internal boundary. The other two intersect the lines $(\mathcal{Z}_3\mathcal{Z}_1)$ and $(\mathcal{Z}_1\mathcal{Z}_2)$ respectively. Let these intersections be $\mathcal{Z}_{B_1}$ and $\mathcal{Z}_{B_2}$. It turns out that the points $\mathcal{Z}_A$, $\mathcal{Z}_{B_1}$ and $\mathcal{Z}_{B_2}$ lie on the same line, and

$$\mathcal{C}_I \;=\; \epsilon_{IJK} \mathcal{Z}_A^J \mathcal{Z}_{B_j}^J, \qquad j = 1, 2 \tag{4.10}$$

fixing the adjoint for the weighted cosmological polytope under consideration.

Some comments are now in order. Notice that just $\mathcal{Z}_A$ respects a straightforward generalisation of the adjoint as the locus of the intersection of the (both external and internal) facets outside of the weighted polytope. The points $\mathcal{Z}_{B_1}$ and $\mathcal{Z}_{B_2}$ instead are both the intersection of one of the external boundaries with a line identified by one of the pair of midpoints $\{(\mathbf{x}_j, \mathbf{y}) \, j = 1, 2\}$, and none of them is an internal boundary of the original weighed cosmological polytope. In this particular case, the usual definition of adjoint allows to identify just one of the points on it, while the other point needed to completely fix it was obtained by inspection. Taking these other types of conditions as part of the definition of the adjoint, we will extend the discussion to arbitrary weighted cosmological polytopes in Section 4.3.

Let us conclude with one more comment. Beginning with the polytope subdivision (4.2) of $\mathcal{P}_{\mathcal{G}}^{(w)}$ into the quadrilateral $(\mathbf{x}_1\mathcal{Z}_2\mathcal{Z}_3\mathbf{x}_2)$ and the triangle $(\mathbf{x}_1\mathcal{Z}_1\mathbf{x}_2)$, we can further triangulate the quadrilateral. Let us consider its signed triangulation via $\mathcal{Z}_1$. Then, $\mathcal{P}_{\mathcal{G}}^{(w)}$ can be written as

$$\begin{aligned}
\mathcal{P}_{\mathcal{G}}^{(w)} &= (\mathcal{Z}_1\mathcal{Z}_2\mathcal{Z}_3) + (\mathbf{x}_1\mathcal{Z}_1\mathbf{x}_2) + (\mathbf{x}_1\mathcal{Z}_1\mathbf{x}_2) = \\
&= \mathcal{P}_{\mathcal{G}} - 2\,(\mathcal{Z}_1\mathbf{x}_1\mathbf{x}_2)
\end{aligned} \tag{4.11}$$

where the quadrilateral is decomposed in the original cosmological polytope $\mathcal{P}_{\mathcal{G}} \equiv (\mathcal{Z}_1\mathcal{Z}_2\mathcal{Z}_3)$ and the very same triangle $(\mathbf{x}_1\mathcal{Z}_1\mathbf{x}_2)$ – the second line is obtained by reversing the orientation of the triangle which produces a minus sign. Remarkably, the triangulation (4.11) turns out to recover the diagrammatic rule (2.27) for computing the cosmological correlator associated with the 2-site line graph, where the form of the first term corresponds to the wavefunction and the second term coincides

with the contribution from the disconnected graphs. Taking the canonical forms of the quadrilateral and of the triangle as given in (4.3), and using the local patch $\mathcal{Y} := (x_1, y, x_2)$, the triangulation (4.11) on the weighted canonical form of $\mathcal{P}_{\mathcal{G}}^{(w)}$ reads

$$\omega(\mathcal{Y}, \mathcal{P}_{\mathcal{G}}^{(w)}) = \left[ \frac{2}{(x_1 + x_2)(x_1 + y)(y + x_2)} + \frac{2}{y(x_1 + y)(y + x_2)} \right] \frac{dx_1 \wedge dy \wedge dx_2}{\text{Vol}\{GL(1)\}} \tag{4.12}$$

with the canonical function, which is included in the square brackets, reproducing (2.28).

Notice that in this picture, the expression of the cosmological correlators in terms of wavefunction graphs (2.14) is associated to one of the possible triangulations of $\mathcal{P}_{\mathcal{G}}^{(w)}$. The polytope subdivision (4.2) provides instead a novel representation, which explicitly reads:

$$\omega\left(\mathcal{Y}, \mathcal{P}_{\mathcal{G}}^{(w)}\right) = \left[ \frac{x_1 + x_2 + 2y}{(x_1 + x_2)(x_1 + y)(y + x_2)y} + \frac{1}{y(x_1 + y)(y + x_2)} \right] \frac{dx_1 \wedge dy \wedge dx_2}{\text{Vol}\{GL(1)\}}. \tag{4.13}$$

The weighted cosmological polytope $\mathcal{P}_{\mathcal{G}}^{(w)}$ provides an invariant formulation of the cosmological correlators, with other representations of the latter encoded in the different ways of triangulating it.

In this first part of the Section, we have discussed in detail the simplest case of a weighted cosmological polytope, which has been defined via an operation on the cosmological polytope associated to the 2-site graph. We also inferred how to characterise it, including compatibility conditions among all its boundaries

In the rest of this Section, we will extend the discussion to the weighted geometries obtained from cosmological polytopes associated to arbitrary graphs. They turn out to encode the cosmological correlators for arbitrary graphs.

## 4.1 Weighted cosmological polytopes and their canonical forms

Let us go back to the general definition of the weighted cosmological polytope $\mathcal{P}_{\mathcal{G}}^{(w)}$ as the geometry obtained as a polytope subdivision identified by the 2-planes containing the triangles which define $\mathcal{P}_{\mathcal{G}}$ and changing the mutual orientation of the elements of the subdivision along their codimension-1 boundaries which are not facets of $\mathcal{P}_{\mathcal{G}}$. In the next paragraphs, we will provide a formal definition of this operation and characterise the resulting geometry, which turns out to be a special class of the weighted positive geometries introduced in [30]. We will follow the following path: we will introduce a characterisation of the cosmological polytopes in terms of a *weight function* and an *orientation* and we will define the *weighted cosmological polytopes* by modifying the definition of such a pair.

**Cosmological polytopes in terms of orientation and weights** – Let us begin with considering a cosmological polytope $\mathcal{P}_{\mathcal{G}}$ and, as usual, let $\{\mathcal{W}^{(\mathfrak{g})}, \mathfrak{g} \subseteq \mathcal{G}\}$ be the set of hyperplanes containing its facets. A generic point is inside $\mathcal{P}_{\mathcal{G}}$ if $\mathcal{Y} \cdot \mathcal{W}^{(\mathfrak{g})} \geq 0 \, \forall \, \mathfrak{g} \subseteq \mathcal{G}$ [10]. Then, the orientation of $\mathcal{P}_{\mathcal{G}}$ can be described by the top form

$$O\left(\mathcal{Y}, \mathcal{P}_{\mathcal{G}}\right) = \sigma \left\langle \mathcal{Y} d^{n_s + n_e - 1} \mathcal{Y} \right\rangle \tag{4.14}$$

where $\sigma = \pm 1$. For $\sigma = 1$, the cosmological polytope $\mathcal{P}_{\mathcal{G}}$ is said to be *positively oriented* – since now on we will set the cosmological polytopes to have such a convention. The orientation constitutes

---

[10]If the equality is satisfied than a point is on a boundary of $\mathcal{P}_{\mathcal{G}}$

an equivalence class of top forms under positive rescaling. It is also possible to define a piece-wise constant function, the *weight function*, such as

$$w\left(\mathcal{Y}, \mathcal{P}_\mathcal{G}\right) \;=\; \prod_{\mathfrak{g} \subseteq \mathcal{G}} \vartheta\left(\mathcal{Y} \cdot \mathcal{W}^{(\mathfrak{g})}\right). \tag{4.15}$$

which is 1 inside $\mathcal{P}_\mathcal{G}$ and 0 outside. Note that the boundaries of $\mathcal{P}_\mathcal{G}$ correspond to the regions where the pair $(w, O)$ is discontinuous. Importantly, the pair $(w, O)$ uniquely identifies – up to the choice for $\sigma$ in (4.14) – the cosmological polytope $\mathcal{P}_\mathcal{G}$ and it is such that

$$(w, O) \;\sim\; (-w, -O) \tag{4.16}$$

*i.e.* the pair obtained by changing sign locally for any $\mathcal{Y}$ to both the weight function and the orientation still describes the same cosmological polytope $\mathcal{P}_\mathcal{G}$. The orientation of the cosmological polytope induces an orientation on its boundaries. For a codimension-1 boundary $\mathcal{P}_\mathcal{G} \cap \mathcal{W}^{(\mathfrak{g})}$, it is possible to take the local patch where $q_\mathfrak{g}(\mathcal{Y}) := \mathcal{Y} \cdot \mathcal{W}^{(\mathfrak{g})}$ is a local coordinate. Then one can write:

$$O(\mathcal{Y}, \mathcal{P}_\mathcal{G}) \;=\; dq_\mathfrak{g} \wedge O(\mathcal{Y}_\mathfrak{g}, \mathcal{P}_\mathcal{G} \cap \mathcal{W}^{(\mathfrak{g})}) \tag{4.17}$$

$\mathcal{Y}_\mathfrak{g}$ is the restriction of $\mathcal{Y}$ on the codimension-1 hyperplane $\mathcal{W}^{(\mathfrak{g})}$. The weight function of the codimension-1 boundary $\mathcal{P}_\mathcal{G} \cap \mathcal{W}^{(\mathfrak{g})}$ is instead given by the discontinuity of the weight function (4.15) along the hyperplane $\mathcal{W}^{(\mathfrak{g})}$:

$$w\left(\mathcal{Y}, \mathcal{P}_\mathcal{G} \cap \mathcal{W}^{(\mathfrak{g})}\right) \;=\; \mathrm{Disc}_{\mathcal{W}^{(\mathfrak{g})}}\left\{w\left(\mathcal{Y}, \mathcal{P}_\mathcal{G}\right)\right\} := \lim_{q_\mathfrak{g} \to 0^+} w\left(\mathcal{Y}, \mathcal{P}_\mathcal{G}\right) - \lim_{q_\mathfrak{g} \to 0^-} w\left(\mathcal{Y}, \mathcal{P}_\mathcal{G}\right). \tag{4.18}$$

As the weight function of $\mathcal{P}_\mathcal{G}$ vanishes for $q_\mathfrak{g}$ negative, then the weight function for its codimension-1 boundaries is always 1. For higher codimension boundaries, it can be $\pm 1$ depending on how these boundaries are approached.

As a short-hand notation for $w\left(\mathcal{Y}, \mathcal{P}_\mathcal{G} \cap \mathcal{W}^{(\mathfrak{g})}\right)$ and $O\left(\mathcal{Y}, \mathcal{P}_\mathcal{G} \cap \mathcal{W}^{(\mathfrak{g})}\right)$, let us use $w_\mathfrak{g}$ and $O_\mathfrak{g}$, respectively. Then, in terms of the canonical form

$$\omega\left(w_\mathfrak{g}, O_\mathfrak{g}\right) \;=\; \mathrm{Res}_{\mathcal{W}^{(\mathfrak{g})}}\omega\left(w, O\right) \tag{4.19}$$

and the maximal residues of the canonical form of the cosmological polytope along the codimension-$(n_s + n_e - 1)$ hyperplanes satisfying the compatibility conditions in [29] are given by

$$\begin{aligned}
\omega\left(w_{\mathfrak{g}_1 \ldots \mathfrak{g}_{N-1}}, O_{\mathfrak{g}_1 \ldots \mathfrak{g}_{N-1}}\right) &= \mathrm{Res}_{\mathcal{W}^{(\mathfrak{g}_1)}} \ldots \mathrm{Res}_{\mathcal{W}^{(\mathfrak{g}_{N-1})}}\left\{\omega\left(w, O\right)\right\} = \\
&= \mathrm{sign}\left\{O_{\mathfrak{g}_1 \ldots \mathfrak{g}_{N-1}}\right\} \mathrm{Disc}_{\mathcal{W}^{(\mathfrak{g}_1)}} \ldots \mathrm{Disc}_{\mathcal{W}^{(\mathfrak{g}_{N-1})}}\left\{w\right\} = \pm 1
\end{aligned} \tag{4.20}$$

where $w_{\mathfrak{g}_1 \ldots \mathfrak{g}_{N-1}}$ and $O_{\mathfrak{g}_1 \ldots \mathfrak{g}_{N-1}}$ are respectively the weight function and the orientation of $\mathcal{P}_\mathcal{G} \cap \mathcal{W}^{(\mathfrak{g}_1)} \cap \ldots \cap \mathcal{W}^{(\mathfrak{g}_{N-1})} \neq \varnothing$, and $N := n_s + n_e$. This is the statement that the maximal residues for the cosmological polytope are $\pm 1$.

Let now consider the polytope subdivision of $\mathcal{P}_\mathcal{G}$ induced by its definition as intersections among $n_e$ triangles. Let $\left\{\mathcal{P}_\mathcal{G}^{(j)}, j = 1, \ldots, 2^{n_e} \;\middle|\; \bigcup_{j=1}^{2^{n_e}} \mathcal{P}_\mathcal{G}^{(j)} = \mathcal{P}_\mathcal{G}\right\}$ be the collection of elements of such a subdivision. Let also $\{\widetilde{\mathcal{W}}^{(\mathfrak{g}_e)}, e \in \mathcal{E}\}$ be the set of hyperplanes containing the codimension-1 boundaries

of this polytope subdivision which are not facets of $\mathcal{P}_{\mathcal{G}}$. Such hyperplanes intersect the polytope in its interior so that both the positive and negative half-spaces they determine have a non-empty intersection with $\mathcal{P}_{\mathcal{G}}$. Let us consider the local patch where $q_{\mathfrak{g}_e}(\mathcal{Y}) := \mathcal{Y} \cdot \widetilde{\mathcal{W}}^{(\mathfrak{g}_e)}$ is a local coordinate, and let $\sigma_e = \mathrm{sign}\{q_{\mathfrak{g}_e}\}$ then

$$O^{(\sigma_e)}(\mathcal{Y}, \mathcal{P}_{\mathcal{G}}) \;=\; \sigma_e \, dq_{\mathfrak{g}_e} \wedge O^{(\sigma_e)}(\mathcal{Y}, \mathcal{P}_{\mathcal{G}} \cap \widetilde{\mathcal{W}}^{(\mathfrak{g}_e)}) \tag{4.21}$$

and the weighted orientation of the boundary has a contribution from the restriction of both the positive and negative half-spaces. As far as the weight function of $\mathcal{P}_{\mathcal{G}} \cap \widetilde{\mathcal{W}}^{(\mathfrak{g}_e)}$ is concerned, its definition as the discontinuity of $w(\mathcal{Y}, \mathcal{P}_{\mathcal{G}})$ along the hyperplane $\widetilde{\mathcal{W}}^{(\mathfrak{g}_e)}$ implies that it is zero: the weight function of the cosmological polytope is equal to 1 anywhere in its interior. In the characterisation of $\mathcal{P}_{\mathcal{G}}$ in terms of weights and orientation, this is the statement that these boundaries are actually spurious.

The pair $(w, O)$ for $\mathcal{P}_{\mathcal{G}}$ can then be expressed in terms of the collection of pairs $\{(w_j, O_j), j = 1, \ldots, 2^{n_e}\}$ of the polytope subdivision elements $\{\mathcal{P}_{\mathcal{G}}^{(j)}, j = 1 \ldots 2^{n_e}\}$

$$(w, O) \;=\; \left( \sum_{j=1}^{2^{n_e}} w_j, \, \{\sim O_j\} \right) \tag{4.22}$$

where the second argument on the right-hand side just indicates that all the orientations are in the same equivalence class, and the arguments of the weights and the orientations have been suppressed for notational legibility. This structure reflects itself into the regular triangulations and the corresponding canonical form triangulations

$$\omega\,(w, O) \;=\; \sum_{j=1}^{2^{n_e}} \omega\,(w_j, O_j) \tag{4.23}$$

where the argument of the canonical forms indicates just the characterisation of the polytopes involved in terms of weights and orientations – hence, $\omega(w, O) \equiv \omega(\mathcal{Y}, \mathcal{P}_{\mathcal{G}})$, and $\omega(w_j, O_j) \equiv \omega(\mathcal{Y}, \mathcal{P}_{\mathcal{G}}^{(j)})$.

A similar discussion holds whether were we to choose a collection of polytopes sign-triangulating $\mathcal{P}_{\mathcal{G}}$ via a subspace of its adjoint as in [29], or in any other possible signed triangulation, with (4.22) which can be generalised to

$$(w, O) \;=\; \left( \sum_{j \in \{\mathcal{P}_{\mathcal{G}}^{(j)}\}} w_j, \, \{\sim O_j\} \right) \tag{4.24}$$

$\{\mathcal{P}_{\mathcal{G}}^{(j)}\}$ being the collection of polytopes which provides any signed triangulation of $\mathcal{P}_{\mathcal{G}}$ and $j$ the index which labels its elements – the additive structure of the weight function (4.24) generalises the notion of signed triangulation.

So far, we have just provided an alternative description of the cosmological polytopes in terms of the weight function and the orientation. This allows us to transparently formalise the operation that maps a cosmological polytope into a weighted cosmological polytope: it is the geometric equivalent of (2.10) in perturbation theory.

**Weighted cosmological polytopes** – The definition of the weighted cosmological polytope we gave so far, starts with a cosmological polytope associated with a given graph, considers its polytope

subdivision via the hyperplane containing the triangles whose intersections define the cosmological polytope itself, and finally reverses the mutual orientation of the elements of the subdivision along such hyperplanes. What does this practically mean?

Let us consider the collection $\{\widetilde{\mathcal{W}}^{(\mathfrak{g}_e)},\, e \in \mathcal{E}\}$ of hyperplanes determining such a subdivision of $\mathcal{P}_{\mathcal{G}}$. Then, a weighted cosmological polytope $\mathcal{P}_{\mathcal{G}}^{(w)}$ is defined a cosmological polytope $\mathcal{P}_{\mathcal{G}}$ equipped with a weight function $w(\mathcal{Y},\, \mathcal{P}_{\mathcal{G}}^{(w)})$ and a orientation $O(\mathcal{Y},\, \mathcal{P}_{\mathcal{G}}^{(w)})$ defined as

$$
\begin{aligned}
w(\mathcal{Y},\, \mathcal{P}_{\mathcal{G}}^{(w)}) \;&=\; w(\mathcal{Y},\, \mathcal{P}_{\mathcal{G}}) \\
O\left(\mathcal{Y},\, \mathcal{P}_{\mathcal{G}}^{(w)}\right) \;&=\; \text{sign}\left\{ \prod_{e \in \mathcal{E}} \left( \mathcal{Y} \cdot \widetilde{\mathcal{W}}^{(\mathfrak{g}_e)} \right) \right\} O\left(\mathcal{Y},\, \mathcal{P}_{\mathcal{G}}\right).
\end{aligned}
\tag{4.25}
$$

This definition implies that the weight function of the weighted cosmological polytope is zero outside $\mathcal{P}_{\mathcal{G}}$, while equal 1 inside it, while the orientation flip sign along any hyperplane $\widetilde{\mathcal{W}}^{(\mathfrak{g}_e)}$ ($e \in \mathcal{E}$). Because the weight function and the orientation are defined up to (4.16), we can equivalently define $\mathcal{P}_{\mathcal{G}}^{(w)}$ as

$$
\begin{aligned}
w(\mathcal{Y},\, \mathcal{P}_{\mathcal{G}}^{(w)}) \;&=\; \text{sign}\left\{ \prod_{e \in \mathcal{E}} \left( \mathcal{Y} \cdot \widetilde{\mathcal{W}}^{(\mathfrak{g}_e)} \right) \right\} w(\mathcal{Y},\, \mathcal{P}_{\mathcal{G}}) \\
O\left(\mathcal{Y},\, \mathcal{P}_{\mathcal{G}}^{(w)}\right) \;&=\; O\left(\mathcal{Y},\, \mathcal{P}_{\mathcal{G}}\right).
\end{aligned}
\tag{4.26}
$$

Given any polytope, a more general weight function and orientation can be assigned to it, in which case one would just talk of *weighted polytopes*. The specific choice (4.25) is motivated by the special role that the collection of 2-planes $\left\{ \bigcap_{e' \in \mathcal{E} \setminus \{e\}} \widetilde{\mathcal{W}}^{(\mathfrak{g}_{e'})},\, e \in \mathcal{E} \right\}$ plays in the cosmological polytope construction as they contain its building blocks.

In this formalism, the boundaries of the geometry are defined whenever the pair $(w,\, O)$ is discontinuous. From the definition (4.25), it is straightforward to note that $\mathcal{P}_{\mathcal{G}}^{(w)}$ shares all the boundaries of $\mathcal{P}_{\mathcal{G}}$ but also have novel boundaries, identified by the hyperplanes $\widetilde{\mathcal{W}}^{(\mathfrak{g}_e)}$. For codimension-1 boundaries, the orientation induced on $\mathcal{P}_{\mathcal{G}} \cap \widetilde{\mathcal{W}}^{(\mathfrak{g}_e)}$ by the positive and negative half-space is the same, and the weight function is non-zero. This can be seen easily as follows. Recall that for a codimension-1 boundary, in this case, $\mathcal{P}_{\mathcal{G}}^{(w)} \cap \widetilde{\mathcal{W}}^{(\mathfrak{g}_e)}$, the orientation induced on it can be seen by considering a local patch where $q_{\mathfrak{g}_e}(\mathcal{Y}) := \mathcal{Y} \cdot \widetilde{\mathcal{W}}^{(\mathfrak{g}_e)}$ to be

$$
O^{(\sigma_e)}\left(\mathcal{Y},\, \mathcal{P}_{\mathcal{G}}^{(w)}\right) \;=\; \sigma_e^2\, dq_{\mathfrak{g}_e} \wedge O\left(\mathcal{Y}_{\mathfrak{g}_e},\, \mathcal{P}_{\mathcal{G}}^{(w)} \cap \widetilde{\mathcal{W}}^{(\mathfrak{g}_e)}\right)
\tag{4.27}
$$

where, as earlier, $\sigma_e := \text{sign}\{q_{\mathfrak{g}_e}\}$ – the second power of $\sigma_e$ comes from the sign function in (4.25). As far as the weight function of $\mathcal{P}_{\mathcal{G}}^{(w)} \cap \widetilde{\mathcal{W}}^{(\mathfrak{g}_e)}$ is concerned, it is given by

$$
w\left(\mathcal{Y},\, \mathcal{P}_{\mathcal{G}}^{(w)} \cap \widetilde{\mathcal{W}}^{(\mathfrak{g}_e)}\right) \;=\; \text{Disc}\left\{ w\left(\mathcal{Y},\, \mathcal{P}_{\mathcal{G}}^{(w)}\right) \right\} \;=\; \lim_{q_e \to 0^+} w\left(\mathcal{Y},\, \mathcal{P}_{\mathcal{G}}^{(w)}\right) - \lim_{q_e \to 0^-} w\left(\mathcal{Y},\, \mathcal{P}_{\mathcal{G}}^{(w)}\right) \;=\; \pm 2
\tag{4.28}
$$

where the $\pm$ depends on the sign functions associated to the other boundaries. Note that (4.28) strictly holds everywhere but on the eventual internal boundaries that $\mathcal{P}_{\mathcal{G}}^{(w)} \cap \mathcal{W}^{(\mathfrak{g})}$ might have, and it returns the overall weight of the canonical form associated to $\mathcal{P}_{\mathcal{G}}^{(w)} \cap \mathcal{W}^{(\mathfrak{g})}$.

Contrarily to the cosmological polytope case discussed in the previous paragraph, the change in the definition (4.25) maps the boundaries $\{\widetilde{\mathcal{W}}^{(\mathfrak{g}_e)},\, e \in \mathcal{E}\}$ from being spurious to be proper boundaries.

As both the positive and negative half-spaces they identify intersect the geometry, these boundaries are referred to as *internal boundaries*.

The canonical form defined for the cosmological polytope can be generalised to define a *weighted canonical form* attached to the weighted cosmological polytope by extending (4.19) to both ordinary [11] and internal boundaries:

$$\omega\left(w_\alpha, O_\alpha\right) = \operatorname{Res}_{\mathcal{W}^{(\alpha)}}\{\omega\left(w, O\right)\} \tag{4.29}$$

where $\mathcal{W}^{(\alpha)} = \mathcal{W}^{(\mathfrak{g})}$ for $\mathfrak{g} \subseteq \mathcal{G}$, identifying an ordinary boundary, or $\mathcal{W}^{(\alpha)} = \widetilde{\mathcal{W}}^{(\mathfrak{g}_e)}$ for $e \in \mathcal{E}$, identifying an internal boundary, $(w, O)$ and $(w_\alpha, O_\alpha)$ are short-hand notations for the pair weight function and orientation for, respectively, $\mathcal{P}_{\mathcal{G}}^{(w)}$ and the codimension-1 boundary $\mathcal{P}_{\mathcal{G}}^{(w)} \cap \mathcal{W}^{(\alpha)}$. In words, this is the statement that the residue of the canonical form along any (ordinary or internal) boundary identified by the hyperplane $\mathcal{W}^{(\alpha)}$, is still a weighted polytope which is identified by the pair $(w_\alpha, O_\alpha)$, *i.e.* respectively the discontinuity of the weight function of $\mathcal{P}_{\mathcal{G}}$ along $\mathcal{W}^{(\alpha)}$ and the induced orientation on $\mathcal{P}_{\mathcal{G}}^{(w)} \cap \mathcal{W}^{(\alpha)}$.

Importantly, the maximal residues can be computed in a similar fashion as in (4.20):

$$\begin{aligned}
\omega\left(w_{\alpha_1\ldots\alpha_{N-1}}, O_{\alpha_1\ldots\alpha_{N-1}}\right) &= \operatorname{Res}_{\mathcal{W}^{(\alpha_1)}}\ldots\operatorname{Res}_{\mathcal{W}^{(\alpha_{N-1})}}\omega\left(w, O\right) = \\
&= \operatorname{sign}\left\{O_{\alpha_1\ldots\alpha_{N-1}}\right\}\operatorname{Disc}_{\mathcal{W}^{(\alpha_1)}}\ldots\operatorname{Disc}_{\mathcal{W}^{(\alpha_{N-1})}}\{w\} = \pm 2^m
\end{aligned} \tag{4.30}$$

where, as before, $N := n_s + n_e$, while $m \in [0, n_e]$. The fact that the maximal residues can acquire the values $\pm 2^m$ is just a consequence of the fact that the facets of $\mathcal{P}_{\mathcal{G}}^{(w)}$ can have weights $\pm 1$ or $\pm 2$ and the weights of higher codimension faces will be given by suitable products of $\pm 1$ and $\pm 2$.

**Weighted cosmological polytopes and graphs** – One of the most useful features of cosmological polytopes is their $1-1$ correspondence with graphs. A similar association can also be obtained for the weighted cosmological polytopes.

Let us begin with considering the simplest case, which we have extensively discussed at the beginning of Section 4. The starting point was the cosmological polytope associated to the two-site line graph, a triangle in $\mathbb{P}^2$. Let $(s_e, s_e')$ label the two sites of the graph and $e$ the edge connecting them. In the language of the weight function and orientation, it is mapped into a weighed cosmological polytope by endowing it with the following weight functions and orientation

$$w\left(\mathcal{Y}, \mathcal{P}_{\mathcal{G}}^{(w)}\right) = \operatorname{sign}\left\{\mathcal{Y} \cdot \widetilde{\mathcal{W}}^{(\mathfrak{g}_e)}\right\}w\left(\mathcal{Y}, \mathcal{P}_{\mathcal{G}}\right), \qquad O\left(\mathcal{Y}, \mathcal{P}_{\mathcal{G}}^{(w)}\right) = O\left(\mathcal{Y}, \mathcal{P}_{\mathcal{G}}\right) \tag{4.31}$$

where $\widetilde{\mathcal{W}}^{(\mathfrak{g}_e)}$ is the co-vector identifying the line passing through the points $\mathbf{x}_{s_e}$ and $\mathbf{x}_{s_e'}$. With such a definition, it is straightforward to see that the segment $\mathcal{P}_{\mathcal{G}} \cap \widetilde{\mathcal{W}}^{(\mathfrak{g}_e)}$ is a codimension-1 internal boundary of $\mathcal{P}_{\mathcal{G}}^{(w)}$, while $\{\mathbf{x}_{s_e}, \mathbf{x}_{s_e'}\}$ are codimension-2 internal boundaries. Hence, we can associate to the two-site graph the weighted cosmological polytope $\mathcal{P}_{\mathcal{G}}^{(w)} \subset \mathbb{P}^2$, which is characterised by 5 vertices

$$\{\mathcal{Z}_1^{(e)}, \mathcal{Z}_2^{(e)}, \mathcal{Z}_3^{(e)}, \mathcal{Z}_4^{(e)}, \mathcal{Z}_5^{(e)}\} := \{\mathbf{x}_{s_e} - \mathbf{y}_e + \mathbf{x}_{s_e'}, \mathbf{x}_{s_e} + \mathbf{y}_e - \mathbf{x}_{s_e'}, -\mathbf{x}_{s_e} + \mathbf{y}_e + \mathbf{x}_{s_e'}, \mathbf{x}_{s_e'}, \mathbf{x}_{s_e}\} \tag{4.32}$$

---

[11] We refer as *ordinary* to the boundaries which intersect the geometry just in their positive half-space. In the case of the weighted cosmological polytope, they are the same as for the cosmological polytope and correspond to the set of hyperplanes $\{\mathcal{W}^{(\mathfrak{g})}, \mathfrak{g} \subseteq \mathcal{G}\}$.

Note that five vertices in $\mathbb{P}^2$ cannot be all linearly independent. In fact, they satisfy the following linear relations

$$2\mathcal{Z}_4^{(e)} \sim \mathcal{Z}_3^{(e)} + \mathcal{Z}_1^{(e)}, \qquad 2\mathcal{Z}_5^{(e)} \sim \mathcal{Z}_1^{(e)} + \mathcal{Z}_2^{(e)} \tag{4.33}$$

As we analysed earlier, this weighted triangle has 4 facets, three shared with original triangle and the internal one $\widetilde{\mathcal{W}}_I^{(\mathfrak{g}e)} = \epsilon_{IJK} \mathcal{Z}_4^{(e),\,J} \mathcal{Z}_5^{(e),\,K}$. While, with respect to the ordinary facets, all the vertices which are not on it lie on the positive half-space they identify, for the internal boundary $\mathcal{Z}_1^{(e)}$ lies in its negative half-plane, *i.e.* $\mathcal{Z}_1^{(e)} \cdot \mathcal{W}^{(\mathfrak{g}e)} < 0$.

Let us now consider a collection of $n_e$ 2-site graphs and their corresponding weighted triangles. All such weighted triangles can be embedded all together in $\mathbb{P}^{3n_e-1}$. Now, a generic connected graph $\mathcal{G}$ with $n_s$ sites and $n_e$ edges can be obtained from the collection of $n_e$ 2-site graphs by suitably identifying $r = 2n_e - n_s$ of their sites. From the geometrical side, this corresponds to intersect the $n_e$ weighted triangles in $r$ of their internal vertices $\{\mathcal{Z}_4^{(e)},\,\mathcal{Z}_5^{(e)}\}_{e\in\mathcal{E}}$.

The weighted cosmological polytope $\mathcal{P}_\mathcal{G}^{(w)}$ associated to a generic graph $\mathcal{G}$ has $3n_e$ ordinary vertices, $\{\mathcal{Z}_j^{(e)},\, j = 1, 2, 3\}_{e\in\mathcal{E}}$, and $2n_e - r = n_s$ internal vertices which now are again more conveniently to be labelled as $\{\mathbf{x}_s\}_{s\in\mathcal{V}}$ – depending on the convenience, we will keep using both notations $\{\mathcal{Z}_4^{(e)},\,\mathcal{Z}_5^{(e)}\}_{e\in\mathcal{E}}$ and $\{\mathbf{x}_{s_e'},\,\mathbf{x}_{s_e}\}_{e\in\mathcal{E}}$.

**Facet structure of the weighted cosmological polytopes** – As we briefly mentioned earlier, a facet is given by $\mathcal{P}_\mathcal{G}^{(w)} \cap \mathcal{W}^{(\alpha)} \neq \varnothing$, *i.e.* the non-vanishing intersection between the weighted cosmological polytope and the hyperplane identified by the co-vector $\mathcal{W}_I^{(\alpha)}$, and such that:

❑ if $\mathcal{W}^{(\alpha)} = \mathcal{W}^{(\mathfrak{g})}$, then $\mathcal{Z}_j^{(e)} \cdot \mathcal{W}^{(\mathfrak{g})} \geq 0$, $\forall j = 1, \ldots, 5$, $\mathfrak{g} \subseteq \mathcal{G}$ and $e \in \mathcal{E}$, with the (in)equality satisfied if the vertex is (not) on the facet;

❑ if $\mathcal{W}^{(\alpha)} = \widetilde{\mathcal{W}}^{(\mathfrak{g}e)}$, then $\mathcal{Z}_j^{(e)} \cdot \mathcal{W}^{(e)} \geq 0$, $\forall j = 2, \ldots, 5$ and $e \in \mathcal{E}$ as well as $\mathcal{Z}_1^{(e')} \cdot \widetilde{\mathcal{W}}^{(\mathfrak{g}e)} = 0$, $\forall e' \in \mathcal{E} \setminus \{e\}$ and $\mathcal{Z}_1^{(e)} \cdot \mathcal{W}^{(e)} < 0$, where again the (in)equalities are satisfied if the vertex is (not) on the facet[12].

Let us consider a generic hyperplane expressed in terms of the basis of co-vectors $\{\tilde{\mathbf{x}}_s,\, \tilde{\mathbf{y}}_\mathbf{e},\, s \in \mathcal{V},\, e \in \mathcal{E}\}$:

$$\mathcal{W} = \sum_{s\in\mathcal{V}} \tilde{x}_s \tilde{\mathbf{x}}_s + \sum_{e\in\mathcal{E}} \tilde{y}_e \tilde{\mathbf{y}}_s \tag{4.34}$$

with the set $\{\tilde{x}_s,\, \tilde{y}_e,\, s \in \mathcal{V},\, e \in \mathcal{E}\}$ containing arbitrary coefficients. The positivity conditions $\{\mathcal{Z}_j^{(e)} \cdot \mathcal{W} \geq 0,\, \forall j = 2, \ldots, 5,\, \forall e \in \mathcal{E}\}$ can be conveniently written as

$$\begin{aligned}
\alpha_{(e,\,s_e)} &:= \mathcal{Z}_2^{(e)} \cdot \mathcal{W} = \tilde{x}_{s_e} + \tilde{y}_e - \tilde{x}_{s_e'} \geq 0, \\
\alpha_{(e,\,s_e')} &:= \mathcal{Z}_3^{(e)} \cdot \mathcal{W} = -\tilde{x}_{s_e} + \tilde{y}_e + \tilde{x}_{s_e'} \geq 0, \\
\alpha_{(s_e',\,s_e')} &:= \mathcal{Z}_4^{(e)} \cdot \mathcal{W} = \tilde{x}_{s_e'} \geq 0, \\
\alpha_{(s_e,\,s_e)} &:= \mathcal{Z}_5^{(e)} \cdot \mathcal{W} = \tilde{x}_{s_e} \geq 0.
\end{aligned} \tag{4.35}$$

---

[12] Taking $\widetilde{\mathcal{W}}^{(\mathfrak{g}e)}$ such that $\mathcal{Z}_j^{(e)} \cdot \mathcal{W}^{(e)} \geq 0$, $\forall j = 2, \ldots, 5$ and $e \in \mathcal{E}$ as well as $\mathcal{Z}_1^{(e')} \cdot \mathcal{W}^{(e)} = 0$, $\forall e' \in \mathcal{E} \setminus \{e\}$ and $\mathcal{Z}_1^{(e)} \cdot \mathcal{W}^{(e)} < 0$, corresponds to a choice of orientation. Equivalently, we could change all the inequalities, and hence choose the opposite orientation.

Let us also denote

$$\alpha_{(e,e)} := \mathcal{Z}_1^{(e)} \cdot \mathcal{W} = \tilde{x}_{s_e} - \tilde{y}_e + \tilde{x}_{s'_e} \tag{4.36}$$

which now can be positive, zero or negative. The $\alpha$'s just introduced satisfy the same linear conditions as the vertices

$$\begin{aligned}
\alpha_{(e,e)} + \alpha_{(e,s_e)} &= \alpha_{(e',e')} + \alpha_{e',s_{e'}}, \\
2\alpha_{(s_e,s_e)} &= \alpha_{(e,e)} + \alpha_{(e,s_e)}, \\
2\alpha_{(s'_e,s'_e)} &= \alpha_{(e,e)} + \alpha_{(e,s'_e)}
\end{aligned} \tag{4.37}$$

In this terms, a vertex of $\mathcal{P}_{\mathcal{G}}^{(w)}$ is on a facet if the related $\alpha$ vanishes and $\mathcal{W}$ identifies the hyperplane containing a facet of $\mathcal{P}_{\mathcal{G}}^{(w)}$ if it corresponds to setting a maximal non-trivial subset of the $\alpha$'s to zero, without setting all of them, compatibly with the conditions (4.37). In order to keep track of such conditions on an arbitrary weighted cosmological polytope $\mathcal{P}_{\mathcal{G}}^{(w)}$, it is convenient to introduce a marking identifying all the *non-zero* $\alpha$'s:

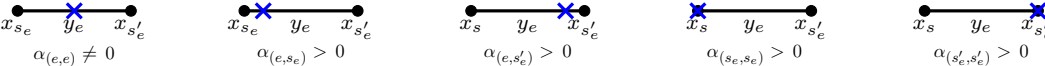

The positivity conditions (4.35) put constraints on the $\alpha$'s which can be simultaneously set to zero. This is very similar to what happens for the cosmological polytopes, with the novel feature of having just the subset $\{\alpha_{(e,e)}, e \in \mathcal{E}\}$ which can be either positive or negative rather than just positive. Concretely

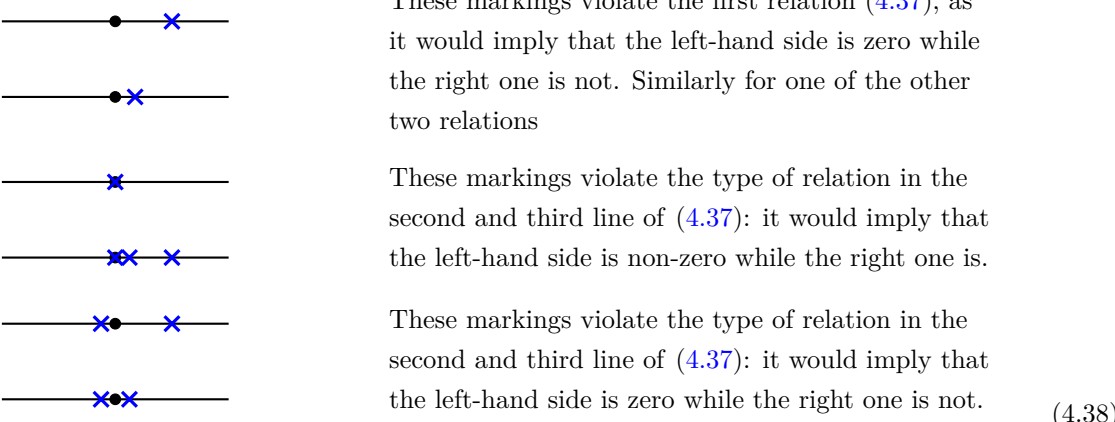

These markings violate the first relation (4.37), as it would imply that the left-hand side is zero while the right one is not. Similarly for one of the other two relations

These markings violate the type of relation in the second and third line of (4.37): it would imply that the left-hand side is non-zero while the right one is.

These markings violate the type of relation in the second and third line of (4.37): it would imply that the left-hand side is zero while the right one is not.

$$\tag{4.38}$$

Hence, given an arbitrary graph $\mathcal{G}$, the co-vector $\mathcal{W}$ that identifies one of the facets of $\mathcal{P}_{\mathcal{G}}^{(w)}$ contains as many linearly independent vertices as possible without containing the full polytope. In the language of the markings just introduced, $\mathcal{P}_{\mathcal{G}}^{(w)} \cap \mathcal{W} \neq \varnothing$ is a facet of $\mathcal{P}_{\mathcal{G}}^{(w)}$ if it does not contain any of the marking configurations in (4.38) and removing any of the markings force to either remove all the other ones as well or to land in a non-allowed configuration.

Such constraints can be translated into the following graphical rules. Given an arbitrary graph $\mathcal{G}$, we can associate the hyperplane $\mathcal{W}^{(\mathfrak{g})}$ to each subgraph $\mathfrak{g} \subseteq \mathcal{G}$, such that the vertex structure of the intersection $\mathcal{P}_{\mathcal{G}}^{(w)} \cap \mathcal{W}^{(\mathfrak{g})}$ is obtained by marking in the middle all the edges in $\mathfrak{g}$ as well as marking all

sites in $\mathfrak{g}$ and the edges departing from it close to their endpoints which are in $\mathfrak{g}$. If $\mathfrak{g}$ includes an edge only, there is a second hyperplane $\widetilde{\mathcal{W}}^{(\mathfrak{g})}$ associated to it and the vertex structure of $\mathcal{P}_{\mathcal{G}}^{(w)} \cap \widetilde{\mathcal{W}}^{(\mathfrak{g})}$ is instead obtained by marking completely all the edges included in $\mathfrak{g}$:

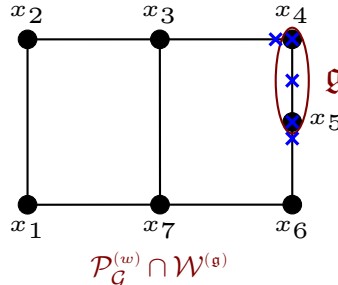 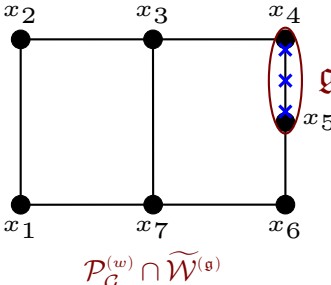

$$\mathcal{P}_{\mathcal{G}}^{(w)} \cap \mathcal{W}^{(\mathfrak{g})} \qquad\qquad \mathcal{P}_{\mathcal{G}}^{(w)} \cap \widetilde{\mathcal{W}}^{(\mathfrak{g})}$$

From now on, we will indicate the 2-site line subgraphs as $\mathfrak{g}_e$ with $e \in \mathcal{E}$ labelling the edge of $\mathcal{G}$ they include. Importantly, for the facets $\mathcal{P}_{\mathcal{G}}^{(w)} \cap \widetilde{\mathcal{W}}^{(\mathfrak{g}_e)}$, the vertex configuration shown in the right figure above corresponds to the constraints (4.37) to be reduced to

$$0 = \alpha_{(e',e')} + \alpha_{(e',s_{e'})}, \tag{4.39}$$

which can be satisfied if and only if $\alpha_{(e',e')}$ is negative given that $\alpha_{(e',s_{e'})}$ is always positive. Note that just the subgraphs

$$\{\mathfrak{g}_e \subset \mathcal{G} \,|\, \mathfrak{g}_e := \{\mathcal{V}_{\mathfrak{g}_e}, \mathcal{E}_{\mathfrak{g}_e}\}, \mathcal{V}_{\mathfrak{g}_e} := \{s_e, s_e'\}, \mathcal{E}_{\mathfrak{g}_e} := \{e\}\}_{e \in \mathcal{E}}, \tag{4.40}$$

*i.e.* those containing a single edge, are associated to a pair of facets identified by the hyperplanes $(\mathcal{W}^{(\mathfrak{g}_e)}, \widetilde{\mathcal{W}}^{(\mathfrak{g}_e)})$. The same marking configuration of the facets for this subset of subgraphs extended to all the other subgraphs, identifies hyperplanes such that their intersection with $\mathcal{P}_{\mathcal{G}}^{(w)}$ is empty in codimension-1. This is straightforward to show. Let $\mathfrak{g} = \mathcal{G}$, the intersection $\mathcal{P}_{\mathcal{G}}^{(w)} \cap \widetilde{\mathcal{W}}^{(\mathcal{G})}$ shows just $\{\mathbf{x}_s, s \in \mathcal{V}\}$ as vertices. However they are $n_s$ linearly independent vertices, while $\mathcal{P}_{\mathcal{G}}^{(w)} \cap \widetilde{\mathcal{W}}^{(\mathcal{G})}$ should live in $\mathbb{P}^{n_s+n_e-2}$. Hence, in order to be a facet, we must have $n_s = n_s + n_e - 1$, which is satisfied just for the 2-site line graph and, hence, for the weighted triangle. Let $\mathfrak{g} \subset \mathcal{G}$, then the intersection $\mathcal{P}_{\mathcal{G}}^{(w)} \cap \widetilde{\mathcal{W}}^{(\mathfrak{g})} \subset \mathbb{P}^{n_s+n_e-2}$ would have all the vertices of $\mathcal{P}_{\mathcal{G}}^{(w)}$ but the $3n_e^{(\mathfrak{g})}$ associated to the edges contained in $\mathfrak{g}$ – with $n_e^{(\mathfrak{g})}$ being the number of edges in $\mathfrak{g}$. Said differently, the marking for $\widetilde{\mathcal{W}}^{(\mathfrak{g})}$ corresponds to erasing the edges contained of $\mathfrak{g}$ but leaving its sites. It is useful to write $n_s^{(\mathfrak{g})} = n_s^{(\mathfrak{g}_{ext})} + n_s^{(\mathfrak{g}_{int})}$, where $n_s^{(\mathfrak{g}_{ext})}$ is the number of sites of $\mathfrak{g}$ which are endpoints of the edges departing from it, while $n_s^{(\mathfrak{g}_{int})}$ is the number of its sites which are not, and hence, are internal to it. Then, the intersection $\mathcal{P}_{\mathcal{G}}^{(w)} \cap \widetilde{\mathcal{W}}^{(\mathfrak{g})}$ factorises into a polytope associated to $\mathcal{G} \setminus \mathcal{E}_{\mathfrak{g}}$ that lives in $\mathbb{P}^{n_s-n_s^{(\mathfrak{g}_{int})}+n_e-n_e^{(\mathfrak{g})}-1}$ and the simplex $\Sigma_{\mathfrak{g}} \subset \mathbb{P}^{n_s^{(\mathfrak{g}_{int})}-1}$ formed by the $n_s^{(\mathfrak{g}_{int})}$ internal vertices of $\mathfrak{g}$. In order for $\mathcal{P}_{\mathcal{G}}^{(w)} \cap \widetilde{\mathcal{W}}^{(\mathfrak{g})}$ to be a facet, it should live in $\mathbb{P}^{n_s+n_e-2}$. Matching the dimensions of the two subspaces with the expected ones

$$n_s - n_s^{(\mathfrak{g}_{int})} + n_e - n_e^{(\mathfrak{g})} + n_s^{(\mathfrak{g}_{int})} - 1 = n_s + n_e - 1 \tag{4.41}$$

the only case in which the equality holds is for $n_e^{(\mathfrak{g})} = 1$.

Also note that if $\mathfrak{g}$ is made by just a site, there is no marking associated to $\mathcal{P}_\mathcal{G}^{(w)} \cap \widetilde{\mathcal{W}}^{(\mathfrak{g})}$. Hence, if $\tilde{\nu}$ is the number of subgraphs of $\mathcal{G}$ (and consequently the number of the facets of the cosmological polytope $\mathcal{P}_\mathcal{G}$), then the weighted cosmological polytope $\mathcal{P}_\mathcal{G}^{(w)}$ has $\tilde{\nu} + n_e$ facets.

This analysis allows to straightforwardly distinguish between ordinary and internal facets: the ordinary facets are identified by having all the non-zero $\alpha$'s positive in order to satisfy the constraints (4.37); the internal ones are instead identified by having one of the $\alpha_{(e,e)}$'s negative. They precisely correspond to the $n_e$'s co-vectors $\widetilde{\mathcal{W}}^{(\mathfrak{g}_e)}$ and their collection coincide with $\{\mathcal{W}^{(\mathfrak{g}_e)}, \, e \in \mathcal{E}\}$ described earlier. Finally, the co-vectors $\mathcal{W}^{(\mathfrak{g})}$ and $\widetilde{\mathcal{W}}^{(\mathfrak{g}_e)}$ can be written in terms of the basis $\{\tilde{\mathbf{x}}_s, \, \tilde{\mathbf{y}}_e, \, s \in \mathcal{V}, \, e \in \mathcal{E}\}$:

$$
\begin{aligned}
\mathcal{W}^{(\mathfrak{g})} &= \sum_{s \in \mathcal{V}} \tilde{\mathbf{x}}_s + \sum_{e \in \mathcal{E}_\mathfrak{g}^{\text{ext}}} \tilde{\mathbf{y}}_e, \qquad \forall \, \mathfrak{g} \subseteq \mathcal{G}, \\
\widetilde{\mathcal{W}}^{(\mathfrak{g}_e)} &= \tilde{\mathbf{y}}_e, \qquad\qquad\qquad\quad \forall \, e \in \mathcal{E}.
\end{aligned}
\tag{4.42}
$$

**Higher codimension faces and compatibility conditions** – Let us now consider higher codimension faces. A face of codimension-$k$ is defined as the weighted polytope $\mathcal{P}_\mathcal{G}^{(w)} \cap \mathcal{W}^{(\mathfrak{g}_1 \cdots \mathfrak{g}_k)} \neq \varnothing$ in $\mathbb{P}^{n_s + n_e - k - 1}$ identified by the codimension-$k$ hyperplane $\mathcal{W}^{(\mathfrak{g}_1 \cdots \mathfrak{g}_k)} := \bigcap_{j=1}^k \widehat{\mathcal{W}}^{(\mathfrak{g}_j)}$, with each $\widehat{\mathcal{W}}^{(\mathfrak{g}_j)}$ ($j = 1, \ldots, k$) being either $\mathcal{W}^{(\mathfrak{g}_j)}$ or $\widetilde{\mathcal{W}}^{(\mathfrak{g}_k)}$.

The facets such that $\mathcal{P}_\mathcal{G}^{(w)} \cap \mathcal{W}^{(\mathfrak{g}_1 \cdots \mathfrak{g}_k)} \neq \varnothing$ in codimension-$k$ determine the codimension-$k$ faces of $\mathcal{P}_\mathcal{G}^{(w)}$ and are said to be *compatible*. If instead $\mathcal{P}_\mathcal{G}^{(w)} \cap \mathcal{W}^{(\mathfrak{g}_1 \cdots \mathfrak{g}_k)} = \varnothing$ they are said to be *incompatible* and determine a codimension-$k$ subspace of the adjoint surface of $\mathcal{P}_\mathcal{G}^{(w)}$. As the hyperplanes $\mathcal{W}^{(\mathfrak{g})}$'s are the same as for the cosmological polytope, they satisfy their same compatibility conditions (3.7): despite the boundaries they determine can be still weighted polytopes rather than coinciding with the related face of the cosmological polytope, the dimensionality of the space they live in is the same and the additional internal vertices are a linear combination of the other vertices. Hence, we need to determine just the compatibility conditions among $\mathcal{W}^{(\mathfrak{g}_j)}$'s and $\widetilde{\mathcal{W}}^{(\mathfrak{g}_j)}$'s, as well as among the $\widetilde{\mathcal{W}}^{(\mathfrak{g}_j)}$ themselves.

Let us begin with considering $\mathcal{W}^{(\mathfrak{g}_1 \cdots \mathfrak{g}_k)} = \bigcap_{j=1}^k \widetilde{\mathcal{W}}^{(\mathfrak{g}_j)}$ with $\mathfrak{g}_j = \mathfrak{g}_{e_j}$. The graph $\mathcal{G}$ gets factorised into $k$ graphs for purely loop graphs, or at most $(k+1)$ graphs if they contain a tree-level (sub)structure, which are obtained by erasing $k$ edges

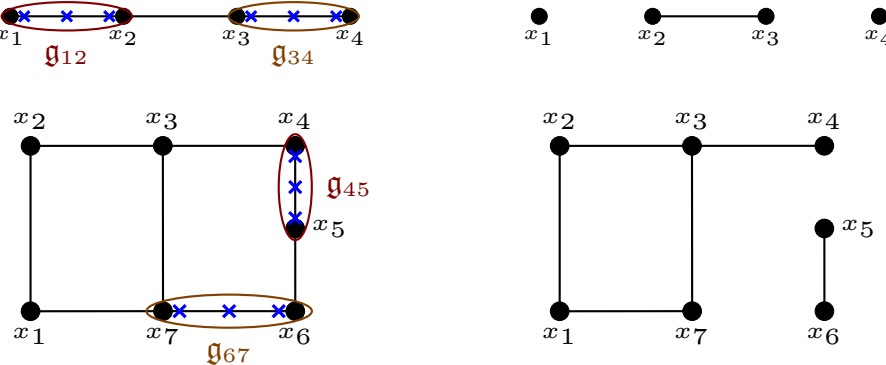

As usual, in order for this factorised structure to correspond to a codimension-$k$ face, it should live in $\mathbb{P}^{n_s + n_e - k - 1}$. The subspace corresponding to each factorised graph lives in $\mathbb{P}^{n_s^{(\mathfrak{g}_j)} + n_e^{(\mathfrak{g}_j)} - 1}$. The

dimension where $\mathcal{P}_{\mathcal{G}}^{(w)} \cap \mathcal{W}^{(\mathfrak{g}_1 \cdots \mathfrak{g}_k)}$ is

$$
\begin{aligned}
\dim \left\{ \mathcal{P}_{\mathcal{G}}^{(w)} \cap \mathcal{W}^{(\mathfrak{g}_1 \cdots \mathfrak{g}_k)} \right\} \;&=\; \sum_{\{\mathcal{P}\}_\mathfrak{g}} \left[ n_s^{(\mathfrak{g}_j)} + n_e^{(\mathfrak{g}_j)} \right] - 1 \;= \\
&=\; n_s + n_e - k - 1
\end{aligned}
\tag{4.43}
$$

where the sum runs over the set of polytopes $\{\mathcal{P}_\mathfrak{g}\}$ associated to the factorised graphs. Importantly, the factorised graphs all together involve all the sites of $\mathcal{G}$ and all its edges but $k$ of them, returning the second line in (4.43), which is precisely the correct dimension for a codimension-$k$ face. Thus, all the intersections of the type $\mathcal{P}_{\mathcal{G}}^{(w)} \cap \mathcal{W}^{(\mathfrak{g}_1 \cdots \mathfrak{g}_k)}$ are codimension-$k$ faces of $\mathcal{P}_{\mathcal{G}}^{(w)}$.

Let us now consider $\mathcal{W}^{(\mathfrak{g}_1 \cdots \mathfrak{g}_k)}$ such that it is given by the intersection of both type of hyperplane $\mathcal{W}^{(\mathfrak{g}_j)}$ and $\widetilde{\mathcal{W}}^{(\mathfrak{g}_m)}$. Let us begin with considering faces of codimension $k = 2$. There are three possible configurations for $\mathcal{W}^{(\mathfrak{g}_1 \mathfrak{g}_2)} := \mathcal{W}^{(\mathfrak{g}_1)} \cap \widetilde{\mathcal{W}}^{(\mathfrak{g}_2)}$ we ought to consider: $\mathfrak{g}_2 \subseteq \mathfrak{g}_1$, $\mathfrak{g}_1 \cap \mathfrak{g}_2 \neq \varnothing$ with $\mathfrak{g}_2 \nsubseteq \mathfrak{g}_1$, and $\mathfrak{g}_1 \cap \mathfrak{g}_2 = \varnothing$.

❑ $\mathfrak{g}_2 \subseteq \mathfrak{g}_1$ – The vertex configuration for $\mathcal{P}_{\mathcal{G}}^{(w)} \cap \mathcal{W}^{(\mathfrak{g}_1)} \cap \widetilde{\mathcal{W}}^{(\mathfrak{g}_2)}$ is given by the factorised structure between the scattering facet $\mathcal{S}_{\mathfrak{g}_1 \setminus \mathcal{E}_{\mathfrak{g}_2}}$, corresponding to the graph $\mathfrak{g}_1 \setminus \mathcal{E}_{\mathfrak{g}_2}$ obtained by erasing the edge of $\mathfrak{g}_2$ in $\mathfrak{g}_1$, and the polytope $\mathcal{P}_{\bar{\mathfrak{g}}_1 \cup \mathcal{E}}$ [13], $\mathcal{E}$ being the set of edges connecting $\mathfrak{g}_1$ and $\mathfrak{g}_2$

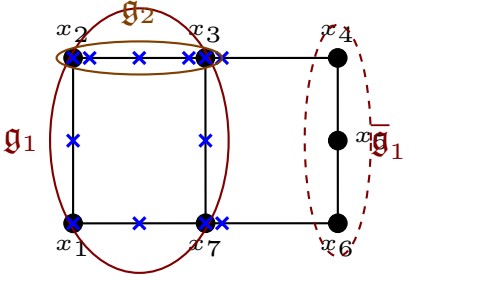 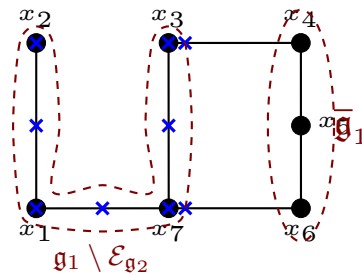

The dimension of this codimension-2 faces is therefore

$$
\dim \left\{ \mathcal{P}_{\mathcal{G}}^{(w)} \cap \mathcal{W}^{(\mathfrak{g}_1)} \cap \widetilde{\mathcal{W}}^{(\mathfrak{g}_2)} \right\} \;=\; n_s^{(\mathfrak{g}_{12})} + n_e^{(\mathfrak{g}_{12})} - 1 + n_{\mathcal{E}} + n_s^{(\bar{\mathfrak{g}}_1)} + n_e^{(\bar{\mathfrak{g}}_1)} - 1
\tag{4.44}
$$

where $\mathfrak{g}_{12} := \mathfrak{g}_1 \setminus \mathcal{E}_{\mathfrak{g}_2}$, the first three terms provide the affine dimension of $\mathcal{S}_{\mathfrak{g}_1 \setminus \varepsilon_{\mathfrak{g}_2}}$, while the following three the affine dimension $\mathcal{P}_{\bar{\mathfrak{g}} \cup \mathcal{E}}$, and the last $-1$ is a consequence of projectivity. Note that $n_s^{(\mathfrak{g}_{12})} + n_s^{(\bar{\mathfrak{g}}_1)} = n_s$ and $n_e^{(\mathfrak{g}_{12})} + n_{\mathcal{E}} + n_e^{(\bar{\mathfrak{g}}_1)} = n_e - 1$. Hence

$$
\dim \left\{ \mathcal{P}_{\mathcal{G}}^{(w)} \cap \mathcal{W}^{(\mathfrak{g}_1)} \cap \widetilde{\mathcal{W}}^{(\mathfrak{g}_2)} \right\} \;=\; n_s + n_e - 2 - 1
\tag{4.45}
$$

which is the expected dimension for a codimension-2 face. However, if the $\mathfrak{g}_1$ contains a tree

---

[13]Note that $\bar{\mathfrak{g}}_1 \cup \mathcal{E}$ constitutes an abuse of notation as $\bar{\mathfrak{g}}_1 \cup \mathcal{E}$ is not strictly a graph as it does not contain all the sites which are endpoints of $\mathcal{E}$.

substructure, and $\mathfrak{g}_2 \subset \mathfrak{g}_1$ is in it, *e.g.*

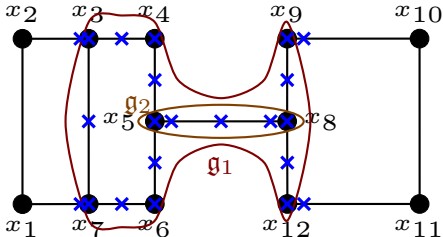

then, the subgraph $\mathfrak{g}_1 \setminus \mathfrak{g}_2$ is actually a disconnected graph, made out by the two subgraphs $\mathfrak{g}_1^{(L)}$ and $\mathfrak{g}_1^{(R)}$ at the left and right of the egde of $\mathfrak{g}_2$. The scattering facet $\mathcal{S}_{\mathfrak{g}_1 \setminus \mathfrak{g}_2}$ then factorises in two lower dimensional scattering facets, $\mathcal{S}_{\mathfrak{g}_1^{(L)}}$ and $\mathcal{S}_{\mathfrak{g}_1^{(R)}}$, associated to $\mathfrak{g}_1^{(L)}$ and $\mathfrak{g}_1^{(R)}$ respectively:

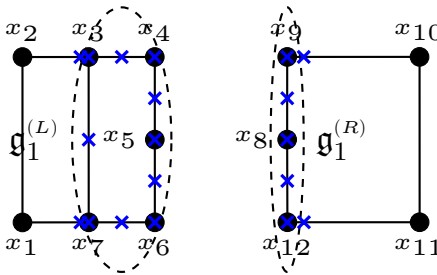

In this case, each disconnected part factorises into a lower-dimensional scattering facet and a polytope associated to $\overline{\mathfrak{g}}_1^{(L/R)} \cup \mathcal{L}^{(L/R)}$. Consequently, the dimension of the intersection $\mathcal{P}_{\mathcal{G}}^{(w)} \cap \mathcal{W}^{(\mathfrak{g}_1)} \cap \widetilde{\mathcal{W}}^{(\mathfrak{g}_2)}$ is given by

$$
\begin{aligned}
\dim\left\{ \mathcal{P}_{\mathcal{G}}^{(w)} \cap \mathcal{W}^{(\mathfrak{g}_1)} \cap \widetilde{\mathcal{W}}^{(\mathfrak{g}_2)} \right\} &= n_s^{(L)} + n_e^{(L)} - 1 + n_s^{(\overline{L})} + n_e^{(\overline{L})} + n_{\mathscr{L}}^{(L)} + \\
&\quad + n_s^{(R)} + n_e^{(R)} - 1 + n_s^{(\overline{R})} + n_e^{(\overline{R})} + n_{\mathscr{L}}^{(R)} - 1
\end{aligned}
\tag{4.46}
$$

where we used $n_{s/e}^{(L/R)}$ as a short-hand notation for the number of sites/edges of $\mathfrak{g}_1^{(L/R)}$ as well as $n_{s/e}^{(\overline{L}/\overline{R})}$ for the number of sites/edges of $\overline{\mathfrak{g}}_1^{(L/R)}$, while $n_{\mathscr{L}}^{(L/R)}$ is the number of egdes between $\mathfrak{g}_1^{(L/R)}$ and $\overline{\mathfrak{g}}_1^{(L/R)}$. Note that

$$
n_s^{(L)} + n_s^{(\overline{L})} + n_s^{(R)} + n_s^{(\overline{R})} = n_s, \qquad n_e^{(L)} + n_e^{(\overline{L})} + n_e^{(R)} + n_e^{(\overline{R})} + n_{\mathscr{L}}^{(L)} + n_{\mathscr{L}}^{(R)} = n_e - 1 \tag{4.47}
$$

and, consequently,

$$
\dim\left\{ \mathcal{P}_{\mathcal{G}}^{(w)} \cap \mathcal{W}^{(\mathfrak{g}_1)} \cap \widetilde{\mathcal{W}}^{(\mathfrak{g}_2)} \right\} = n_s + n_e - 3 - 1. \tag{4.48}
$$

This intersection is therefore empty in codimension 2.

❑ $\mathfrak{g}_1 \cap \mathfrak{g}_2 \neq \varnothing$ & $\mathfrak{g}_2 \nsubseteq \mathfrak{g}_1$ – This configuration occurs when $\mathfrak{g}_1$ and $\mathfrak{g}_2$ intersect each other in a

single site

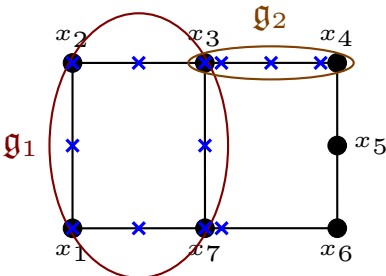 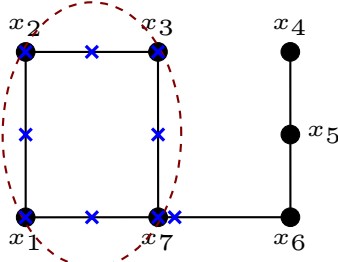

Such an intersection is non-empty if $\mathcal{W}^{(\mathfrak{g}_1)}$ contains a facet of the weighted cosmological polytope associated to the graph $\mathcal{G} \setminus \mathcal{E}_{\mathfrak{g}_2}$, which is always the case.

❏ $\mathfrak{g}_1 \cap \mathfrak{g}_2 = \varnothing$ – This configuration corresponds to have $\mathfrak{g}_1$ and $\mathfrak{g}_2$ non-overlapping

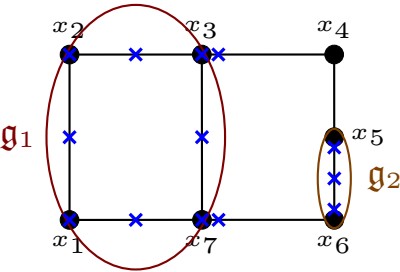 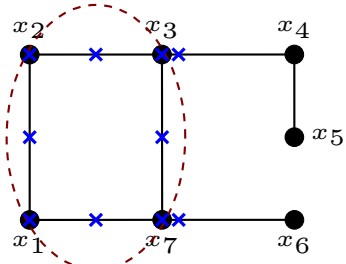

As in the previous case, such an intersection is non-empty in codimension-2 if $\mathcal{W}^{\mathfrak{g}_1}$ contains a facet of the weighted cosmological polytope associated to the graph $\mathcal{G} \setminus \mathcal{E}_{\mathfrak{g}_2}$, which, again, is always the case.

It is interesting now to analyse all together the dimension counting for the intersection of two hyperplanes containing two facets of $\mathcal{P}_{\mathcal{G}}^{(w)}$ and $\mathcal{P}_{\mathcal{G}}^{(w)}$ itself, focusing in particular to the $-1$'s appearing. Besides the one due to projectivity, there is a set of $-1$'s due to the number of lower-dimensional scattering facets appearing in this type of intersection, and a further one related to the fact there is one edge – the one of $\mathfrak{g}_2$ – with no vertices associate to it. Hence, we can write the counting for all the three classes of cases above all together as

$$\dim \left\{ \mathcal{P}_{\mathcal{G}}^{(w)} \cap \mathcal{W}^{(\mathfrak{g}_1)} \cap \widetilde{\mathcal{W}}^{(\mathfrak{g}_2)} \right\} = n_s + n_e - \sum_{\{\mathcal{S}_{\mathfrak{g}}\}} 1 - \not{n} - 1. \tag{4.49}$$

where the sum runs over the set of lower-dimensional scattering facets $\{\mathcal{S}_{\mathfrak{g}}\}$. Finally, from equation (4.49) it is easy to write the compatibility condition for codimension-2 intersections as

$$\not{n} + \sum_{\{\mathcal{S}_{\mathfrak{g}}\}} 1 = 2 \tag{4.50}$$

which is the very same form for the compatibility condition (3.7) for the cosmological polytope, with $k = 2$.

For intersections of higher codimension, the analysis generalises straightforwardly: the case where all the hyperplanes contain just facets of the cosmological polytope has been already analysed in

[29] and precisely led to the compatibility condition (3.7), while for all the other cases extending the discussion just carried out does not present any subtlety. It is thus possible to write a single compatibility condition irrespectively that the hyperplanes under consideration contain ordinary or internal boundaries

$$\not{n} + \sum_{\{\mathcal{S}_\mathfrak{g}\}} 1 \; = \; k, \tag{4.51}$$

$k$ being, as usual, the codimension.

Finally, let us consider the cases in which the compatibility condition (4.51) is satisfied and $\not{n} \neq 0$. Recall that $\not{n}$ counts the number of edges of the graph whose associated triple of vertices is not on the codimension-$k$ face $\mathcal{P}_\mathcal{G}^{(w)} \cap \mathcal{W}^{(\mathfrak{g}_1 \cdots \mathfrak{g}_k)}$. Interestingly, the intersection with the hyperplanes $\widetilde{\mathcal{W}}^{(\mathfrak{g}_e)}$ associated to any of the edges counted by $\not{n}$ is still in codimension-$k$ as each of them (and hence their intersection) intersects $\mathcal{P}_\mathcal{G}^{(w)} \cap \mathcal{W}^{(\mathfrak{g}_1 \cdots \mathfrak{g}_k)}$ in all its vertices: on the intersection each of the hyperplanes the hyperplanes $\widetilde{\mathcal{W}}^{(\mathfrak{g}_e)}$ eliminate all the vertices associated to the relevant edge $e$, but they are already not on $\mathcal{P}_\mathcal{G}^{(w)} \cap \mathcal{W}^{(\mathfrak{g}_1 \cdots \mathfrak{g}_k)}$.

In the cosmological polytope case, the knowledge of the compatible and incompatible facets, expressed in terms of the compatibility condition (3.7), allows not only to fix its full face structure but also to uniquely determine its adjoint surface as well as provide a way to determine all the signed triangulations which do not use hyperplanes different than the ones containing the facets and, consequently, for which the canonical forms of each simplex does not show any spurious pole [29]. As we will see later on, in the case of the weighted cosmological polytope the locus of the intersections of the hyperplanes containing the facets outside of the polytope, does not determine uniquely its adjoint surface. We already saw this phenomenon in the case of the weighted triangle in Section 4: by projective invariance, the adjoint surface was expected to be a line, but there was just a point determined as an outer intersection of the facets of the weighted triangle. We supplemented this condition by requiring that the outer intersection between any other facet and special codimension-1 hyperplanes was also on the adjoint, and this allowed us to univocally determine the adjoint of the weighted triangle. These new conditions can be extended to arbitrary weighted cosmological polytopes. However, as we will see, it turns out that, in general, the original conditions together with these additional ones, do not determine the adjoint surface uniquely. Thinking about the weighted cosmological polytopes as a weighted sum of regular polytopes, this is the same phenomenon which has been observed in the context polypols, a generalisation of polytopes with non-linear hypersurfaces as boundaries [39, 40]: they have a unique canonical form but no unique adjoint surface [32].

## 4.2 Weighted cosmological polytopes and cosmological correlators

Our discussion so far has been focusing on the mathematical structure of the newly-defined weighted cosmological polytope, and in particular on the structure of their boundaries for arbitrary codimensions. Here we will turn to their connection with the physics of the cosmological correlators. As already mentioned, the direct link between weighted cosmological polytopes on one side, and cosmological correlators on the other is given by the identification of the canonical function [14] of the former

---

[14] Recall that the canonical function of a polytope is obtained from its canonical form by stripping out the standard measure in projective space $\langle \mathcal{Y} d^{n_s+n_e-1} \mathcal{Y} \rangle$ – see (3.1).

with the latter

$$\Omega(\mathcal{Y}, \mathcal{P}_{\mathcal{G}}^{(w)}) = \mathcal{C}_{\mathcal{G}}(x, y) \tag{4.52}$$

in the local patch $\mathcal{Y} := (\{x_s\}_{s \in \mathcal{V}}, \{y_e\}_{e \in \mathcal{E}})$. In particular, we will show how different polytope sub-divisions give rise to different representations for the correlator universal integrand, one of which corresponds to the diagrammatic discussed in Section 2.5, as well as the implication of the boundary structure of $\mathcal{P}_{\mathcal{G}}^{(w)}$ on the analytic structure of $\mathcal{C}_{\mathcal{G}}$.

**A contour integral representation for cosmological correlators** – In the previous section we showed how the weighted cosmological polytopes can be associated to graphs. In particular, a given graph $\mathcal{G}$ with $n_s$ sites and $n_e$ edges can be obtained from a collection of 2-site graphs by suitably merging their sites via $r = 2n_e - n_s$ constraints. The associated weighted cosmological polytope $\mathcal{P}_{\mathcal{G}}^{(w)}$ can therefore be generared via intersecting $n_e$ weighted triangles into their internal vertices $\{\mathbf{x}_s\}$.

If on one side such geometrical definition provides directly the full set of (both ordinary and internal) vertices characterising $\mathcal{P}_{\mathcal{G}}$ as well as the correspondence with the graph $\mathcal{G}$, on the other it can also be implemented directly at the level of the canonical functions and, consequently, of the cosmological correlators.

In the discussion in Section 2.6 about the relation between graphs at higher order in perturbation theory and lower order graphs, it was shown that a connected graph of arbitrary topology $\mathcal{G}$ can generate a graph with one more loop by merging two of its sites. Such an operation is implemented via a contour integral over the cosmological correlator associated to $\mathcal{G}$ – see formula (2.36).

The operation of merging two sites is independent of the topology of the graph. In Section 2.6 we were interested in relating the correlators related to graphs at different loop orders and, consequently, the discussion was focused on connected graphs. However, the proof of which poles are physical and which ones become spurious do not rely on this assumption, neither depends on it the rest of the proof of (2.36).

Let us consider a collection of $n_e$ 2-site graphs $\{\mathfrak{g}_e, \, e \in \mathcal{E}\}$, where the index $e$ runs over the set of edges $\mathcal{E}$ of such collection of graphs. Let $\{x_{s_e}, \, x_{s'_e}\}$ be the set of site-weights for a given graph $\mathfrak{g}_e$ in this collection. Finally, let $\mathcal{C}_{\mathfrak{g}_e}(x_{s_e}, x_{s'_e})$ be the cosmological correlator associated to $\mathfrak{g}_e$. We can consider a set of $r = 2n_e - n_s$ one-parameter deformations on the site-weights, one for each pair of sites to be merged:

$$x_{s_e}(z) := x_{s_e} + z_{ss'}, \qquad x_{s'_e}(z) := x_{s'_e} - z_{ss'} \tag{4.53}$$

Then, the cosmological correlator associated to $\mathcal{G}$ is given by the following contour integral:

$$\mathcal{C}_{\mathcal{G}} = \int\limits_{-i\infty}^{+i\infty} \prod_{\{ss'\}} \left[ \frac{dz_{ss'}}{2\pi i} \right] \prod_{e \in \mathcal{E}} \mathcal{C}_{\mathfrak{g}_e}(z_{ss'}) \tag{4.54}$$

*Example* – It is instructive to explicitly discuss a simple example. Let us consider a collection of two

2-site graphs and their associated canonical functions

$$\mathcal{C}_{\mathfrak{g}}(x,y) \equiv \Omega_{\mathfrak{g}}(x,y) = \frac{2(x_1 + x_2 + y)}{(x_1 + x_2)(x_1 + y)(y + x_2)y}$$

$$\mathcal{C}_{\mathfrak{g}'}(x',y') \equiv \Omega_{\mathfrak{g}'}(x',y') = \frac{2(x_1' + x_2' + y')}{(x_1' + x_2')(x_1' + y')(y' + x_2')y'} \tag{4.55}$$

Let us consider the following 1-parameter deformation $x_2(z) := x_2 + z_2$, $x_2'(z) := x_2' - z_2$ in order to merge the sites with weights $x_2$ and $x_2'$ and obtain a 3-site line graph. Then, the associated cosmological correlator $\mathcal{C}_{\mathcal{G}}$ is given by

$$\mathcal{C}_{\mathcal{G}} = \int_{-i\infty}^{+i\infty} \frac{dz}{2\pi i} \, \mathcal{C}_{\mathfrak{g}}(x,y;\, z_2) \, \mathcal{C}_{\mathfrak{g}'}(x',y';\, z_2) \tag{4.56}$$

The contour integral can be closed into the positive or negative half-plane. In the first case:

$$\mathcal{C}_{\mathcal{G}} = \oint_{\gamma_+} \frac{dz}{2\pi i} \, \mathcal{C}_{\mathfrak{g}}(x,y;\, z_2) \, \mathcal{C}_{\mathfrak{g}'}(x',y';\, z_2) =$$

$$= \operatorname{Res}_{E_{\mathfrak{g}}(z_2)} \left\{ \mathcal{C}_{\mathfrak{g}}(x,y;\, z_2) \, \mathcal{C}_{\mathfrak{g}'}(x',y';\, z_2) \right\} + \operatorname{Res}_{E_{\mathfrak{g}_2}(z_2)} \left\{ \mathcal{C}_{\mathfrak{g}}(x,y;\, z) \, \mathcal{C}_{\mathfrak{g}'}(x',y';\, z) \right\} =$$

$$= \frac{4 \left[ x_1 + x_1' + (x_2 + x_2' + y') \right]}{(y - x_1)(x_1 + y) \left[ x_1 + (x_2 + x_2') + x_1' \right] \left[ x_1 + (x_2 + x_2') + y' \right] (x_1' + y')y'} + \tag{4.57}$$

$$+ \frac{4x_1 \left[ y + (x_2 + x_2') + y' + x_1' \right]}{(x_1 - y)(x_1 + y)y \left[ y + (x_2 + x_2') + x_1' \right] (y' + x_1') \left[ y + (x_2 + x_2') + y' \right] y'} =$$

$$= \frac{\mathfrak{n}_3(x_1, y, X_2, y', x_1')}{yy'(x_1 + X_2 + x_1')(x_1 + y)(y + X_2 + y')(y' + x_1')(x_1 + X_2 + y')(y + X_2 + x_1')}$$

where $X_2 := x_2 + x_2'$, $\mathfrak{g}_2$ is the subgraph constaing only the site $x_2$ and the numerator $\mathfrak{n}_3(x_1, y, X_2, y', x_1')$ has the following explicit form

$$\begin{aligned}
\mathfrak{n}_3 :=& 4 \left[ y(y')^2 + x_1'(y')^2 + x_1(y')^2 + X_2(y')^2 + y^2 y' + 2x_1' yy' + 2x_1 yy' + 3X_2 yy' + (x_1')^2 y' + \right. \\
& + 2x_1 x_1' y' + 3X_2 x_1' y' + x_1^2 y' + 3X_2 x_1 y' + 2X_2^2 y' + x_1' y^2 + x_1 y^2 + X_2 y^2 + (x_1')^2 y + \\
& + 2x_1 x_1' y + 3X_2 x_1' y + x_1^2 y + 3X_2 x_1 y + 2X_2^2 y + x_1(x_1')^2 + X_2(x_1')^2 + x_1^2 x_1' + 3X_2 x_1 x_1' + \\
& \left. + 2X_2^2 x_1' + X_2 x_1^2 + 2X_2^2 x_1 + X_2^3 \right]
\end{aligned}$$

We can obtain the same result via a different representation by closing the integration contour in the negative half-plane.

It is now possible to apply a second deformation and contour integral to obtain, via the tree theorem in Section (2.6) the correlator associated to either the of two one-loop topologies obtainable by merging two vertices of the three-site chain: the 2-site one-loop graph is obtained via the deformation $x_1(z_1) := x_1 + z_1$, $x_1'(z_1) := x_1' - z_1$, while the 2-site tadpole graph via $X_2(z_3) := X_2 + z_3$, $x_1'(z_3) := x_1' - z_3$. Hence, the cosmological correlators associated to such one-loop graphs can be expressed as a double

contour integral of the original building blocks:

$$
\begin{aligned}
x_1 \,\,\overset{y}{\underset{y'}{\bigcirc}}\,\, x_2 \;&=\; \int_{-i\infty}^{+i\infty} \prod_{j=1}^{2} \left[\frac{dz_j}{2\pi i}\right] \mathcal{C}_{\mathfrak{g}}(x_1(z_1), x_2(z_2), y)\, \mathcal{C}_{\mathfrak{g}'}(x_1'(z_1), x_2'(z_2), y') \\[2mm]
\overset{y}{\underset{x_1 \quad x_2}{\bullet\!-\!\!\!\bigcirc}}\,\, y' \;&=\; \int_{-i\infty}^{+i\infty} \prod_{j=2}^{3} \left[\frac{dz_j}{2\pi i}\right] \mathcal{C}_{\mathfrak{g}}\left(x_1, x_2(z_2, z_3), y\right)\, \mathcal{C}_{\mathfrak{g}'}(x_1'(z_3), x_2'(x_2), y')
\end{aligned}
\tag{4.58}
$$

As a final comment, such contour representation provides a direct way to obtain canonical form subdivisions [15] for the weighted cosmological polytopes. The number of possible subdivisions is $2^{2n_e - n_s}$ ($n_e$ and $n_s$ being the number of edges and sites of the final graph) and it is given by all the possible combinations of the choices of integration contours. In the next paragraph, we will discuss in detail polytope subdivisions for the weighted cosmological polytopes and representations for the cosmological correlators.

**From polytope subdivisions to representations for $\mathcal{C}_{\mathcal{G}}$** — Given a graph $\mathcal{G}$ and the associated cosmological polytope $\mathcal{P}_{\mathcal{G}}$ and weighted cosmological polytope $\mathcal{P}_{\mathcal{G}}^{(w)}$, let us consider the collection of hyperplanes $\{\widetilde{\mathcal{W}}^{(\mathfrak{g}_e)}, e \in \mathcal{E}\}$ used in the map from $\mathcal{P}_{\mathcal{G}}$ to $\mathcal{P}_{\mathcal{G}}^{(w)}$, which are such that $\bigcap_{e' \in \mathcal{E}\setminus\{e\}} \mathcal{W}^{(\mathfrak{g}_{e'})}$ is the 2-plane containing the triangle $\{\mathbf{x}_{s_e} - \mathbf{y}_e + \mathbf{x}'_{s_e},\, \mathbf{x}_{s_e} + \mathbf{y}_e - \mathbf{x}'_{s_e},\, -\mathbf{x}_{s_e} + \mathbf{y}_e + \mathbf{x}'_{s_e}\}$, for all $e \in \mathcal{E}$. It provides a polytope subdivision for $\mathcal{P}_{\mathcal{G}}$ whose elements $\mathcal{P}_{\mathcal{G}_j}$ are defined by the following pairs

$$
\begin{aligned}
w\left(\mathcal{Y}, \mathcal{P}_{\mathcal{G}_j}\right) \;&=\; w\left(\mathcal{Y}, \mathcal{P}_{\mathcal{G}}\right) \prod_{e \in \mathcal{E}_j} \vartheta\left(\mathcal{Y} \cdot \widetilde{\mathcal{W}}^{(\mathfrak{g}_e)}\right) \prod_{e' \notin \mathcal{E}_j} \vartheta\left(-\mathcal{Y} \cdot \widetilde{\mathcal{W}}^{(\mathfrak{g}_{e'})}\right), \\[2mm]
O\left(\mathcal{Y}, \mathcal{P}_{\mathcal{G}_j}\right) \;&=\; O\left(\mathcal{Y}, \mathcal{P}_{\mathcal{G}}\right)
\end{aligned}
\tag{4.59}
$$

where $\mathcal{E}_j \subseteq \mathcal{E}$ is a subset of edges containing $j \in [0, n_e]$ edges, with $\mathcal{E}_{n_e} = \mathcal{E}$, and $\{\mathcal{E}_j\}$ is the collection of all the possible, inequivalent, such subsets. It is straightforward to see that

$$
\sum_{j=0}^{n_e} \sum_{\{\mathcal{G}_j\}} w\left(\mathcal{Y}, \mathcal{P}_{\mathcal{G}_j}\right) \;=\; w\left(\mathcal{Y}, \mathcal{P}_{\mathcal{G}}\right) \sum_{j=0}^{n_e} \sum_{\{\mathcal{E}_j\}} \prod_{e \in \mathcal{E}_j} \vartheta\left(\mathcal{Y} \cdot \widetilde{\mathcal{W}}^{(\mathfrak{g}_e)}\right) \prod_{e' \notin \mathcal{E}_j} \vartheta\left(-\mathcal{Y} \cdot \widetilde{\mathcal{W}}^{(\mathfrak{g}_{e'})}\right) \;=\; w\left(\mathcal{Y}, \mathcal{P}_{\mathcal{G}}\right)
\tag{4.60}
$$

with $\{\mathcal{G}_j\}$ being of the collection of labels $\mathcal{G}_j$ which keeps track of the subset of edges $\mathcal{E}_j$. Hence, the collection $\{\mathcal{P}_{\mathcal{G}_j}, \{\mathcal{G}_j\}, j = 0, \ldots, n_E\}$ endowed with the weight function and the orientation in (4.59) provides a polytope subdivision for the cosmological polytope $\mathcal{P}_{\mathcal{G}}$. In terms of the canonical function, it yields

$$
\Omega\left(\mathcal{Y}, \mathcal{P}_{\mathcal{G}}\right) \;=\; \sum_{j=0}^{n_e} \sum_{\{\mathcal{G}_j\}} \Omega\left(\mathcal{Y}, \mathcal{P}_{\mathcal{G}_j}\right)
\tag{4.61}
$$

which provides a representation for the wavefunction universal integrand $\psi_{\mathcal{G}}$ because of the identification $\Omega(\mathcal{Y}, \mathcal{P}_{\mathcal{G}}) = \psi_{\mathcal{G}}(x, y)$.

---

[15] In this case we prefer to use the term *subdivision* given that each term does not necessarily correspond to a simplex.

To the elements of the collection $\{\mathcal{P}_{\mathcal{G}_j}, \{\mathcal{G}_j\}, j = 1, \ldots n_e\}$ one can also associate the following weight functions

$$w'\left(\mathcal{Y}, \mathcal{P}_{\mathcal{G}_j}\right) = (-1)^{n_e-j} w\left(\mathcal{Y}, \mathcal{P}_{\mathcal{G}_j}\right) \tag{4.62}$$

leaving the orientation unchanged. Then,

$$\sum_{j=0}^{n_e} \sum_{\{\mathcal{G}_j\}} w'\left(\mathcal{Y}, \mathcal{P}_{\mathcal{G}_j}\right) = (-1)^{n_e} w\left(\mathcal{Y}, \mathcal{P}_{\mathcal{G}}\right) \sum_{j=0}^{n_e} (-1)^j \sum_{\{\mathcal{E}_j\}} \prod_{e \in \mathcal{E}_j} \vartheta\left(\mathcal{Y} \cdot \widetilde{\mathcal{W}}^{(\mathfrak{g}_e)}\right) \prod_{e' \notin \mathcal{E}_j} \vartheta\left(-\mathcal{Y} \cdot \widetilde{\mathcal{W}}^{(\mathfrak{g}_{e'})}\right) =$$

$$= \text{sign}\left\{ \prod_{e \in \mathcal{E}} \left(\mathcal{Y} \cdot \widetilde{\mathcal{W}}^{(\mathfrak{g}_e)}\right) \right\} w\left(\mathcal{Y}, \mathcal{P}_{\mathcal{G}}\right) = w\left(\mathcal{Y}, \mathcal{P}_{\mathcal{G}}^{(w)}\right) \tag{4.63}$$

and the collection $\{\mathcal{P}_{\mathcal{G}_j}, \{\mathcal{G}_j\}, j = 0, \ldots, n_e\}$ provides a polytope subdivision for the weighted cosmological polytope $\mathcal{P}_{\mathcal{G}}^{(w)}$. In terms of the canonical function, we have

$$\Omega\left(\mathcal{Y}, \mathcal{P}_{\mathcal{G}}^{(w)}\right) = (-1)^{n_e} \sum_{j=0}^{n_e} (-1)^j \sum_{\{\mathcal{G}_j\}} \Omega\left(\mathcal{Y}, \mathcal{P}_{\mathcal{G}_j}\right) \tag{4.64}$$

Because of the identification (4.52), the canonical function subdivision (4.64) provides a novel representation for the cosmological correlator associated to the graph $\mathcal{G}$. It is useful to use the markings introduced in Section 4.1 in order to easily deduce from the graph the elements of this polytope subdivision and compute their canonical functions. First, recall that for a given triangle $\{\mathcal{Z}_1^{(e)}, \mathcal{Z}_2^{(e)}, \mathcal{Z}_3^{(e)}\}$ in the fundamental definition of the cosmological polytope, the hyperplane $\widetilde{\mathcal{W}}^{(\mathfrak{g}_e)}$ intersects it in its interior with $\mathcal{Z}_j^{(e)} \cdot \widetilde{\mathcal{W}}^{(\mathfrak{g}_e)} > 0$ for $j = 2, 3$ and $\mathcal{Z}_1^{(e)} \cdot \widetilde{\mathcal{W}}^{(\mathfrak{g}_e)} < 0$. Given a two site graph with associated the five vertices $\{\mathcal{Z}_j^{(e)}, j = 1, \ldots, 5\}$, the two polytopes identified by the intersection between $\mathcal{P}_{\mathcal{G}}^{(w)}$ and positive/negative half-plane of $\widetilde{\mathcal{W}}^{(\mathfrak{g}_e)}$ respectively correspond to the following markings

$$
\begin{array}{cc}
x_{s_e} \quad y_e \quad x_{s'_e} & x_{s_e} \quad y_e \quad x_{s'_e} \\
Z_{2,3}^{(e)} \cdot \mathcal{W}^{(\mathfrak{g}_e)} < 0 & Z_1^{(e)} \cdot \mathcal{W}^{(\mathfrak{g}_e)} > 0
\end{array}
$$

with $\textcolor{red}{+}$ excluding the corresponding vertex: thus, the first marking identifies a polytope in $\mathbb{P}^2$ which has all the vertices but $\mathcal{Z}_1^{(e)}$, while the second marking identifies a polytope in $\mathbb{P}^2$ with all the vertices but $\mathcal{Z}_2^{(e)}$ and $\mathcal{Z}_3^{(e)}$. Importantly, both of them are ordinary polytopes, with the one associated to the first marking being a quadrilateral and the second being a triangle: using the same analysis for the facets described earlier, it is easy to see that the first polytope has all the boundaries of the weighted cosmological polytope associated to the two site graph, but now with all the vertices lying only on the intersection of the positive half-spaces of the hyperplanes containing the facets; the second polytope instead is a simplex in $\mathbb{P}^2$ and, thus, a triangle – its intersection with the hyperplane $\mathcal{W}^{(\mathcal{G})}$ is empty in codimension-1 as it contains just the vertex $\mathcal{Z}_1^{(e)}$. Also, the first marking above identifies the set $\mathcal{E}_1 = \mathcal{E}$ of edges associated to the positive half-space of $\mathcal{W}^{(\mathfrak{g}_e)}$, while the second marking identifies $\mathcal{E}_\circ = \varnothing$ – we can label these marked graphs respectively as $\mathcal{G}_1$ and $\mathcal{G}_0$ and their associated polytopes as $\mathcal{P}_{\mathcal{G}_1}$ and $\mathcal{P}_{\mathcal{G}_0}$ respectively. Given an arbitrary graph $\mathcal{G}$, $\mathcal{G}_j$ is a graph with $j$ edges marked in the middle with $\textcolor{red}{+}$ – and hence with $n_e - j$ edges marked close to both their sites –, while $\{\mathcal{G}_j\}$ is the collection of all the possible inequivalent ways of marking this way the graph $\mathcal{G}$ – see Figure 6 for an example.

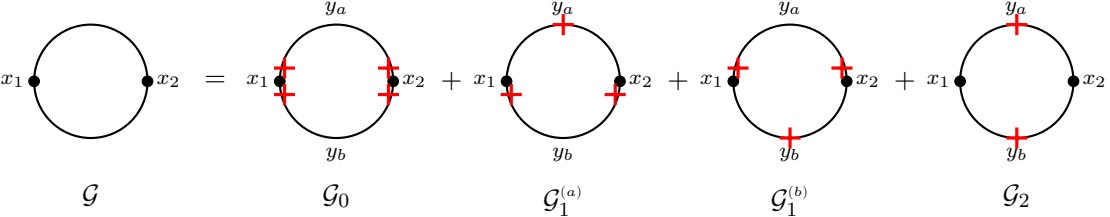

$\mathcal{G}$       $\mathcal{G}_0$       $\mathcal{G}_1^{(a)}$       $\mathcal{G}_1^{(b)}$       $\mathcal{G}_2$

Figure 6: Example of marked graphs for the elements of the fundamental polytope subdivision. Each term lives in the same space $\mathbb{P}^{n_s+n_e-1}$, and $\mathcal{G}_j$ indicates a graph with $j$ edges marked in the middle and $n_e - j$ of them close to both their sites.

A marked graph $\mathcal{G}_j$ shows $j + 2(n_e - j) = 2n_e - j$ markings. Consequently, the associated polytope $\mathcal{P}_{\mathcal{G}_j} \subset \mathbb{P}^{n_s+n_e-1}$ has $3n_e + n_s - (2n_e - j) = n_s + n_e + j$ vertices: just the element $\mathcal{P}_{\mathcal{G}_0}$ turns out to be a simplex. As all the elements of this collection are ordinary polytopes, their canonical form can be fixed by the knowledge of their facets and their compatibility conditions. In particular, following [29], let $\mathfrak{M}_o^{(j)} := \{\mathfrak{m}_r, \, r = 1, \ldots, k\}$ be the set of markings that identifies the hyperplane $\mathcal{W}^{(\alpha_1 \cdots \alpha_k)}$ such that $\mathcal{P}_{\mathcal{G}_j} \cap \mathcal{W}^{(\alpha_1 \cdots \alpha_k)} = \varnothing$[16]. Let also $\mathfrak{M}_c^{(j)}$ be the set of $(n_s + n_e - k)$ markings $\mathfrak{m} \notin \mathfrak{M}_o^{(j)}$ which are associated to compatible facets. Then, the canonical function associated to $\mathcal{P}_{\mathcal{G}_j}$ can be written as

$$\Omega\left(\mathcal{Y}, \mathcal{P}_{\mathcal{G}_j}\right) = \sum_{\{\mathfrak{M}_c^{(j)}\}} \prod_{\mathfrak{m}' \in \mathfrak{M}_c^{(j)}} \frac{1}{q_{\mathfrak{m}'}(\mathcal{Y})} \prod_{\mathfrak{m} \in \mathfrak{M}_o^{(j)}} \frac{1}{q_{\mathfrak{m}}(\mathcal{Y})}. \tag{4.65}$$

The canonical function subdivision given by (4.64), together with (4.65), provides a novel representation for the cosmological correlators which shows physical singularities only. In the next paragraph, we will see some explicit examples. However, let us conclude this one with one last comment.

It is possible to define a further collection of polytopes $\{\mathcal{P}_{\mathcal{G}(\mathcal{E}_j)}, \{\mathcal{E}_j\}, \, j = 0, \ldots, n_e\}$ via the following weight functions and orientations

$$w\left(\mathcal{Y}, \mathcal{P}_{\mathcal{G}(\mathcal{E}_j)}\right) = (-2)^j \prod_{e \in \mathcal{E}_j} \vartheta\left(\mathcal{Y} \cdot \widetilde{\mathcal{W}}^{(\mathfrak{g}_e)}\right),$$
$$O\left(\mathcal{Y}, \mathcal{P}_{\mathcal{G}(\mathcal{E}_j)}\right) = O\left(\mathcal{Y}, \mathcal{P}_{\mathcal{G}}\right) \tag{4.66}$$

This collection of polytopes turns out to represent a polytope subdivision for the weighted cosmological polytope $\mathcal{P}_{\mathcal{G}}^{(w)}$ as well. This is straightforward to see considering that the weight functions for $\mathcal{P}_{\mathcal{G}_j}$ multiplied by $(-1)^j$ form a subdivision of $\mathcal{P}_{\mathcal{G}}^{(w)}$. It is then sufficient to prove that

$$\sum_{\mathcal{E}_j \in \{\mathcal{E}_j\}} w\left(\mathcal{Y}, \mathcal{P}_{\mathcal{G}(\mathcal{E}_j)}\right) = (-1)^j \qquad \text{for } \mathcal{Y} \in \mathcal{P}_{\mathcal{G}_j}. \tag{4.67}$$

From the definition of $\mathcal{P}_{\mathcal{G}_j}$ and $\mathcal{P}_{\mathcal{G}(\mathcal{E}_j)}$ it directly follows that

$$\begin{cases} \mathcal{P}_{\mathcal{G}_j} \subseteq \mathcal{P}_{\mathcal{G}(\mathcal{E}_{j'})}, & \mathcal{E}_{j'} \subseteq \mathcal{E}_j \\ \\ \mathcal{P}_{\mathcal{G}_j} \cap \mathcal{P}_{\mathcal{G}(\mathcal{E}_{j'})} = \varnothing, & \text{otherwise} \end{cases} \tag{4.68}$$

---

[16]Recall that $\mathcal{W}^{(\alpha)} = \mathcal{W}^{(\mathfrak{g})} \, \forall \, \mathfrak{g} \subseteq \mathcal{G}$ or $\mathcal{W}^{(\alpha)} = \widetilde{\mathcal{W}}^{(\mathfrak{g}_e)} \, \forall \, e \in \mathcal{E}$.

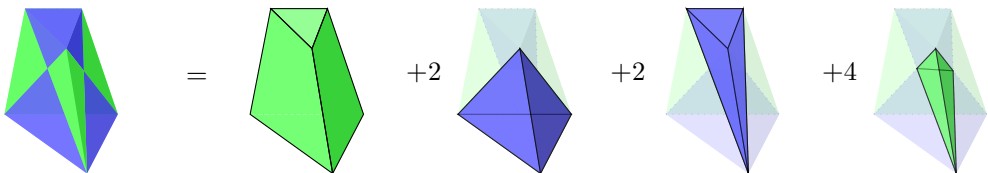

Figure 7: Polytope subdivision of the weighted cosmological polytope corresponding to a two-site one-loop cosmological correlator. The green shapes have positive weight and blue shapes have negative weight. Note that, for overlapping polytopes on the right-hand side, the sum over the weights correctly reproduces the expected $\pm 1$ weight. The right-hand side provides the diagrammatics (2.29). The canonical function of the first polytope in the r.h.s. is related to the 2-site 1-loop graph for the wavefunction; the second and the third terms are related to the connected wavefunction graph obtained by erasing one edge; the last polytope provides the disconnected contribution obtained by erasing both edges in the wavefunction graph.

The sum (4.67) therefore reduces to

$$\sum_{\mathcal{E}_{j'} \subseteq \mathcal{E}_j} (-2)^{j'} w\left(\mathcal{Y}, \mathcal{P}_{\mathcal{G}_{j'}}\right) = \sum_{\mathcal{E}_{j'} \subseteq \mathcal{E}_j} (-2)^{j'} = \sum_{j'=0}^{j} (-2)^{j'} \binom{j}{j'} = (-1)^j \tag{4.69}$$

where it has been used the fact that the number of $\mathcal{E}_{j'}$, $\dim\{\mathcal{E}_{j'}\} = j'$ is given by the binomial above as well as the relation

$$\sum_{r=0}^{n} \binom{n}{r} x^r = (1+x)^n \ , \tag{4.70}$$

for $x = -2$. Hence, (4.69) proves equation (4.67).

At the level of the canonical function, the polytope subdivision of $\mathcal{P}_{\mathcal{G}}^{(w)}$ provided by the collection $\{\mathcal{P}_{\mathcal{G}(\mathcal{E}_j)}, \{\mathcal{E}_j\}, j = 0, \ldots, n_e\}$, can be written as

$$\Omega\left(\mathcal{Y}, \mathcal{P}_{\mathcal{G}}^{(w)}\right) = \sum_{j=0}^{n_e} (-2)^j \sum_{\{\mathcal{E}_j\}} \Omega\left(\mathcal{Y}, \mathcal{P}_{\mathcal{G}(\mathcal{E}_j)}\right) \tag{4.71}$$

Note that the polytope $\mathcal{P}_{\mathcal{G}(\mathcal{E}_j)}$ has $j \in [0, n_e]$ facets associated to the hyperplanes $\mathcal{W}^{(\mathfrak{g}_{e_r})}$, while all the others are identified by a subset of the subgraphs $\{\mathfrak{g}, \mathfrak{g} \subseteq \mathcal{G}\}$. This is precisely the structure of the diagrammatic rules emerging from the definition of the cosmological correlators as a path integral of the relevant operator with the probability distribution provided by the squared-modulus of the wavefunction.

**An illustrative example** – It is useful to discuss a simple example in some detail in order to fix the ideas. We consider the two-site one-loop graph: despite its simplicity, it provides a sufficiently rich example that works as a prototype for the computation of the canonical function of weighted cosmological polytopes associated to arbitrary graphs.

The two-site one-loop graph has an associated cosmological polytope which is a prism in $\mathbb{P}^3$, while the related weighted cosmological polytope is a weighted prism with the very same five ordinary facets

as the prism, and two additional internal boundaries:

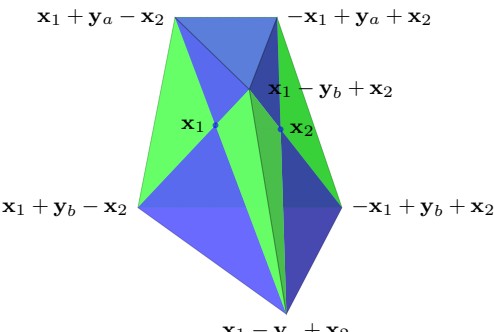

where the regions with positive weight (or orientation) have been represented in green and those with negative weight in blue. This subdivision can be graphically represented as

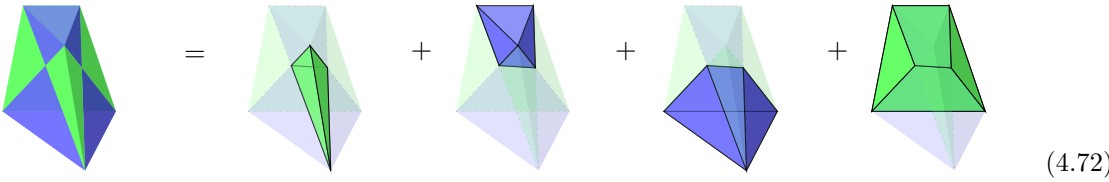

$$(4.72)$$

which correspond, term by term, to the graphs in Figure 6 and, thus, they are respectively $\mathcal{P}_{\mathcal{G}_0}$, $\mathcal{P}_{\mathcal{G}_1^{(a)}}$, $\mathcal{P}_{\mathcal{G}_1^{(b)}}$, and $\mathcal{P}_{\mathcal{G}_2}$. As it is apparent from (4.72) and from the counting of the linearly independent vertices in Figure 6, $\mathcal{P}_{\mathcal{G}_0}$ is a simplex with facets identified by the hyperplanes $\mathcal{W}^{(\mathfrak{g}_1)}$, $\mathcal{W}^{(\mathfrak{g}_2)}$, $\mathcal{W}^{(e_a)}$, $\mathcal{W}^{(e_b)}$, with $e_a$ and $e_b$ respectively labeling the upper and lower edges of the graph. Its canonical function is then given by

$$\Omega\left(\mathcal{Y},\mathcal{P}_{\mathcal{G}_0}\right) = \frac{\langle 41\mathbf{x}_2\mathbf{x}_1\rangle^3}{q_{\mathfrak{m}_1}(\mathcal{Y})q_{\mathfrak{m}_2}(\mathcal{Y})q_{\mathfrak{m}_a}(\mathcal{Y})q_{\mathfrak{m}_b}(\mathcal{Y})} = \frac{\langle 41\mathbf{x}_2\mathbf{x}_1\rangle^3}{\langle\mathcal{Y}\mathbf{x}_141\rangle\langle\mathcal{Y}\mathbf{x}_214\rangle\langle\mathcal{Y}\mathbf{x}_2\mathbf{x}_14\rangle\langle\mathcal{Y}\mathbf{x}_1\mathbf{x}_21\rangle} =$$
$$= \frac{1}{(y_a+x_2+y_b)(y_a+x_1+y_b)y_ay_b} \ .$$

$$(4.73)$$

Let us move to $\mathcal{P}_{\mathcal{G}_1^{(a/b)}}$, which are both pyramids. For defininess, let us focus on $\mathcal{P}_{\mathcal{G}_1^{(a)}}$

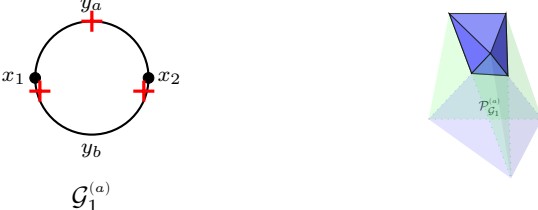

It has two sets of zeroes given by the locus of the intersection of the hyperplanes containing its non-adjacent facets. They can be identified from the graph via those subgraphs whose associated

markings eliminate all the vertices and are complementary, *i.e.* the markings do not overlap

$$(4.74)$$

Note that the first two markings in each line identify two incompatible facets: just the vertex $\mathcal{Z}_1^{(e_b)}$ live on their intersection, that is not enough to span $\mathbb{P}^1$ which is the space where a codimension-2 face should live. The two graphs in each line determine the two lines $\mathcal{Z}_{(A)}^{IJ}$ and $\mathcal{Z}_{(B)}^{IJ}$ on the adjoint that intersect each other in $\mathcal{Z}_1^{(e_b)}$. The last graph in each line is the same and defines two points $\mathcal{Z}_{(A\star)}$ and $\mathcal{Z}_{(B\star)}$ on the adjoint. The adjoint is a 2-plane $\mathcal{C}_I^{(1a)}$ which can be identified by any of the two pairs $\left(\mathcal{Z}_{(A)}^{IJ}, \mathcal{Z}_{(B\star)}^{K}\right)$ and $\left(\mathcal{Z}_{(A\star)}^{I}, \mathcal{Z}_{(B)}^{JK}\right)$

$$
\begin{aligned}
\mathcal{Z}_{(A)}^{IJ} &= \epsilon^{IJKL}\mathcal{W}_K^{(\mathfrak{g}_1)}\mathcal{W}_L^{(\mathfrak{g}_2)}, & \mathcal{Z}_{(A\star)}^{I} &= \epsilon^{IJKL}\widetilde{\mathcal{W}}_J^{(\mathfrak{g}_{12a})}\mathcal{W}_K^{(\mathfrak{g}_1)}\mathcal{W}_L^{(\mathfrak{g}_2)}, \\
\mathcal{Z}_{(B)}^{IJ} &= \epsilon^{IJKL}\mathcal{W}_K^{(\mathfrak{g}_{12b})}\widetilde{\mathcal{W}}_L^{(\mathfrak{g}_{12b})}, & \mathcal{Z}_{(B)}^{I} &= \epsilon^{IJKL}\widetilde{\mathcal{W}}_J^{(\mathfrak{g}_{12a})}\mathcal{W}_K^{(\mathfrak{g}_{12b})}\widetilde{\mathcal{W}}_L^{(\mathfrak{g}_{12b})}, \\
\mathcal{C}_I^{(1a)} &= \epsilon_{IJKL}\mathcal{Z}_{(A)}^{JK}\mathcal{Z}_{(B\star)}^{L} = \epsilon_{IJKL}\mathcal{Z}_{(A\star)}^{J}\mathcal{Z}_{(B)}^{KL}.
\end{aligned}
$$

$$(4.75)$$

Hence, the canonical function of $\mathcal{P}_{\mathcal{G}_1}^{(a)}$ can be written as

$$
\begin{aligned}
\Omega\left(\mathcal{Y}, \mathcal{P}_{\mathcal{G}_1^{(a)}}\right) &= \frac{\left(\mathcal{Y}\cdot\mathcal{C}^{(1a)}\right)}{\left(\mathcal{Y}\cdot\mathcal{W}^{(\mathfrak{g}_1)}\right)\left(\mathcal{Y}\cdot\mathcal{W}^{(\mathfrak{g}_2)}\right)\left(\mathcal{Y}\cdot\mathcal{W}^{(\mathfrak{g}_{12b})}\right)\left(\mathcal{Y}\cdot\widetilde{\mathcal{W}}^{(\mathfrak{g}_{12a})}\right)\left(\mathcal{Y}\cdot\widetilde{\mathcal{W}}^{(\mathfrak{g}_{12b})}\right)} = \\
&= \frac{x_1+x_2+2y_a+2y_b}{(y_a+x_2+y_b)(y_a+x_1+y_b)(x_1+x_2+2yb)y_by_a}
\end{aligned}
$$

$$(4.76)$$

More instructively, the knowledge of (4.74) allows us to write the canonical function of the polytope $\mathcal{P}_{\mathcal{G}_1}^{(a)}$ as a triangulation via subspaces in its adjoint via (4.65). Choosing $\mathfrak{M}_\circ$ to be the set of markings in the first line, $\{\mathfrak{M}_c\}$ has two elements given by a single marking each corresponding to the two remaining ones (the first two marking in the second line):

$$
\begin{aligned}
\Omega\left(\mathcal{Y}, \mathcal{P}_{\mathcal{G}_1^{(a)}}\right) &= \frac{1}{\left(\mathcal{Y}\cdot\mathcal{W}^{(\mathfrak{g}_1)}\right)\left(\mathcal{Y}\cdot\mathcal{W}^{(\mathfrak{g}_2)}\right)\left(\mathcal{Y}\cdot\widetilde{\mathcal{W}}^{(\mathfrak{g}_{12a})}\right)}\left[\frac{\langle A\star234\rangle}{\mathcal{Y}\cdot\mathcal{W}^{(\mathfrak{g}_{12b})}} + \frac{\langle A\star\mathbf{x}_2\mathbf{x}_14\rangle}{\mathcal{Y}\cdot\widetilde{\mathcal{W}}^{(\mathfrak{g}_{12b})}}\right] = \\
&= \frac{1}{(y_a+x_2+y_b)(y_a+x_1+y_b)y_b}\left[\frac{2}{x_1+x_2+2y_b}+\frac{1}{y_a}\right]
\end{aligned}
$$

$$(4.77)$$

Alternatively, we can choose $\mathfrak{M}_\circ$ to be the second line in (4.74), and $\{\mathfrak{M}_c\}$ to have the first two

markings as elements. Then

$$
\Omega\left(\mathcal{Y}, \mathcal{P}_{\mathcal{G}_1^{(a)}}\right) = \frac{1}{\left(\mathcal{Y}\cdot\mathcal{W}^{(\mathfrak{g}_{12b})}\right)\left(\mathcal{Y}\cdot\widetilde{\mathcal{W}}^{(\mathfrak{g}_{12b})}\right)\left(\mathcal{Y}\cdot\widetilde{\mathcal{W}}^{(\mathfrak{g}_{12a})}\right)}\left[\frac{\langle B\star\mathbf{x}_2 43\rangle}{\mathcal{Y}\cdot\mathcal{W}^{(\mathfrak{g}_1)}} + \frac{\langle B\star\mathbf{x}_1 42\rangle}{\mathcal{Y}\cdot\mathcal{W}^{(\mathfrak{g}_2)}}\right] =
$$

$$
= \frac{1}{(x_1+x_2+2y_b)y_a y_b}\left[\frac{1}{y_a+x_2+y_b} + \frac{1}{y_a+x_1+y_b}\right]
$$

(4.78)

As $\mathcal{P}_{\mathcal{G}_1^{(b)}}$ is isomorphic to $\mathcal{P}_{\mathcal{G}_1^{(a)}}$, its canonical function can be obtained from the former via the label exchange $e_b \longleftrightarrow e_a$ or, locally, via $y_b \longleftrightarrow y_a$.

The last element $\mathcal{P}_{\mathcal{G}_2}$ of the polytope subdivision is a prism

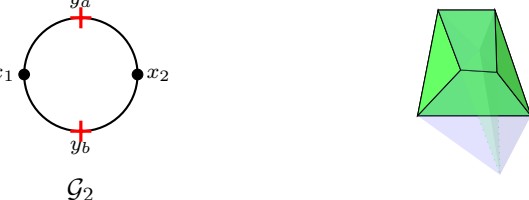

$$\mathcal{G}_2$$

and its analysis follows precisely the one we just carried out for $\mathcal{P}_{\mathcal{G}_1^{(a)}}$. Its adjoint is fixed by the subspaces identified by the following two sets of markings

$$(4.79)$$

The three markings in the first line identify the point $\mathcal{Z}_{(A)}^I := \epsilon^{IJKL}\mathcal{W}_J^{(\mathcal{G})}\widetilde{\mathcal{W}}_K^{(\mathfrak{g}_{12b})}\widetilde{\mathcal{W}}_L^{(\mathfrak{g}_{12a})}$, while the two in the second line the line $\mathcal{Z}_{(B)}^{IJ} := \epsilon^{IJKL}\mathcal{W}_K^{(\mathfrak{g}_1)}\mathcal{W}_L^{(\mathfrak{g}_2)}$. Thus, the adjoint is a plane identified by the co-vector $\mathcal{C}_I^{(2)} := \epsilon_{IJKL}\mathcal{Z}_{(A)}^J\mathcal{Z}_{(B)}^{KL}$. As in the previous case, with this information at hand, we can directly write the canonical function associated to $\mathcal{P}_{\mathcal{G}_2}$:

$$
\Omega\left(\mathcal{Y}, \mathcal{P}_{\mathcal{G}_2}\right) = \frac{\left(\mathcal{Y}\cdot\mathcal{C}^{(2)}\right)}{\left(\mathcal{Y}\cdot\mathcal{W}^{(\mathcal{G})}\right)\left(\mathcal{Y}\cdot\mathcal{W}^{(\mathfrak{g}_1)}\right)\left(\mathcal{Y}\cdot\mathcal{W}^{(\mathfrak{g}_2)}\right)\left(\mathcal{Y}\cdot\widetilde{\mathcal{W}}^{(\mathfrak{g}_{12b})}\right)\left(\mathcal{Y}\cdot\widetilde{\mathcal{W}}^{(\mathfrak{g}_{12a})}\right)} =
$$

$$
= \frac{x_1+x_2+2y_a+2y_b}{(x_1+x_2)(y_a+x_1+y_b)(y_a+x_2+y_b)y_a y_b}
$$

(4.80)

Triangulating instead via the subspaces of the adjoint identified by the two lines in (4.79), the canonical

function can be written as

$$\Omega\left(\mathcal{Y}, \mathcal{P}_{\mathcal{G}_2}\right) = \frac{1}{(\mathcal{Y}\cdot\mathcal{W}^{(\mathcal{G})})\left(\mathcal{Y}\cdot\widetilde{\mathcal{W}}^{(\mathfrak{g}_{12b})}\right)\left(\mathcal{Y}\cdot\widetilde{\mathcal{W}}^{(\mathfrak{g}_{12a})}\right)}\left[\frac{\langle A\mathbf{x}_2 36\rangle^3}{(\mathcal{Y}\cdot\mathcal{W}^{(\mathfrak{g}_1)})} + \frac{\langle A\mathbf{x}_1 25\rangle^3}{(\mathcal{Y}\cdot\mathcal{W}^{(\mathfrak{g}_2)})}\right] =$$
$$= \frac{1}{(x_1+x_2)y_a y_b}\left[\frac{1}{y_a+x_2+y_b} + \frac{1}{y_a+x_1+y_b}\right] \tag{4.81}$$

and

$$\Omega\left(\mathcal{Y}, \mathcal{P}_{\mathcal{G}_2}\right) = \frac{1}{(\mathcal{Y}\cdot\mathcal{W}^{(\mathfrak{g}_1)})(\mathcal{Y}\cdot\mathcal{W}^{(\mathfrak{g}_2)})}\left[\frac{\langle 3B_{\mathcal{G}}\widetilde{B}_b 2\rangle^3}{(\mathcal{Y}\cdot\mathcal{W}^{(\mathcal{G})})\left(\mathcal{Y}\cdot\widetilde{\mathcal{W}}^{(\mathfrak{g}_{12b})}\right)} + \frac{\langle X_1\widetilde{B}_a\widetilde{B}_b X_2\rangle^3}{\left(\mathcal{Y}\cdot\widetilde{\mathcal{W}}^{(\mathfrak{g}_{12b})}\right)\left(\mathcal{Y}\cdot\widetilde{\mathcal{W}}^{(\mathfrak{g}_{12a})}\right)} + \right.$$
$$\left. + \frac{\langle 6B_{\mathcal{G}}\widetilde{B}_a 5\rangle^3}{\left(\mathcal{Y}\cdot\widetilde{\mathcal{W}}^{(\mathfrak{g}_{12a})}\right)(\mathcal{Y}\cdot\mathcal{W}^{(\mathcal{G})})}\right] =$$
$$= \frac{1}{(y_a+x_2+y_b)(y_a+x_1+y_b)}\left[\frac{2}{(x_1+x_2)y_a} + \frac{1}{y_a y_b} + \frac{2}{y_b(x_1+x_2)}\right] \tag{4.82}$$

where $B_{\mathcal{G}}$, $\widetilde{B}_a$ and $\widetilde{B}_b$ appearing in the numerators label the vectors $\mathcal{Z}^I_{B_{\mathcal{G}}} := Z^{IJ}_B\mathcal{W}^{(\mathcal{G})}_J$, $\mathcal{Z}^I_{B_a} := Z^{IJ}_B\widetilde{\mathcal{W}}^{(\mathfrak{g}_{12a})}_J$, and $\mathcal{Z}^I_{B_b} := Z^{IJ}_B\widetilde{\mathcal{W}}^{(\mathfrak{g}_{12b})}_J$ respectively.

We can now write explicitly, in the local coordinates $\mathcal{Y} := (x_1, y_a, y_b, x_2)$, the canonical function for the weighted cosmological polytope associated to the 2-site one-loop graph

$$\Omega\left(\mathcal{Y}, \mathcal{P}^{(w)}_{\mathcal{G}}\right) = \frac{1}{(y_a+x_2+y_b)(y_a+x_1+y_b)y_a y_b} + \frac{x_1+x_2+2y_a+2y_b}{(y_a+x_2+y_b)(y_a+x_1+y_b)(x_1+x_2+2y_b)y_a y_b} +$$
$$+ \frac{x_1+x_2+2y_a+2y_b}{(y_a+x_2+y_b)(y_a+x_1+y_b)(x_1+x_2+2y_a)y_a y_b} +$$
$$+ \frac{x_1+x_2+2y_a+2y_b}{(x_1+x_2)(y_a+x_2+y_b)(y_a+x_1+y_b)y_a y_b} \tag{4.83}$$

A comment is now in order. We have seen how, for each element of the polytope subdivision we have considered, the knowledge of the intersections of the hyperplanes containing its facets outside it allows us to invariantly write the canonical function, irrespectively of any triangulation. Then, why bother writing, for each term, a triangulation? For arbitrary polytopes, whose adjoint surface can be represented by a homogeneous polynomial of degree higher than 1, it is not in general straightforward to write such invariant expression. Even if in this particular case was not needed, we illustrated how the procedure originally formulated in [29] for cosmological polytopes, translates to the elements of our polytope subdivision: it is completely general and works for arbitrary (marked) graphs.

**Boundaries of $\mathcal{P}^{(w)}_{\mathcal{G}}$ and discontinuities of $\mathcal{C}_{\mathcal{G}}$** − So far we have discussed how the knowledge of the weighted cosmological polytopes allows for novel ways of computing cosmological correlators. However, the compatibility conditions determining their higher codimension faces provides us with information about the discontinuity structure of the associated correlators.

Let us begin with considering the individual discontinuities. As already discussed, they correspond to the facets of the weighted cosmological polytopes. It is straightforward to see that

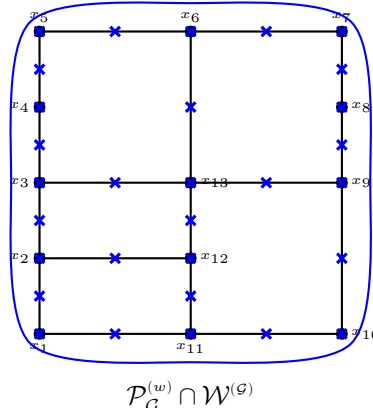 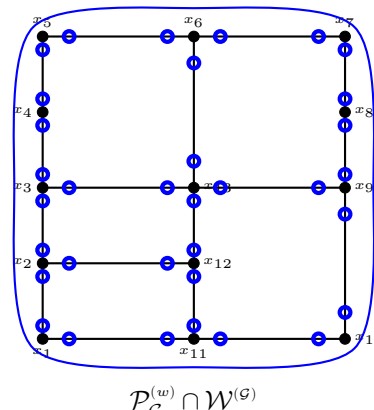

$$\mathcal{P}_{\mathcal{G}}^{(w)} \cap \mathcal{W}^{(\mathcal{G})} \qquad\qquad \mathcal{P}_{\mathcal{G}}^{(w)} \cap \mathcal{W}^{(\mathcal{G})}$$

Figure 8: Vertex configuration for the facet $\mathcal{P}_{\mathcal{G}}^{(w)} \cap \mathcal{W}^{(\mathcal{G})}$. *On the left* – The marking rules reveal those vertices which *are not* on this facet. *On the right* – The open circles indicate instead the vertices *on* this facet. This is useful to recognise the organization of the vertices on this facet. In this case, the vertex structure is precisely one of the scattering facets of the cosmological polytope associated with the same graph and hence the canonical function returns the flat-space scattering amplitude.

❏ the facet $\mathcal{P}_{\mathcal{G}}^{(w)} \cap \mathcal{W}^{(G)}$ is just the scattering facet of the cosmological polytope, as the only vertices on such a facet are just $\left\{ \mathcal{Z}_2^{(e)},\ \mathcal{Z}_3^{(e)},\ \forall\, e \in \mathcal{E} \right\}$ – see Figure 8. Thus, the canonical function for this facet is just the contribution from the graph $\mathcal{G}$ to the flat-space scattering amplitude $\mathcal{A}_{\mathcal{G}}$

$$\mathrm{Res}_{\mathcal{W}^{(G)}} \Omega\left(\mathcal{Y}, \mathcal{P}_{\mathcal{G}}^{(w)}\right) \;=\; \Omega\left(\mathcal{Y}, \mathcal{P}_{\mathcal{G}}^{(w)} \cap \mathcal{W}^{(G)}\right) \;=\; \mathcal{A}_{\mathcal{G}}. \tag{4.84}$$

❏ the facets $\mathcal{P}_{\mathcal{G}} \cap \mathcal{W}^{(\mathfrak{g})},\ \forall\, \mathfrak{g} \subset \mathcal{G}$ factorise into a lower dimensional scattering facet and a weighted polytope $\mathcal{P}_{\mathfrak{g}\cup\mathscr{E}}^{(w)}$ associated to $\overline{\mathfrak{g}} \cup \mathscr{E}$ – as usual $\overline{\mathfrak{g}}$ is defined by the vertices $\mathcal{V}_{\overline{\mathfrak{g}}} = \mathcal{V} \setminus \mathcal{V}_{\mathfrak{g}}$ of $\mathcal{G}$ which are not in $\mathfrak{g}$, connected as they are in $\mathcal{G}$, and $\mathscr{E}$ are the edges connecting $\mathfrak{g}$ and $\overline{\mathfrak{g}}$

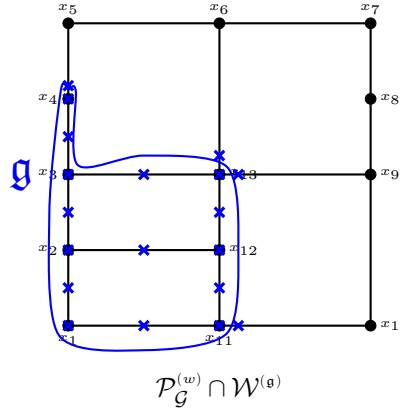 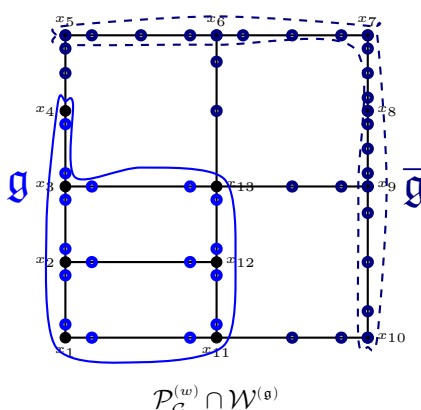

$$\mathcal{P}_{\mathcal{G}}^{(w)} \cap \mathcal{W}^{(\mathfrak{g})} \qquad\qquad \mathcal{P}_{\mathcal{G}}^{(w)} \cap \mathcal{W}^{(\mathfrak{g})}$$

and hence the canonical form factorises as

$$\mathrm{Res}_{\mathcal{W}^{(\mathfrak{g})}} \Omega\left(\mathcal{Y}, \mathcal{P}_{\mathcal{G}}^{(w)}\right) \;=\; \Omega\left(\mathcal{Y}, \mathcal{P}_{\mathcal{G}}^{(w)} \cap \mathcal{W}^{(\mathfrak{g})}\right) \;=\; \Omega\left(\mathcal{Y}_{\mathfrak{g}}, \mathcal{S}_{\mathfrak{g}}\right) \times \Omega\left(\mathcal{Y}_{\overline{\mathfrak{g}}\cup\mathscr{E}}, \mathcal{P}_{\overline{\mathfrak{g}}\cup\mathscr{E}}^{(w)}\right) \tag{4.85}$$

with the canonical function of the lower-dimensional scattering facet providing a lower-dimensional flat-space amplitude. What about the second canonical function in (4.85)? Let us consider $\overline{\mathfrak{g}}$ to be a connected graph, as in the picture above. Note that there are two vertices of $\mathcal{P}_{\overline{\mathfrak{g}} \cup \mathcal{E}}$ associated to each edges in $\mathcal{E}$. Together with the two vertices on any of the adjacent edges, they identify a quadrilateral. It is then possible to define a polytope subdivision for $\mathcal{P}_{\overline{\mathfrak{g}} \cup \mathcal{E}}$ such that each of its elements involves only one of the two vertices on a given edge in $\mathcal{E}$, triangulating the quadrilateral. Each element of this polytope subdivision has therefore the same vertex structure of a lower-dimensional dimensional weighted cosmological polytope $\mathcal{P}_{\overline{\mathfrak{g}}}^{(w)}$ associated to $\overline{\mathfrak{g}}$, and a simplex $\Sigma_{\mathcal{E}}$ defined by the vertices associated to edges in $\mathcal{E}$

$$\Omega\left(\mathcal{Y}_{\overline{\mathfrak{g}} \cup \mathcal{E}}, \mathcal{P}_{\overline{\mathfrak{g}} \cup \mathcal{E}}^{(w)}\right) = \sum_{\{\Sigma_{\mathcal{E}}\}} \Omega\left(\mathcal{Y}_{\mathcal{E}}, \Sigma_{\mathcal{E}}\right) \times \Omega\left(\mathcal{Y}_{\mathfrak{g}}(\Sigma_{\mathcal{E}}), \mathcal{P}_{\overline{\mathfrak{g}}}^{(w)}\right) \tag{4.86}$$

with the sum running over simplices formed by the different arrangements of the vertices associated to the edges in $\mathcal{E}$. The canonical function associated to $\mathcal{P}_{\mathfrak{g} \cup \mathcal{E}}^{(w)}$ thus returns a linear combination of smaller correlators, whose energies associated to the site attached to the edges in $\mathcal{E}$ are shifted by $\pm$ the energies of the relevant edges themselves, depending on which vertex associated to $e \in \mathcal{E}$ is involved:

$$\Omega\left(\mathcal{Y}_{\overline{\mathfrak{g}} \cup \mathcal{E}}, \mathcal{P}_{\overline{\mathfrak{g}} \cup \mathcal{E}}^{(w)}\right) = \left(\prod_{e \in \mathcal{E}} \frac{1}{2y_e}\right) \sum_{\{\sigma_e = \pm\}} \mathcal{C}_{\overline{\mathfrak{g}}}\left(x_s(\sigma_e), y_e\right) \tag{4.87}$$

where $x_s(\sigma_e)$ is the shifted energy associated to the site $s$ defined as

$$x_s(\sigma_e) := x_s + \sum_{e \in \mathcal{E} \cap \mathcal{E}_s} \sigma_e y_e \tag{4.88}$$

with $\mathcal{E}_s$ denoting the set of edges departing from the site $s$ – in this way the energies associated with the sites which are not connected to the edges in $\mathcal{E}$ do not get shifted. Thus, the canonical function of $\mathcal{P}_{\mathcal{G}}^{(w)} \cap \mathcal{W}^{(\mathfrak{g})}$ and, consequently, the residue of the correlator $\mathcal{C}_{\mathcal{G}}$ with respect to the total energy of a subprocess $\mathfrak{g} \subset \mathcal{G}$ factorises into a lower-point/lower-level flat-space scattering amplitude and a linear combination of lower-point/lower-level correlators

$$\text{Res}_{E_{\mathfrak{g}}} \mathcal{C}_{\mathcal{G}} = \mathcal{A}_{\mathfrak{g}} \times \left(\prod_{e \in \mathcal{E}} \frac{1}{2y_e}\right) \sum_{\{\sigma_e = \pm\}} \mathcal{C}_{\overline{\mathfrak{g}}}\left(x_s(\sigma_e), y_e\right) \tag{4.89}$$

The formula (4.87), and therefore (4.89), assume that $\overline{\mathfrak{g}}$ is a connected graph. However, it can be also disconnected

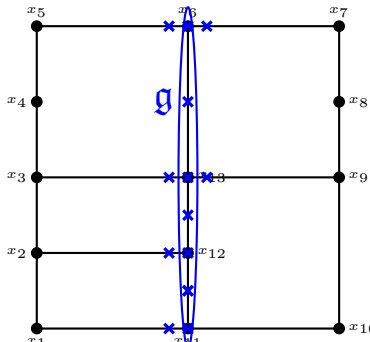


In such a case, the marking shows explicitly how the vertex structure encountered earlier for $\overline{\mathfrak{g}} \cup \mathscr{E}$ gets replicated for each $\overline{\mathfrak{g}}_j \cup \mathscr{E}_j$, with $\bigcup_j \overline{\mathfrak{g}}_j = \overline{\mathfrak{g}}$ and $\bigcup_j \mathscr{E}_j = \mathscr{E}$. Thus, the canonical function of $\mathcal{P}_{\mathcal{G}}^{(w)} \cap \mathcal{W}^{(\mathfrak{g})}$ factorises into the canonical function of the scattering facet associated to $\mathfrak{g}$ and the ones of each polytope associated to $\overline{\mathfrak{g}}_j \cup \mathscr{E}_j$:

$$\Omega\left(\mathcal{Y}, \mathcal{P}_{\mathcal{G}}^{(w)} \cap \mathcal{W}^{(\mathfrak{g})}\right) = \Omega\left(\mathcal{Y}_{\mathfrak{g}}, \mathcal{S}_{\mathfrak{g}}\right) \times \prod_{\{\overline{\mathfrak{g}}_j \cup \mathscr{E}_j\}} \Omega\left(\mathcal{Y}_{\overline{\mathfrak{g}}_j \cup \mathscr{E}_j}, \mathcal{P}_{\overline{\mathfrak{g}}_j \cup \mathscr{E}_j}\right) \tag{4.90}$$

As a matter of physical interpretation, the canonical function for each lower-dimensional polytope $\mathcal{P}_{\overline{\mathfrak{g}}_j \cup \mathscr{E}_j}$ returns a linear combination of lower-point/lower-level correlators, as in (4.87). Thus:

$$\mathrm{Res}_{E_{\mathfrak{g}}} \mathcal{C}_{\mathcal{G}} = \mathcal{A}_{\mathfrak{g}} \times \left(\prod_{\ell \in \mathscr{E}} \frac{1}{2 y_\ell}\right) \times \prod_{\{\overline{g}_j\}} \sum_{\{\sigma_\ell\}=\pm} \mathcal{C}_{\mathfrak{g}_j}\left(x_s(\sigma_\ell), y_e\right) \tag{4.91}$$

where now

$$x_s(\sigma_\ell) := x_s + \sum_{\ell \in \mathscr{E}_j \cap \mathcal{E}_s} \sigma_\ell y_\ell. \tag{4.92}$$

❑ the facets $\mathcal{P}_{\mathcal{G}}^{(w)} \cap \widetilde{\mathcal{W}}^{(\mathfrak{g}_e)} \; \forall e \in \mathcal{E}$ are weighted cosmological polytopes associated to the graph $\mathcal{G}_e := \mathcal{G} \setminus \{e\}$ obtained from $\mathcal{G}$ by removing the edge $e$ contained in $\mathfrak{g}_e$: from the marking rules, the vertices of $\mathcal{P}_{\mathcal{G}}^{(w)}$ with are on $\mathcal{P}_{\mathcal{G}}^{(w)} \cap \widetilde{\mathcal{W}}^{(\mathfrak{g}_e)}$ are all but the ones associated to the edge e in $\mathfrak{g}_e$. Depending on the topology of $\mathcal{G}$, the graph $\mathcal{G}_e$ can be either connected or disconnected. Therefore the canonical form is given by

$$\Omega\left(\mathcal{Y}, \mathcal{P}_{\mathcal{G}}^{(w)} \cap \widetilde{\mathcal{W}}^{(\mathfrak{g}_e)}\right) = \begin{cases} \Omega\left(\mathcal{Y}_{\mathcal{G}_e}, \mathcal{P}_{\mathcal{G}_e}^{(w)}\right), & \text{for } \mathcal{G}_e \text{ connected} \\[3mm] \Omega\left(\mathcal{Y}_{\mathfrak{g}_{s_e}}, \mathcal{P}_{\mathfrak{g}_{s_e}}^{(w)}\right) \times \Omega\left(\mathcal{Y}_{\mathfrak{g}_{s'_e}}, \mathcal{P}_{\mathfrak{g}_{s'_e}}^{(w)}\right), & \text{for } \mathcal{G}_e = \mathfrak{g}_{s_e} \cup \mathfrak{g}_{s'_e} \end{cases} \tag{4.93}$$

where $s_e$ and $s'_e$ are the endpoints of the erased edge $e$, and $\mathfrak{g}_{s_e}$ and $\mathfrak{g}_{s'_e}$ are the subgraphs containing $s_e$ and $s'_e$ respectively. This implies that the residue of a correlator with respect to any of the internal energy is either a lower-point/lower-level correlator or a product of two lower-point/lower-level correlators:

$$\mathrm{Res}_{y_e} \mathcal{C}_{\mathcal{G}} = \begin{cases} \mathcal{C}_{\mathcal{G}_e}, & \text{for } \mathcal{G}_e \text{ connected} \\[3mm] \mathcal{C}_{\mathfrak{g}_{s_e}} \times \mathcal{C}_{\mathfrak{g}_{s'_e}}, & \text{for } \mathcal{G}_e = \mathfrak{g}_{s_e} \cup \mathfrak{g}_{s'_e} \end{cases} \tag{4.94}$$

Let us now turn to the non-vanishing higher-codimension discontinuities. They correspond to codimension-$k$ faces of the weighted cosmological polytope, *i.e.* the intersections $\mathcal{P}_{\mathcal{G}}^{(w)} \cap \mathcal{W}^{(\mathfrak{g}_1 \cdots \mathfrak{g}_k)}$ satisfying the compatibility condition (4.51). The analysis that led to the compatibility condition showed that such faces factorise into $k - \not{n}$ lower-dimensional scattering facets and a polytope associated to $\overline{\mathfrak{g}} \cup \mathscr{E} := \left(\bigcup_{j=1}^{k} \overline{\mathfrak{g}}_j\right) \cup \mathscr{E}$, with this latter polytope can be factorised if $\overline{\mathfrak{g}}$ is disconnected. Hence, the canonical form factorises accordingly

$$\Omega\left(\mathcal{Y}, \mathcal{P}_{\mathcal{G}}^{(w)} \cap \mathcal{W}^{(\mathfrak{g}_1 \cdots \mathfrak{g}_k)}\right) = \prod_{\{\mathcal{S}_{\mathfrak{g}}\}} \Omega\left(\mathcal{Y}_{\mathfrak{g}}, \mathcal{S}_{\mathfrak{g}}\right) \times \Omega\left(\mathcal{Y}_{\overline{\mathfrak{g}}}, \mathcal{P}_{\overline{\mathfrak{g}} \cup \mathscr{E}}^{(w)}\right), \tag{4.95}$$

with the sum running on the set of lower-dimensional scattering facets in which the face $\mathcal{P}_\mathcal{G}^{(w)} \cap \mathcal{W}^{((\mathfrak{g}_1 \cdots \mathfrak{g}_k))}$. Therefore, the correlator $\mathcal{C}_\mathcal{G}$ factorises as sequential, compatible, discontinuities are approached

$$\text{Res}_{\mathcal{W}^{(\mathfrak{g}_1 \cdots \mathfrak{g}_k)}} \mathcal{C}_\mathcal{G} = \prod_{\{\mathfrak{g}\}} \mathcal{A}_\mathfrak{g} \times \prod_{\ell \in \mathcal{E}} \frac{1}{2y_\ell} \times \sum_{\{\sigma_\ell\}} \mathcal{C}_{\overline{\mathfrak{g}}} \left( x_s(\sigma_\ell),\, y_e \right) \tag{4.96}$$

where $x_s(\sigma_\ell)$ is given by (4.88) – if $\mathfrak{g}$ is disconnected, then the sum in the right-hand-side above factorises in a product of sums.

## 4.3 Adjoint surface and zeroes of cosmological correlators

The knowledge of the compatibility conditions constrains the canonical form and in particular its numerator: the subspaces where the hyperplanes containing the facets intersect each other outside of the geometry are associated to vanishing multiple residues along such collections of hyperplanes. In the case of the usual cosmological polytopes, such constraints are enough to fix the numerator of the canonical form: as for any polytope, the definition of a cosmological polytope in terms of inequalities determines automatically which intersections are part of its boundary structure and which ones lie outside. From a physics perspective, such compatibility conditions are associated to the Steinmann-like relations [3, 29] which, despite their physical origin in cosmology is not understood yet, provide the usual flat-space Steinmann-relation that are an imprint of flat-space causality. In this section, we provide a first discussion of the locus of the zeroes of the cosmological correlators: understanding it can provide a great deal of physical information beyond the Steinmann-like relation, such that the existence of a hidden symmetry, the existence of degenerate vacua (identified by the Adler's zeroes [41]), or some novel feature. Also, at a mere computational level, a full understanding of the conditions on the numerator can provide a novel way to bootstrap the cosmological correlators even beyond the class of toy models which is directly associated to our geometrical description, in case such conditions had some degree of universality.

In the case of the weighted cosmological polytopes, we have already observed that the knowledge of the usual compatibility conditions is not enough to univocally fix the numerator. It is instructive to revise the simplest example of the weighted triangle already discussed at the very beginning of Section 4. It is characterised by three ordinary vertices – $\mathcal{Z}_1$, $\mathcal{Z}_2$, $\mathcal{Z}_3$ –, two internal vertices – which are alternatively indicated with $\mathcal{Z}_4$, $\mathcal{Z}_5$ or $\mathbf{x}_1$, $\mathbf{x}_2$, depending on notational convenience –, as well as three ordinary facets and one internal one. Its canonical form can be written as – see equations (4.1) and (4.9):

$$\omega \left( \mathcal{Y}, \mathcal{P}_\mathcal{G}^{(w)} \right) = \frac{\mathfrak{n}_1(\mathcal{Y}) \langle \mathcal{Y} d^2 \mathcal{Y} \rangle}{\langle \mathcal{Y} \mathbf{x}_1 2 \rangle \langle \mathcal{Y} 23 \rangle \langle \mathcal{Y} 3 \mathbf{x}_2 \rangle \langle \mathcal{Y} \mathbf{x}_2 \mathbf{x}_1 \rangle} = \frac{\langle \mathcal{Y} AB \rangle \langle \mathcal{Y} d^2 \mathcal{Y} \rangle}{\langle \mathcal{Y} \mathbf{x}_1 2 \rangle \langle \mathcal{Y} 23 \rangle \langle \mathcal{Y} 3 \mathbf{x}_2 \rangle \langle \mathcal{Y} \mathbf{x}_2 \mathbf{x}_1 \rangle} \tag{4.97}$$

where $A$ and $B$ in the very right-hand-side label the (for the moment arbitrary) points $\mathcal{Z}_A$ and $\mathcal{Z}_B$ respectively, with the line $\mathcal{C}_I := \epsilon_{IJK} \mathcal{Z}_A^J \mathcal{Z}_B^K$ identifying the numerator. For the usual cosmological polytope, the numerator is returned by the adjoint surface, *i.e.* the locus of the intersection of the hyperplanes containing the facets outside of the geometry. In the case of the weighted triangle, there is just one of such intersections, which fixes just one of the two points: $\mathcal{Z}_A^I := \epsilon^{IJK} \mathcal{W}_J^{(\mathcal{G})} \widetilde{\mathcal{W}}_K^{(\mathcal{G})}$ where $\mathcal{W}_I^{(\mathcal{G})} := \epsilon_{IJK} \mathcal{Z}_2^J \mathcal{Z}_3^K$ and $\widetilde{\mathcal{W}}_I^{(\mathcal{G})} := \epsilon_{IJK} \mathbf{x}_2^J \mathbf{x}_1^K$. It is useful to rewrite the canonical form (4.97) using the

local coordinates $\mathcal{Y} := (x_1, y, x_2)$:

$$\omega\left(\mathcal{Y}, \mathcal{P}_{\mathcal{G}}^{(w)}\right) = \frac{2\left[-\mathfrak{z}_2(x_1+x_2)+(\mathfrak{z}_3+\mathfrak{z}_1)y\right]}{(x_1+x_2)(x_1+y)(y+x_2)y}\,\frac{dx_1 \wedge dy \wedge dx_2}{\mathrm{Vol}\{GL(1)\}} \tag{4.98}$$

where $\mathcal{Z}_B^I := (\mathfrak{z}_1, \mathfrak{z}_2, \mathfrak{z}_3)$. Let us now use the information about the weight function and consider the weights of the highest codimension singularities – see equations (4.28) and (4.30):

$$\begin{aligned}
\mathrm{Res}_{\mathcal{W}^{(\mathfrak{g}_j)}}\mathrm{Res}_{\mathcal{W}^{(\mathcal{G})}}\left\{\omega\left(\mathcal{Y}, \mathcal{P}_{\mathcal{G}}^{(w)}\right)\right\} &= \mathfrak{z}_3+\mathfrak{z}_1 = w_{\mathcal{G}\mathfrak{g}_j} = 1, \\
\mathrm{Res}_{\mathcal{W}^{(\mathfrak{g}_j)}}\mathrm{Res}_{\widetilde{\mathcal{W}}^{(\mathcal{G})}}\left\{\omega\left(\mathcal{Y}, \mathcal{P}_{\mathcal{G}}^{(w)}\right)\right\} &= -2\mathfrak{z}_2 = \widetilde{w}_{\mathcal{G}\mathfrak{g}_j} = 2
\end{aligned} \tag{4.99}$$

where $\mathcal{W}_I^{(\mathfrak{g}_1)} := \epsilon_{\mathrm{IJK}}\mathcal{Z}_3^J\mathbf{x}_2^K$ and $\mathcal{W}_I^{\mathfrak{g}_2} := \epsilon_{\mathrm{IJK}}\mathcal{Z}_3^J\mathbf{x}_2^K$, and the sign due to the way that the boundaries are approached already taken into account. The numerical values on the very right-hand-side of (4.99) can be written if we assume the weight function of the weighted cosmological polytope to be as defined in (4.26). The knowledge of the weights turns out to provide our missing information.

Some comments are now in order. Let $\mathcal{P}_{\mathcal{G}}^{(w)}$ be a weighted cosmological polytope with boundaries identified by the lines $\mathcal{W}^{(\mathcal{G})}$, $\mathcal{W}^{(\mathfrak{g}_1)}$, $\mathcal{W}^{(\mathfrak{g}_2)}$, and $\widetilde{\mathcal{W}}^{(\mathcal{G})}$, and vertices given by $\{\mathcal{Z}_j, j=1,2,3\}\cup\{\mathbf{x}_1, \mathbf{x}_2\}$. First, note that the vector $\mathcal{Z}_B$ parametrises the weights: one obtains a 1-parameter family of canonical forms associated to the weighted cosmological polytope $\mathcal{P}_{\mathcal{G}}^{(w)}$, with the parameter being the relative weight:

$$\omega\left(\mathcal{Y}, \mathcal{P}_{\mathcal{G}}^{(w)}\right) = \frac{\widetilde{w}_{\mathcal{G}\mathfrak{g}_j}(x_1+x_2)+2w_{\mathcal{G}\mathfrak{g}_j}y}{(x_1+x_2)(x_1+y)(y+x_2)y}\,\frac{dx_1 \wedge dy \wedge dx_2}{\mathrm{Vol}\{GL(1)\}} \tag{4.100}$$

Secondly, the knowledge of the intersections among internal and ordinary boundaries – in this specific case just the point $\mathcal{Z}_A$ – organises which relative weights to consider. Third, it is possible to imagine variate the values of the weights – or equivalently of $\mathcal{Z}_B$. Note that taking either $\widetilde{w}_{\mathcal{G}\mathfrak{g}_j} = 0$, or $w_{\mathcal{G}\mathfrak{g}_j} = 0$, implies that both regions on the two sides of the boundary $\widetilde{\mathcal{W}}^{(\mathcal{G})}$, or $\mathcal{W}^{(\mathcal{G})}$, have the same weights: the weight function does not have a discontinuity and hence the intersection of the relevant hyperplane with the geometry does not constitutes a boundary. The canonical form of the weighted triangle reduces to the canonical form of the ordinary triangle $(\mathcal{Z}_1\mathcal{Z}_2\mathcal{Z}_3)$ – the cosmological polytope – and $(\mathcal{Z}_1\mathbf{x}_1\mathbf{x}_2)$, with the highest codimension singularities normalised to $w_{\mathcal{G}\mathfrak{g}_j}$ and $\widetilde{w}_{\mathcal{G}\mathfrak{g}_j}$ respectively. Another possible choice is to have the two weights equal: this is equivalent to consider the boundary $\mathcal{P}_{\mathcal{G}}^{(w)}\cap\widetilde{\mathcal{W}}^{(\mathcal{G})}$ as an external boundary, the point $\mathcal{Z}_B$ is moved to be coincident with $\mathcal{Z}_1$, and the weighted triangle reduce to the quadrilateral $(\mathbf{x}_1 2 3\mathbf{x}_2)$. The choice $\widetilde{w}_{\mathcal{G}\mathfrak{g}_j} = 2w_{\mathcal{G}\mathfrak{g}_j}$ leads instead to the original weighted cosmological polytope. Any other choice gives all the other possible weighted triangles which can be obtained by moving $\mathcal{Z}_B$ in $\mathbb{P}^2$.

Finally, it is reasonable to ask whether there is anything special in the relative weight which gives rise to the weighted cosmological polytope. Indeed, it is determined by the orientation-changing operation onto the cosmological polytope. However, is there anything special from a mere geometrical point of view? As already observed in Section 4, there are two choices for $\mathcal{Z}_B$ that return the relative weight of the weighted canonical form: choosing it as the intersection $\epsilon_{IJK}\mathcal{W}_J^{(\mathbf{x}_2)}\mathcal{W}_K^{(\mathfrak{g}_1)}$ or $\epsilon_{IJK}\mathcal{W}_J^{(\mathbf{x}_1)}\mathcal{W}_K^{(\mathfrak{g}_2)}$ between one of the other ordinary facet and a line passing though the special points $(\mathbf{x}_j, \mathbf{y})$: despite these last lines do not have a definite geometrical meaning (they neither constitute an actual boundary nor anything special happens along them), they can be singled out as determined by pairs of special points in the construction. However, there is a more precise way of phrasing this question. Let us

consider the graph associated to the weighted triangle and the vertex structure of the facet identified by either $\mathcal{W}^{(\mathfrak{g}_1)}$ or $\mathcal{W}^{(\mathfrak{g}_2)}$ – let us take $\mathcal{P}_{\mathcal{G}}^{(w)} \cap \mathcal{W}^{(\mathfrak{g}_1)}$ for definiteness:

$$\mathfrak{g}_1 \ \raisebox{-0.5ex}{\includegraphics[height=1em]{}} \qquad\qquad \mathfrak{g}_1 \ \raisebox{-0.5ex}{\includegraphics[height=1em]{}} \tag{4.101}$$

with the markings on the left/right graph showing the vertices of $\mathcal{P}_{\mathcal{G}}^{(w)}$ which are not/are on $\mathcal{P}_{\mathcal{G}}^{(w)} \cap \mathcal{W}^{(\mathfrak{g}_1)}$. It can be triangulated in two simplices – in this case two segments in $\mathbb{P}^1$ – such that each of them contains just one vertex associated to the graph edge:

$$\mathfrak{g}_1 \ \raisebox{-0.5ex}{\includegraphics[height=1em]{}} \ = \ \mathfrak{g}_1 \ \raisebox{-0.5ex}{\includegraphics[height=1em]{}} \ + \ \mathfrak{g}_1 \ \raisebox{-0.5ex}{\includegraphics[height=1em]{}} \tag{4.102}$$

with each term in the right-hand-side that can be seen as a product of a simplex associated to the vertex marked on the edge and a weighted polytope associated to $\overline{\mathfrak{g}}$ (in this case a single vertex). The two terms differ for the position of the marking on the edge: while in the context of the cosmological polytopes it identifies the direction of the energy flowing through the edge (see [42]), in this case, it is still related to the energy flux, but its direction does not change and the residue of the two terms is the same. Let us impose such condition on (4.97), or equivalently on (4.98), then

$$-\mathfrak{z}_2 \ = \ \mathfrak{z}_3 + \mathfrak{z}_1 \qquad \Longleftrightarrow \qquad \widetilde{w}_{\mathcal{G}\mathfrak{g}_1} \ = \ 2 w_{\mathcal{G}\mathfrak{g}_1}, \tag{4.103}$$

returning the canonical form for the weighted cosmological triangle – up to an overall number given by $\mathfrak{z}_3 + \mathfrak{z}_1$ whose choice, in any case, does not change the relative weights. As a final comment, the $1-1$ correspondence between graphs and weighted cosmological polytopes allows to identify such an extra constraint. Note further that the presence of the internal vertex is responsible for such a condition: were it to be absent, then the two residues would be the same up to a sign, returning the condition $\mathfrak{z}_2 = 0$ and the weighted cosmological polytope reduces to the cosmological polytope associated to the two-site line graph. Interestingly, the condition (4.103) can be equivalently obtained by requiring that the covariant restriction of the canonical form onto the line $\mathcal{H}$ – passing through the two special points $\mathbf{x}_1$ and $\mathbf{y}$ and identified by the co-vector $\mathcal{W}^{(\mathsf{x}_2)}$ – vanishes:

$$\begin{aligned}
\omega^{(1)}\left(\mathcal{Y}_{\mathcal{H}}, \mathcal{P}_{\mathcal{G}}^{(w)} \cap \mathcal{W}^{(\mathfrak{g}_1)} \cap \mathcal{H}\right) \ &:= \ \frac{1}{2\pi i} \oint_{\mathcal{H}} \frac{\omega\left(\mathcal{Y}, \mathcal{P}_{\mathcal{G}}^{(w)} \cap \mathcal{W}^{(\mathfrak{g}_1)}\right)}{\left(\mathcal{Y} \cdot \mathcal{W}^{(\mathsf{x}_2)}\right)} \ = \\
&= \ \frac{1}{2y}\left[\frac{1}{x_2 + y} + \frac{1}{x_2 - y}\right]\bigg|_{x_2 = 0} \frac{dy}{\text{Vol}\{GL(1)\}} \ = \ 0
\end{aligned} \tag{4.104}$$

where the first line represents the definition of covariant restriction onto $\mathcal{H}$ – for a more general discussion see [43] –, while the second line uses explicitly the triangulation (4.102) in the local coordinate patch associated to the weights of the graph.

Thus, the missing condition can be equivalently seen either as a condition on higher dimensional faces or in terms of novel zeroes – the equivalence can be readily seen from the second line in (4.104) which makes use of the fact that the residues of the two terms in the square brackets have the same residue.

Let us now see how this discussion generalises to arbitrary weighted cosmological polytopes.

**Weights and canonical forms** – First, note that each of weights $\widetilde{w}_{\mathcal{G}\mathfrak{g}_j}$ and $w_{\mathcal{W}\mathfrak{g}_j}$ is the product of the discontinuities along the two boundaries along which the residues in (4.99) are taken: if any of $\mathcal{P}_{\mathcal{G}}^{(w)} \cap \mathcal{W}^{(\mathfrak{g}_j)}$ would have been an internal boundary, then the second line in (4.99) would have returned $\widetilde{w}_{\mathcal{G}\mathfrak{g}_j} = 4$. Then considering the subdivision $\{\mathcal{P}_{\mathcal{G}_j}, \{\mathcal{G}_j\}, j = 1, \ldots, n_e\}$ determined by the hyperplanes $\left\{ \widetilde{\mathcal{W}}^{(\mathfrak{g}_e)}, e \in \mathcal{E} \right\}$, the canonical form can of an arbitrary weighted cosmological polytope can be written in terms of such subdivision and the weights associated to each element of the subdivision:

$$
\omega\left(\mathcal{Y}, \mathcal{P}_{\mathcal{G}}^{(w)}\right) = \frac{\sum\limits_{\{\mathcal{G}_j\}} \left[ w_{\mathcal{G}_j} \mathfrak{n}\left(\mathcal{Y}, \mathcal{P}_{\mathcal{G}_j}\right) \left( \prod\limits_{\mathfrak{g}\subseteq\mathcal{G}\backslash\mathcal{G}_j} q_{\mathfrak{g}}(\mathcal{Y}) \right) \left( \prod\limits_{e\in\mathcal{E}\backslash\mathcal{E}_j} \widetilde{q}_{\mathfrak{g}_e}(\mathcal{Y}) \right) \right]}{\left( \prod\limits_{\mathfrak{g}\subseteq\mathcal{G}} q_{\mathfrak{g}}(\mathcal{Y}) \right) \left( \prod\limits_{e\in\mathcal{E}} \widetilde{q}_{\mathfrak{g}_e}(\mathcal{Y}) \right)} \langle \mathcal{Y} d^{n_s+n_e-1} \mathcal{Y} \rangle \quad (4.105)
$$

where $w_{\mathcal{G}_j}$ and $\mathfrak{n}(\mathcal{Y}, \mathcal{P}_{\mathcal{G}_j})$ are respectively the weight and the canonical function numerator associated to a given element $\mathcal{P}_{\mathcal{G}_j}$ of the collection in the polytope subdivision of $\mathcal{P}_{\mathcal{G}}^{(w)}$, while $\{q_{\mathfrak{g}}(\mathcal{Y}) := \mathcal{Y} \cdot \mathcal{W}^{(\mathfrak{g})}, \mathfrak{g} \subseteq \mathcal{G}\}$ and $\{q_{\mathfrak{g}_e}(\mathcal{Y}) := \mathcal{Y} \cdot \widetilde{\mathcal{W}}^{(\mathfrak{g}_e)}, e \in \mathcal{E}\}$ are the set of ordinary and internal facets respectively.

Despite the expression (4.105) is obtained via algebraic manipulation onto the canonical form subdivision of $\mathcal{P}_{\mathcal{G}}^{(w)}$ in the original definition of the weighted cosmological polytope – and, hence, with precise values for $\{w_{\mathcal{G}_j}, \{\mathcal{G}_j\}\}$ –, it can be extended in two ways: First, by just allowing all the weights $w_{\mathcal{G}_j}$ to be arbitrary (this would be the general form obtained from the knowledge of the boundaries and their compatibility conditions); Secondly by considering arbitrary polytope subdivisions of $\mathcal{P}_{\mathcal{G}}^{(w)}$, whose element can be weighted polytopes rather than just ordinary polytopes, and with the $w_{\mathcal{G}_j}$ now being the weights associated to such elements.

**New zeroes** – There is a further comment to be made on the relation between the geometrical structures of polytopes and their adjoint surface and their relation to canonical forms, which helps to have a closer understanding of the weighted cosmological polytope ones. A codimension-$k$ boundary is defined as a non-empty codimension-$k$ intersection between the polytope and a codimension-$k$ hyperplane identified by $k$ hyperplanes containing the facets of the polytope. This corresponds to a logarithmic singularity in the associated canonical form. When such an intersection occurs outside of the polytope, it is not a codimension-$k$ boundary of the polytope and belongs to its adjoint surface. From the perspective of the associated canonical form, this means that its numerator develops a zero which prevents it from forming a logarithmic singularity along this boundary [38]. It can also happen that $k+1$ hyperplanes intersect each other in codimension $k$ on the polytope – one example is provided by the vertex of a pyramid in $\mathbb{P}^3$ opposite to its base, which is given by the intersection of four facets. However, such a boundary is anyhow identified by $k$ hyperplanes. This implies that such an intersection lives inside the adjoint surface in codimension-1. On such a boundary, the numerator of the canonical form develops a zero of multiplicity one, and the canonical form shows a logarithmic singularity – in the case just mentioned of a pyramid in $\mathbb{P}^3$ the adjoint surface passes through the vertex opposite to its base. This can be generalised to $k+m$ hyperplanes which intersect each other on the polytope in codimension $k$: this boundary lives in a codimension-$m$ subspace of the adjoint surface, and the canonical form develops a zero of multiplicity $m$. In the context of the weighted cosmological polytope such a phenomenon can also occur with intersections between the ordinary and

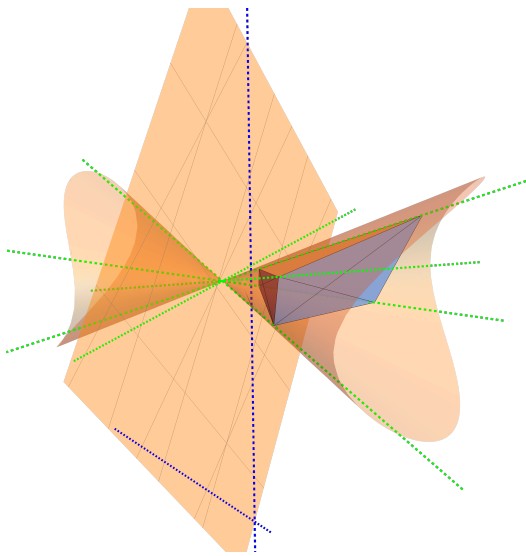

Figure 9: Representation of the locus of the zeroes of 2-sites 1-loop graph contribution to a cosmological correlator. This degenerate variety is given by a plane and a conic intersecting in the singular point of the conic. The green lines represent the zeros predicted by the facets intersections and the blue lines represent the novel conditions.

internal boundaries: this is precisely what we observed in Section 4.1 for the intersection between any codimension-$k$ face $\mathcal{P}_{\mathcal{G}}^{(w)} \cap \mathcal{W}^{(\mathfrak{g}_1 \cdots \mathfrak{g}_k)}$ such that the compatibilty condition (4.51) is satisfied with $\not{n} \neq 0$, and any of the hyperplanes $\widetilde{\mathcal{W}}^{(\mathfrak{g}_e)}$ associated to the edges counted by $\not{n}$ and their intersections – the simplext example is the weighted prism associated to a one-loop bubble graph, with such intersections involving all its vertices, and hence with the related adjoint surface passing through them – see Figure 9.

Let us now consider a facet $\mathcal{P}_{\mathcal{G}}^{(w)} \cap \mathcal{W}^{(\mathfrak{g})}$, such that $\mathcal{W}^{(\mathfrak{g})} \neq \mathcal{W}^{(\mathcal{G})}$, $\mathcal{W}^{(\mathcal{G}_e)}$, $\widetilde{\mathcal{W}}^{(\mathfrak{g}_e)}$, and $\mathcal{G}_e$ being the graph obtained from $\mathcal{G}$ by erasing the edge $e$. As discussed in Section 4.2, as any of such facets is approached, the canonical function factorises into the canonical function of the scattering facet $\mathcal{S}_{\mathfrak{g}}$ associated to $\mathfrak{g}$ and a lower-dimensional weighted polytope $\mathcal{P}_{\overline{\mathfrak{g}} \cup \not{\mathcal{E}}}^{(w)}$ associated to $\overline{\mathfrak{g}} \cup \not{\mathcal{E}}$

$$\Omega\left(\mathcal{Y}, \mathcal{P}_{\mathcal{G}}^{(w)} \cap \mathcal{W}^{(\mathfrak{g})}\right) = \Omega\left(\mathcal{Y}_{\mathfrak{g}}, \mathcal{S}_{\mathfrak{g}}\right) \times \Omega\left(\mathcal{Y}_{\overline{\mathfrak{g}} \cup \not{\mathcal{E}}}, \mathcal{P}_{\overline{\mathfrak{g}} \cup \not{\mathcal{E}}}^{(w)}\right) \qquad (4.106)$$

with $\mathcal{P}_{\overline{\mathfrak{g}} \cup \not{\mathcal{E}}}^{(w)}$ which enjoys the polytope subdivision (4.86) – we rewrite it here for convenience:

$$
\begin{aligned}
\Omega\left(\mathcal{Y}_{\overline{\mathfrak{g}} \cup \not{\mathcal{E}}}, \mathcal{P}_{\overline{\mathfrak{g}} \cup \not{\mathcal{E}}}^{(w)}\right) &= \sum_{\{\Sigma_{\not{\mathcal{E}}}\}} \Omega\left(\mathcal{Y}_{\not{\mathcal{E}}}, \Sigma_{\not{\mathcal{E}}}\right) \times \Omega\left(\mathcal{Y}_{\overline{\mathfrak{g}}}\left(\Sigma_{\not{\mathcal{E}}}\right), \mathcal{P}_{\overline{g}}^{(w)}\right) \\
&= \left(\prod_{\not{e} \in \not{\mathcal{E}}} \frac{1}{2 y_{\not{e}}}\right) \sum_{\{\sigma_{\not{e}} = \pm\}} \mathcal{C}_{\mathfrak{g}}\left(x_s(\sigma_{\not{e}}), y_e\right)
\end{aligned}
\qquad (4.107)
$$

where the last line has been obtained by considering the local coordinates made out of the collection of both site and edge weights, as well as the identification between the canonical function of $\mathcal{P}_{\overline{\mathfrak{g}}}^{(w)}$ and a linear combination of correlators associated to $\overline{\mathfrak{g}}$. Let $\{\mathcal{W}^{(\mathbf{x}_{s_j})}, j = 1, \ldots, n_s\}$ the collection

of codimension-1 hyperplanes identified by all the midpoints of the weighted triangles in the original construction but $\mathbf{x}_{s_j}$, $s_j$ being the $j$-th site of $\mathcal{G}$. Let $\mathcal{H} \equiv \mathcal{W}^{(\mathbf{x}_{\overline{s}_1} \cdots \mathbf{x}_{\overline{s}_{\overline{n}_s}})} := \bigcap_{j=1}^{\overline{n}_s} \mathcal{W}^{(\mathbf{x}_{\overline{s}_j})}$ with $\overline{n}_s$ being the number of sites $\{\overline{s}_j, \, j = 1, \ldots, \overline{n}_s\}$ in $\overline{\mathfrak{g}}$. Considering $\overline{\mathfrak{g}}$ to be a tree graph[17], then, the covariant restriction of the canonical form of this facet onto $\mathcal{W}^{(\mathbf{x}_{\overline{s}_1} \cdots \mathbf{x}_{\overline{s}_{\overline{n}_s}})}$

$$\omega^{(\overline{n}_s)}\left(\mathcal{Y}_{\mathcal{H}}, \, \mathcal{P}_{\mathcal{G}}^{(w)} \cap \mathcal{W}^{(\mathfrak{g})} \cap \mathcal{H}\right) \,=\, \frac{1}{(2\pi i)^{\overline{n}_s}} \oint_{\mathcal{H}} \frac{\omega\left(\mathcal{Y}, \, \mathcal{P}_{\mathcal{G}}^{(w)} \cap \mathcal{W}^{(\mathfrak{g})}\right)}{\prod_{j=1}^{\overline{n}_s}\left(\mathcal{Y} \cdot \mathcal{W}^{(\mathbf{x}_{\overline{s}_j})}\right)} \tag{4.108}$$

vanishes [18] – the upper index in the left-hand-side indicates the degree in which the form would scale under a $GL(1)$ transformation, were it not to be zero. First, because of the definition of the hyperplane $\mathcal{H}$, the covariant restriction operation affects only the canonical form associated to $\mathcal{P}_{\overline{\mathfrak{g}} \cup \mathscr{E}}^{(w)}$. In terms of canonical functions and using (4.107):

$$\Omega^{(\overline{n}_s)}\left(\mathcal{Y}_{\mathcal{H}}, \, \mathcal{P}_{\mathcal{G}}^{(w)} \cap \mathcal{W}^{(\mathfrak{g})} \cap \mathcal{H}\right) \,=\, \Omega\left(\mathcal{Y}_{\mathfrak{g}}, \, \mathcal{S}_{\mathfrak{g}}\right) \times \frac{1}{(2\pi i)^{\overline{n}_s}} \oint_{\mathcal{H}} \frac{\Omega\left(\mathcal{Y}_{\overline{\mathfrak{g}} \cup \mathscr{E}} \, \mathcal{P}_{\overline{\mathfrak{g}} \cup \mathscr{E}}^{(w)}\right)}{\prod_{j=1}^{\overline{n}_s}\left(\mathcal{Y} \cdot \mathcal{W}^{(\mathbf{x}_{\overline{s}_j})}\right)} \tag{4.109}$$

Secondly, it is useful to consider the polytope subdivision (4.107) for $\mathcal{P}_{\overline{\mathfrak{g}} \cup \mathscr{E}}^{(w)}$. It is characterised by having $2^{n_{\mathscr{E}}}$ terms and uses the collection of hyperplanes $\{\widetilde{\mathcal{W}}^{(\mathfrak{g}_{\mathscr{E}})}, \mathscr{E} \in \mathscr{E}\}$ and the associated canonical function can be expressed as

$$\Omega\left(\mathcal{Y}_{\overline{\mathfrak{g}} \cup \mathscr{E}}, \, \mathcal{P}_{\overline{\mathfrak{g}} \cup \mathscr{E}}^{(w)}\right) \,=\, \Omega\left(\mathcal{Y}_{\mathscr{E}}, \, \Sigma_{\mathscr{E}}\right) \times \sum_{\{\sigma_{\mathscr{E}} = \pm\}} \Omega\left(\mathcal{Y}_{\overline{\mathfrak{g}}}(\sigma_{\mathscr{E}}), \, \mathcal{P}_{\overline{\mathfrak{g}}}^{(w)}\right) \tag{4.110}$$

where, as usual, $\Sigma_{\mathscr{E}}$ is the simplex associated to the set $\mathscr{E}$ of edges connecting $\mathfrak{g}$ and $\overline{\mathfrak{g}}$, $\sigma_{\mathscr{E}} = \pm$ is the sign associated to the vertex marked on $\mathscr{E} \in \mathscr{E}$ taken in the element of the subdivision and such that in the local patch of the graph weights, the graph weights of $\overline{\mathfrak{g}}$ are given by (4.88). The canonical function subdivision (4.110) makes also manifest that the covariant restriction (4.109) over $\mathcal{H}$ affects only the canonical functions $\Omega(\mathcal{Y}_{\overline{\mathfrak{g}}}(\sigma_{\mathscr{E}}), \, \mathcal{P}_{\overline{\mathfrak{g}}}^{(w)})$ appearing in the sum. Finally, each of such canonical functions can be represented via a triangulation of $\mathcal{P}_{\overline{\mathfrak{g}}}^{(w)}$ in terms of the hyperplanes associated to the subgraphs of $\overline{g}$ containing at least one of the sites from where the edges in $\mathscr{E}$ depart:

$$\Omega\left(\mathcal{Y}_{\overline{\mathfrak{g}} \cup \mathscr{E}}, \, \mathcal{P}_{\overline{\mathfrak{g}} \cup \mathscr{E}}^{(w)}\right) \,=\, \Omega\left(\mathcal{Y}_{\mathscr{E}}, \, \Sigma_{\mathscr{E}}\right) \times \sum_{\{\sigma_{\mathscr{E}} = \pm\}} \sum_{\mathfrak{g}' \in \overline{\mathfrak{G}}_{\mathscr{E}}} \frac{\Omega\left(\mathcal{Y}_{\overline{\mathfrak{g}}}(\sigma_{\mathscr{E}}), \, \mathcal{P}_{\overline{\mathfrak{g}}}^{(w)} \cap \mathcal{W}^{(\mathfrak{g}')}\right)}{\left(\mathcal{Y}_{\overline{\mathfrak{g}}}(\sigma_{\mathscr{E}}) \cdot \mathcal{W}^{(\mathfrak{g}')}\right)} \tag{4.111}$$

where $\overline{\mathfrak{G}}_{\mathscr{E}}$ is the set of subgraphs of $\overline{\mathfrak{g}}$ such that at least one of the sites where the edges in $\mathscr{E}$ depart from. Such triangulation can be obtained directly on the canonical function via a contour integral. Let $\overline{\mathcal{V}}_{\mathscr{E}}$ be the set of sites which are endpoints of the edges in $\mathscr{E}$. Let us also consider the local patch in which the weights associated to the sites of $\overline{\mathfrak{g}}$ are given by (4.88). Then, taking the one-parameter deformation

$$x_{\mathscr{S}} \,\longrightarrow\, x_{\mathscr{S}} + z, \qquad \mathscr{S} \in \overline{\mathcal{V}}_{\mathscr{E}}, \tag{4.112}$$

[17]Let us emphasise that $\mathcal{G}$ can be a graph with arbitrary topology, but $\mathfrak{g}$ is taken in such a way $\overline{\mathfrak{g}}$ is a tree graph.

[18]Equation (4.108) provides the definition of a covariant restriction, which suffices the purpose of the present work. For a more general discussion, see [43].

where $\not{s}$ identifies one of the sites endpoints of any of the edges in $\mathcal{E}$, the triangulation for $\mathcal{P}_{\overline{\mathfrak{g}}}^{(w)}$ in terms of the canonical function in (4.111) from the Cauchy theorem:

$$0 \; = \; \frac{1}{2\pi i} \oint_{\hat{\mathcal{C}}} \frac{dz}{z} \, \Omega\left(\mathcal{Y}_{\overline{\mathfrak{g}}}(\sigma_{\not{e}}), \mathcal{P}_{\overline{\mathfrak{g}}}^{(w)}; z\right) \; = \; \Omega\left(\mathcal{Y}_{\overline{\mathfrak{g}}}(\sigma_{\not{e}}), \mathcal{P}_{\overline{\mathfrak{g}}}^{(w)}\right) - \sum_{\mathfrak{g}' \in \overline{\mathfrak{G}}_{\mathcal{E}}} \frac{\Omega\left(\mathcal{Y}_{\overline{\mathfrak{g}}}(\sigma_{\not{e}}), \mathcal{P}_{\overline{\mathfrak{g}}}^{(w)} \cap \mathcal{W}^{(\mathfrak{g}')}\right)}{\left(\mathcal{Y}_{\overline{\mathfrak{g}}}(\sigma_{\not{e}}) \cdot \mathcal{W}^{(\mathfrak{g}')}\right)} \quad (4.113)$$

One of the elements of $\overline{\mathfrak{G}}_{\mathcal{E}}$ is $\overline{\mathfrak{g}}$ and boundary $\mathcal{P}_{\overline{\mathfrak{g}}}^{(w)} \cap \mathcal{W}^{(\overline{\mathfrak{g}})}$ is the scatteing facet $\mathcal{S}_{\overline{\mathfrak{g}}}$ associated to $\overline{\mathfrak{g}}$. As $\overline{\mathfrak{g}}$ is assumed to be a tree graph, $\mathcal{S}_{\overline{\mathfrak{g}}}$ is a simplex. In the chosen local patch, the term in (4.111) corresponding to $\mathfrak{g}' = \overline{\mathfrak{g}}$ can be written as [19]

$$\sum_{\{\sigma_{\not{e}}=\pm 1\}} \frac{\Omega\left(\mathcal{Y}_{\overline{\mathfrak{g}}}(\sigma_{\not{e}}), \mathcal{P}_{\overline{\mathfrak{g}}}^{(w)} \cap \mathcal{W}^{(\overline{\mathfrak{g}})}\right)}{\left(\mathcal{Y}_{\overline{\mathfrak{g}}}(\sigma_{\not{e}}) \cdot \mathcal{W}^{(\overline{\mathfrak{g}})}\right)} \; = \; \sum_{\{\sigma_{\not{e}}=\pm 1\}} \frac{1}{\sum\limits_{s\in\mathcal{V}_{\overline{\mathfrak{g}}}} x_s(\sigma_{\not{e}})} \prod_{e\in\mathcal{E}_{\overline{\mathfrak{g}}}} \frac{1}{y_e^2 - \left(\sum\limits_{s\in\mathcal{V}_{\overline{\mathfrak{g}}}^{(e)}} x_s(\sigma_{\not{e}})\right)^2} \quad (4.114)$$

where $\mathcal{V}_{\overline{\mathfrak{g}}}^{(e)}$ is the set of sites of $\overline{\mathfrak{g}}$ on the side of the edge $e$ which does not cointain $\not{s}$. It is easy to see that the right-hand side of (4.114) vanishes as the terms in the sum cancel in pairs upon the covariant restriction on $\mathcal{H}$. On $\mathcal{H}$, all the possible combinations of the signs $\{\sigma_{\not{e}} = \pm 1\}$ can be divided in pairs such that, for each pair, the prefactor $\sum_{s\in\mathcal{V}_{\overline{\mathfrak{g}}}} x_s(\sigma_{\not{e}})$ is the same up to a sign. For the elements of each pair also the canonical form of the scattering facet is the same in the chosen local patch. Hence, the sum over $\{\sigma_{\not{e}} = \pm 1, \; \forall \; \not{e} \in \mathcal{E}\}$ vanishes.

Let us now consider the contribution to (4.113) coming from a graph $\mathfrak{g}' \in \overline{\mathfrak{G}}_{\mathcal{E}}$ such that its complementary graph contains just a site. If the weight of this site does not depend on any of the $\sigma_{\not{e}}$'s, then this contribution cancels individually upon the covariant restriction on $\mathcal{H}$ as shown in (4.104). Were it to be dependent on any of the $\sigma_{\not{e}}$'s, then it can be shown that the contribution from this class of subgraphs vanishes upon summation over the sums in a similar fashion as described above for $\mathfrak{g}' = \overline{\mathfrak{g}}$.

The contributions in (4.111) from the other elements of $\overline{\mathfrak{G}}_{\mathcal{E}}$ can be shown to vanish as all the canonical functions $\Omega\left(\mathcal{Y}_{\overline{\mathfrak{g}}}(\sigma_{\not{e}}), \mathcal{P}_{\overline{\mathfrak{g}}}^{(w)} \cap \mathcal{W}^{(\mathfrak{g}')}\right)$ factorises as in (4.85) into a scattering facet and a term which has the same structure of (4.110), but in lower dimensions. Each of these terms can be put into the form (4.111). This procedure can be iterated, reducing the analysis to sets of lower dimensional scattering facets and factorised canonical functions with factors corresponding to subgraphs with a single site.

As a final comment, whether we were to consider a covariant restriction onto a hyperplane defined as the intersection of $k < \overline{n}_s$ hyperplanes identified by the midpoints of the weighted triangles in the original construction, then it will always be non-zero. A proof of this statement is provided in Appendix A.2.

*Example* – Let us consider a graph $\mathcal{G}$ with at least one tree substructure and let us consider the facet

---

[19]The form of the individual terms in the product (4.114) is just the Lorentz invariant propagator associated to the edge $e$ in energy space, and the product returns the flat-space scattering amplitudes at tree-level [42].

$\mathcal{P}_{\mathcal{G}} \cap \mathcal{W}^{(\mathfrak{g})}$ such that $\overline{\mathfrak{g}}$ is a tree-subgraph

$$(4.115)$$

As shown earlier, the canonical function of this facet factorises into the canonical function of the scattering facet associated to $\mathfrak{g}$ and the canonical function of $\mathcal{P}^{(w)}_{\overline{\mathfrak{g}} \cup \mathscr{E}}$, with $\mathscr{E}$ which is constituted by a single edge with weight $y_{34}$. The polytope subdivision (4.110) for $\mathcal{P}^{(w)}_{\overline{\mathfrak{g}} \cup \mathscr{E}}$ can be graphically represente $\mathcal{P}^{(w)}_{\overline{\mathfrak{g}} \cup \mathscr{E}}$ as

$$\Omega\left(\mathcal{Y}_{\overline{\mathfrak{g}} \cup \mathscr{E}}, \mathcal{P}^{(w)}_{\overline{\mathfrak{g}} \cup \mathscr{E}}\right) = \frac{1}{2y_{34}}\left[\begin{array}{c} \overset{y_{12}}{\bullet} \overset{y_{23}}{\bullet} \\ x_1 \quad x_2 \quad x_3 - y_{34} \end{array} + \begin{array}{c} \overset{y_{12}}{\bullet} \overset{y_{23}}{\bullet} \\ x_1 \quad x_2 \quad x_3 + y_{34} \end{array}\right]$$

$$(4.116)$$

For each of these two terms, let us use the subdivision obtained via (4.113) via the one-parameter deformation $x_3 \longrightarrow x_3 + z$:

$$\begin{array}{c} \overset{y_{12}}{\bullet} \overset{y_{23}}{\bullet} \\ x_1 \quad x_2 \quad x_3 \mp y_{34} \end{array} = \frac{\begin{array}{c} \overset{y_{12}}{} \overset{y_{23}}{} \\ x_1 \quad x_2 \quad x_3 \mp y_{34} \end{array}}{x_1 + x_2 + x_3 \mp y_{34}} + \frac{\begin{array}{c} \overset{y_{12}}{} \overset{y_{23}}{} \\ x_1 \quad x_2 \quad x_3 \mp y_{34} \end{array}}{y_{12} + x_2 + x_3 \mp y_{34}}$$

$$+ \frac{\begin{array}{c} \overset{y_{12}}{} \overset{y_{23}}{} \\ x_1 \quad x_2 \quad x_3 \mp y_{34} \end{array}}{y_{23} + x_3 \mp y_{34}}$$

$$(4.117)$$

with the elements of the subdivision identified by the facets associated to the subgraphs encircled by the red dashed lines.

Upon the restriction onto the hyperplane $\mathcal{H} := \mathcal{W}^{(x_1 x_2 x_3)}$, the sum of the first term in (4.117) for both contributions in (4.116) yields straightforwardly zero as they differ just by a sign. The second class of terms in (4.117) factorises into the canonical function of the lower-dimensional scattering facet associated to the subgraph identified by the dashed line and a canonical function whose covariant restriction on $\mathcal{H}$ has the same structure as in (4.117), vanishing individually. Finally, the class of terms in the second line of (4.117) also factorises. Let $\mathfrak{g}_3$ be the subgraph identified by the red-line encircling the site with weight $x_3 \mp y_{34}$. Then, the canonical function of $\mathcal{P}^{(w)}_{\overline{\mathfrak{g}}_e \cup \mathscr{E}_3}$ – where $\mathscr{E}_3 := \{e_{23}\}$ is the edge connecting $\mathfrak{g}_3$ and $\overline{\mathfrak{g}}_3$– can be expressed according to the same class of polytope subdivision as (4.116):

$$\begin{array}{c} \overset{y_{12}}{} \overset{y_{23}}{} \\ x_1 \quad x_2 \quad x_3 \mp y_{34} \end{array} = \begin{array}{c} \overset{y_{12}}{} \overset{y_{23}}{} \\ x_1 \quad x_2 \quad x_3 \mp y_{34} \end{array} + \begin{array}{c} \overset{y_{12}}{} \overset{y_{23}}{} \\ x_1 \quad x_2 \quad x_3 \mp y_{34} \end{array} =$$

$$= \frac{1}{2y_{23}}\left[\begin{array}{c} \overset{y_{12}}{} \\ x_1 \quad x_2 + y_{23} \end{array} + \begin{array}{c} \overset{y_{12}}{} \\ x_1 \quad x_2 - y_{23} \end{array}\right]$$

$$(4.118)$$

As discussed above, iterating the procedure we can express the canonical functions associated to the graphs in the second line of (4.118) in a representation such as the one in (4.117) via the shift

$x_2 \longrightarrow x_2 + z$:

$$\underset{x_1 \qquad\quad x_2 \mp y_{23}}{\bullet\!\!-\!\!\overset{y_{12}}{-\!\!-\!\!-}\!\!-\!\!\bullet} \;=\; \frac{\overset{y_{12}}{\bullet\!\!-\!\!-\!\!-\!\!-\!\!\bullet}}{x_1 + x_2 \mp y_{23}} \;+\; \frac{\overset{y_{12}}{\bullet\!\!-\!\!-\!\!-\!\!-\!\!\bullet}}{y_{12} + x_2 \mp y_{23}}$$

(4.119)

Upon the covariant restriction on $\mathcal{H}$, the second term in (4.119) is zero, while the first one sums to zero in (4.118).

Interestingly, it turns out that these vanishing covariant restrictions correspond to empty intersections and vice-versa: then, any covariant restriction of the canonical form of a face $\mathcal{P}_{\mathcal{G}}^{(w)} \cap \mathcal{W}^{(\mathfrak{g}_1 \cdot \mathfrak{g}_k)}$ on a hyperplane, is equal to zero if and only if the hyperplane does not intersect the face in its interior.

**Fixing the adjoint surface** – The novel class of vanishing conditions highlighted above holds whenever one starts with a facet $\mathcal{P}_{\mathcal{G}}^{(w)} \cap \mathcal{W}^{(\mathfrak{g})}$ associated to a subgraph $\mathfrak{g} \subset \mathcal{G}$ such that $\overline{\mathfrak{g}}$ is a tree graph. Furthermore, it is straightforward to check that complementing the usual compatibility conditions with these ones, is not enough the fix the adjoint surface and hence to determine the numerator of the canonical form.

In the discussion about the weighted triangle at the beginning of this section, the question about the extra condition to fix its adjoint surface – and hence its weight function – was sharpened by requiring that the residues of the two terms in which the facet can be triangulated by considering one vertex associated to $\mathcal{E}$ at a time, is the same. In that case, we showed that this condition is equivalent to the vanishing conditions we proved in this previous paragraph. However, this equivalence turns out not to be true in general, with the former containing the latter but being more general.

Let us consider a weighted cosmological polytope $\mathcal{P}_{\mathcal{G}}^{(w)}$ associated to an arbitrary graph $\mathcal{G}$. Let $\mathfrak{g} \subset \mathcal{G}$ be one of its subgraphs, then its facet $\mathcal{P}_{\mathcal{G}}^{(w)} \cap \mathcal{W}^{(\mathfrak{g})}$ can be expressed in terms of the polytope subdivision such that just one vertex associated to each of the edges $\mathcal{E}$ connecting $\mathfrak{g}$ and $\overline{\mathfrak{g}}$ belongs to a given element of the subdivision:

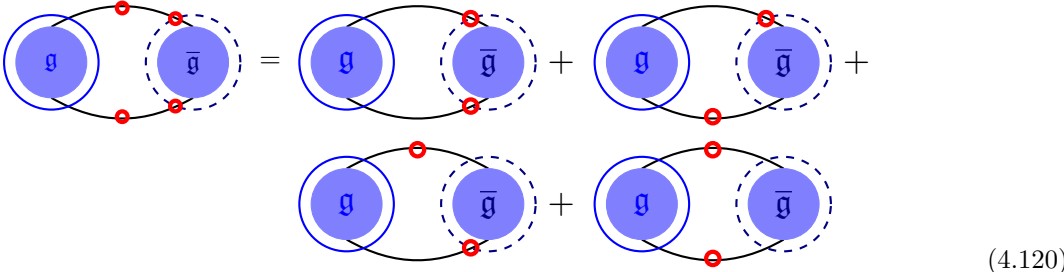

(4.120)

with the canonical function of this facet factorising as in (4.85) and the canonical function of $\mathcal{P}_{\overline{\mathfrak{g}} \cup \mathcal{E}}^{(w)}$ which is subdivided according to (4.110). Then, considering the codimension-$(n_{\mathcal{E}} + 1)$ face obtained as the intersection between the facet $\mathcal{P}_{\mathcal{G}}^{(w)} \cap \mathcal{W}^{(\mathfrak{g})}$ with all the hyperplanes $\{\widetilde{\mathcal{W}}^{(\mathfrak{g}_e)}, \, e \in \mathcal{E}\}$, then the resulting canonical form of the terms in the right-hand-side of (4.120) ought to be the same. This is straightforward to understand as, in terms of graph weights, the subgraphs $\overline{\mathfrak{g}}$ in the subdivision elements differ from each other for site weights, which are given by (4.88) and we rewrite here for

convenience:

$$x_s(\sigma_{\not e}) := x_s + \sum_{\not e \in \mathscr{G} \cap \mathcal{E}_s} \sigma_{\not e} y_{\not e} \qquad (4.121)$$

$\mathcal{E}_s$ being, as before, the set of edges departing from the site $s$. Hence, when the codimension-$(n_{\mathscr{G}} + 1)$ face is considered, the elements of the subdivision are all associated to the same subgraph with the unshifted weights for the sites which are endpoints of the edges in $\mathscr{G}$. The condition can thus be written as

$$\mathrm{Res}_{\widetilde{\mathcal{W}}^{(\mathfrak{g}_{\not e_1})}} \cdots \mathrm{Res}_{\widetilde{\mathcal{W}}^{(\mathfrak{g}_{\not e_{n_{\mathscr{G}}}})}} \Omega\left(\mathcal{Y}_{\mathscr{G}}, \Sigma_{\mathscr{G}}\right) \times \Omega\left(\mathcal{Y}_{\overline{\mathfrak{g}}}\left(\sigma_{\not e}\right), \mathcal{P}_{\overline{\mathfrak{g}}}^{(w)}\right) = \Omega\left(\mathcal{Y}_{\overline{\mathfrak{g}}}, \mathcal{P}_{\overline{\mathfrak{g}}}^{(w)}\right) \qquad (4.122)$$

for all choices of signs $\sigma_{\not e}$ – $i.e.$ the multiple residues on the left-hand-side are equal for all the inequivalent choices of $\sigma_{\not e}$'s.

Despite this condition can look trivial, the compatibility conditions leave unfixed a number of parameters, which are associated to the weight function of $\mathcal{P}_{\mathcal{G}}^{(w)}$, and the condition (4.122) is not in general satisfied unless such parameters acquires a specific value.

Summarising, the knowledge of both internal and ordinary boundaries as well as of their compatibility conditions, define a multi-parameter family of adjoint surfaces, whose parameters are related to the weight function. In the specific case of the weighted cosmological polytopes, by construction, the constraints (4.122) allow to fix such parameters, with some of these constraints that can be phrased as vanishing conditions when some faces are covariantly-restricted on special hyperplanes. Note that setting the right-hand-side of (4.122) to zero returns the usual cosmological polytopes as it becomes equivalent to requiring the absence of internal boundaries.

## 5 Conclusion and outlook

Cosmological correlators are our window to the early universe physics and also represent the initial conditions for the evolution which led to the late-time structure we can observe. Hence, understanding their mathematical structure, and how fundamental physics is encoded into it as well as developing novel computational tools become of primary importance. Despite in recent years several advances have been made via several techniques [44–53], our knowledge remains still primitive.

In this paper, we offer a different point of view, providing a first principle definition of the cosmological correlators in terms of novel geometrical objects we named weighted cosmological polytopes. They constitute the first example in physics of a weighted positive geometry with codimension-1 internal boundaries. This picture allowed us to obtain both novel representations of the cosmological correlators, as well as general information about their properties as individual as well as sequential discontinuities are approached. Our work represents a step forward in understanding the perturbative structure of the cosmological correlators. In what follows, we summarise the main future directions, organised in a similar fashion as our results in the introduction.

**Properties of cosmological observables from the wavefunctional** – The relation between cosmological polytopes and weighted cosmological polytopes via a simple orientation-changing operation, allows to sharpen the connection between the properties of the Bunch-Davies wavefunctional and the ones of the cosmological correlators, and how the latter is related to the former. In particular, we were

able to show how both quantities enjoy the very same Steinmann-like relations and how a generalisation of the Feynmann-tree theorem for the correlators is a direct consequence of the validity of the same theorem for the wavefunction coefficients. It will be interesting to further deepen this connection as well as broadening this question to more general observables. The Bunch-Davies wavefuctional provides the probability distributions for the state configurations at future infinity via its modulus square. As any equal-time observable can be computed as an integral over such a distribution, some of its properties should be inherited from the one of the wavefunctional. A natural observable to look at is the mean square displacement distribution in state space, which allows to quantify the dissimilarity between state configurations at future infinity [54]. Such a quantity shows a non-trivial behaviour even for a free massless scalar, with more dissimilar configurations which appear to be more likely [54–56]. This behaviour turns out to be a direct consequence of the behaviour at future infinity of the real part of the free two-point wavefunction. It would be interesting to study whether in the class of toy models captured by our construction, the relation between the wavefunction and this observable can be sharpened via a specific operation as it happened for cosmological correlators in this paper, as well as how the two operations can be related to each other.

**Pushing this first principle formulation for cosmological correlators** – The weighted cosmological polytope does not only provide a novel way of computing cosmological correlators perturbatively and studying their structure: the fact that they have an intrinsic, first principle, definition which does not rely on any physics notion, also provides a new set of mathematical rules for the cosmological correlators. It would be interesting to explore all the possible operations which can be defined on a weighted cosmological polytope to extract information on the cosmological correlators beyond individual graphs and beyond the perturbative regime in which the equivalence between geometry and physics has been originally studied.

**Novel vanishing conditions** – One of the most interesting features that our analysis uncovers, is the existence of novel vanishing conditions for the cosmological correlators. They allow to fix the cosmological correlators (completely for trees, partially for loops). If in some cases they can be interpreted as sequential operations between a residue and soft limits, it is fair to admit that currently, we do not have a satisfactory physical understanding: our analysis shares some light on the structure of the zeroes of cosmological correlators – which are encoded in the adjoint surface, and is related to the compatibility conditions as well as to the weight function – and it would be interesting to understand whether they can be related to the vacuum structure or to any other physical property. The weighted cosmological polytopes are directly related to a universal integrand for the cosmological correlators. It would be interesting to check how the properties of the zeroes map upon integration to the integrated correlation function.

**The mathematical side: weighted polytopes** – As just mentioned, the zeroes of the cosmological correlators are encoded into the adjoint surface of the weighted cosmological polytope. However, in this case, the compatibility conditions among the facets determine a multi-parameter family of adjoint surfaces, with the un-fixed parameters directly related to the weight function. Said differently, there

is a full class of weighted polytopes with the same ordinary and internal boundaries. We identified the specific conditions that fix the adjoint surface to be the one of the weighted cosmological polytope and how to modify it to extract the cosmological polytope. It would be interesting to explore whether there are further, geometrically motivated, conditions that can be defined. Finally, the weighted cosmological polytopes is a (simpler) example of weighted positive geometries, which were discovered first in the context of the amplituhedron at 2-loops [30]. Our in-depth analysis was made possible thanks to the $1 - 1$ correspondence with graphs and the relation between boundary structure and markings. However, these tools are not available for a general weighted positive geometry. It would be interesting to check whether any of the properties we investigated can be generalised and can be considered not specific to our case.

## Acknowledgements

It is a pleasure to thank Nima Arkani-Hamed and Paul Heslop for valuable discussions. P.B. would like to thank the organisers of the *All Lambdas Holography Workshop 2023* as well as of the workshop *24 hours of Combinatorial Synergies* for the possibility of presenting results related to this work. P.B. would also like to thank the developers of SageMath [57], Maxima [58], Polymake [59–62], TOPCOM [63], and Tikz [64]. P.B. has been partially supported during the first part of this work by the European Research Council under the European Union's Horizon 2020 research and innovation programme (No 725110), and would like to thank Dieter Lüst and Gia Dvali for making finishing it possible. G.D. acknowledges support from the Deutsche Forschungsgemeinschaft (DFG) under Germany's Excellence Strategy – EXC 2121 "Quantum Universe" – 390833306.

## A    A note on the adjoint surface of the weighted cosmological polytopes

In this section, we discuss some mathematical properties of the weighted cosmological polytopes introduced in this paper. In particular, we present some features of the adjoint surface whose structure, together with the internal boundaries, is the main novelty of these geometries.

### A.1    Positivity of the adjoint surface

Before discussing the adjoint surface of the weighted cosmological polytopes, it is useful to recall some properties of more familiar objects such as the convex polytopes. A convex polytope has an adjoint surface that never intersects it in its interior. From the perspective of the associated canonical form, this implies that it is always positive in the interior of the convex polytope. Let $\mathcal{P}$ be a convex projective polytope living in $\mathbb{P}^{N-1}$ defined by its facets that are identified by the set of co-vectors $\{\mathcal{W}_I^{(j)},\, j = 1, \ldots, \tilde{\nu} \,|\, \tilde{\nu} \geq N\}$, so that $\{\mathcal{Y} \cdot \mathcal{W}^{(j)} > 0,\, j = 1, \ldots, \tilde{\nu}\}$ for any point $\mathcal{Y} \in \mathbb{P}^{N-1}$ inside $\mathcal{P}$. Then, the dual polytope $\mathcal{P}^*$ of $\mathcal{P}$ is defined as the convex polytope living in the linear dual space of $\mathbb{P}^{N-1}$, which is still denoted by $\mathbb{P}^{N-1}$, such that

$$\mathcal{P}^* := \left\{ \mathcal{W}^* \in \mathbb{P}^{N-1} \,|\, \mathcal{W}^* \cdot \mathcal{Y} > 0,\, \forall\, \mathcal{Y} \in \mathcal{P} \right\}. \tag{A.1}$$

Equivalently, it can be defined as the convex hull of the vertices identified by the co-vectors $\{\mathcal{W}_I^{(j)},\, j = 1,\ldots,\tilde{\nu}\,|\,\tilde{\nu} \geq N\}$

$$\mathcal{P}^* := \left\{ \sum_{j=1}^{\tilde{\nu}} C_j \mathcal{W}^{(j)} \in \mathbb{P}^{N-1} \,\middle|\, C_j > 0,\, j = 1,\ldots,\tilde{\nu} \right\} . \tag{A.2}$$

Interestingly, the canonical function of a convex polytope is given by the volume of its dual and it has the following integral representation [18]:

$$\Omega(\mathcal{Y},\mathcal{P}) = \mathrm{Vol}_{\mathcal{Y}}(\mathcal{P}^*) := \int_{\mathcal{W}^* \in \mathcal{P}^*} \frac{\langle \mathcal{W}^* d^{N-1} \mathcal{W}^* \rangle}{(\mathcal{Y} \cdot \mathcal{W}^*)^N} , \tag{A.3}$$

where the orientation of $\mathcal{P}^*$ is given by $\mathrm{sign}\left(\prod_{j=1}^{\tilde{\nu}} \mathcal{Y} \cdot \mathcal{W}^{(j)}\right)$. The first important observation is that $\mathrm{Vol}_{\mathcal{Y}}(\mathcal{P}^*)$ is manifestly positive for $\mathcal{Y} \in \mathcal{P}$.

Let us now consider a signed subdivision $\mathcal{P}$ by a set of convex polytopes $\{\mathcal{P}^{(j)},\, j = 1,\ldots,r\}$. As the dual polytope $\mathcal{P}^*$ is signed-subdivided by the set $\{\mathcal{P}^{*(j)},\, j = 1,\ldots,r\}$, $\mathcal{P}^{*(j)}$ being the dual polytope of $\mathcal{P}^{(j)}$, then the canonical function $\Omega(\mathcal{Y},\mathcal{P})$ can be expressed as a sum over the canonical functions $\Omega(\mathcal{Y},\mathcal{P}^{(j)})$, the volume $\mathrm{Vol}_{\mathcal{Y}}(\mathcal{P}^*)$ as a sum over $\mathrm{Vol}_{\mathcal{Y}}(\mathcal{P}^{*(j)})$, with $\Omega(\mathcal{Y},\mathcal{P}^{(j)}) = \mathrm{Vol}_{\mathcal{Y}}(\mathcal{P}^{*(j)})$. follows from (A.1) that

$$\mathcal{P}^{(j)} \subset \mathcal{P} \quad \Rightarrow \quad \mathcal{P}^* \subset \mathcal{P}^{*(j)} . \tag{A.4}$$

This implies that the sign of $\mathrm{Vol}_{\mathcal{Y}}(\mathcal{P}^{*(j)})$ for $\mathcal{Y} \in \mathcal{P}$ is determined only by its orientation.

Let us now consider the specific case of the weighted cosmological polytopes. In particular, let us consider its triangulation in terms of the collection $\{\mathcal{P}_{\mathcal{G}_j},\, \{\mathcal{G}_j,\, j = 0,\ldots,n_e\}\}$ as defined by (4.61) and in the related paragraph. The orientation of the dual of $\mathcal{P}_{\hat{\mathcal{G}}_j}$ is equal to $\mathrm{sign}\left(\prod_{e \in \mathcal{E}} \mathcal{Y} \cdot \widetilde{\mathcal{W}}^{(\mathfrak{g}_e)}\right)$ for $\mathcal{Y} \in \mathcal{P}$. Therefore, for $\mathcal{Y} \in \mathcal{P}$, each term has the same sign which is equal to the weighted cosmological polytope weight function $w(\mathcal{Y},\mathcal{P}_{\mathcal{G}}^{(w)})$ as given in (4.26). For the cosmological polytope instead a sum with alternating signs appears and hence

$$0 < \Omega(\mathcal{Y},\mathcal{P}_{\mathcal{G}}) < w(\mathcal{Y},\mathcal{P}_{\mathcal{G}}^{(w)})\Omega(\mathcal{Y},\mathcal{P}_{\mathcal{G}}^{(w)}) , \qquad \text{for } \mathcal{Y} \in \mathcal{P}_{\mathcal{G}} . \tag{A.5}$$

Finally, since $\Omega(\mathcal{Y},\mathcal{P})$ is never vanishing in the cosmological polytope, then the numerator of the weighted cosmological polytope is also never vanishing.

## A.2 A note on the new vanishing conditions

In Section 4.3 it has been shown that the covariant restriction of a facet $\mathcal{P}_{\mathcal{G}}^{(w)} \cap \mathcal{W}^{(\mathfrak{g})}$ onto the hyperplane $\mathcal{H}$ is zero if $\bar{\mathfrak{g}}$ is a tree subgraph, where $\mathcal{H} \equiv \mathcal{W}^{(\mathbf{x}_{\bar{s}_1} \cdots \mathbf{x}_{\bar{s}_{\bar{n}_s}})} := \bigcap_{j=1}^{\bar{n}_s} \mathcal{W}^{(\mathbf{x}_{\bar{s}_j})}$. In this appendix we show that the covariant restriction onto the hyperplane $\mathcal{H}_k \equiv \mathcal{W}^{(\mathbf{x}_{\bar{s}_1} \cdots \mathbf{x}_{\bar{s}_k})} := \bigcap_{j=1}^{k} \mathcal{W}^{(\mathbf{x}_{\bar{s}_j})}$ is non-vanishing, where $\{\bar{s}_j,\, j = 1,\ldots,k < \bar{n}_s\}$ is an arbitrary subset of sites in $\bar{\mathfrak{g}}$ of dimension $k$.

First, recall that (A.4) implies that any covariant restriction of the canonical form of a face of $\mathcal{P}_{\mathcal{G}}^{(w)}$ on a hyperplane cannot be zero if the hyperplane intersects the interior of the face. Therefore, in order

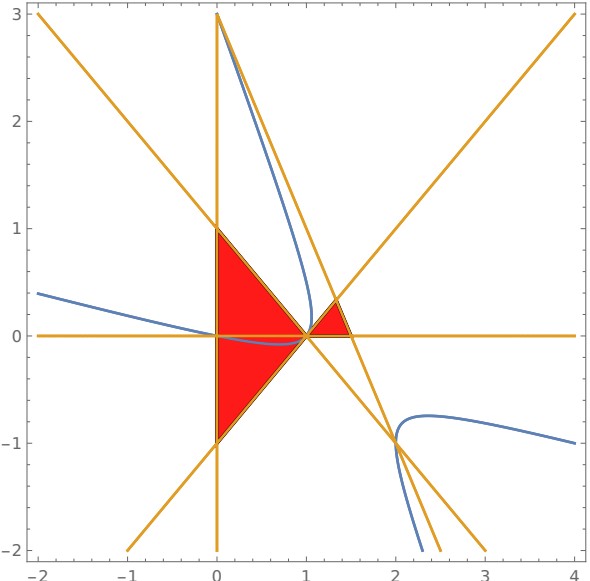

Figure 10: The blue line represents the numerator of the canonical form of the two red triangles with opposite weights. Notice how it approaches zero with different rates depending on the direction.

to prove the above statement, it is necessary to show that if $k < \overline{n}_s$ then $\mathcal{H}_k$ intersects the interior of $\mathcal{P}_{\mathcal{G}}^{(w)}$. This statement can be proven considering that the intersection between a codimension-1 hyperplane $\mathcal{H}_1$ and the interior of $\mathcal{P}_{\mathcal{G}}^{(w)}$ is not empy if there exist two points $P_1$, $P_2$ in $\mathcal{P}_{\mathcal{G}}^{(w)}$, such that $\mathcal{H}_1 \cdot P_1 > 0$ and $\mathcal{H}_1 \cdot P_2 < 0$.

Let us first consider $\mathcal{H}_1 = \mathcal{W}^{(\mathbf{x}_1)}$ and $\overline{n}_s > 1$. Then, two such points $P_1, P_2$ are given by

$$P_1 = \mathbf{x}_1 + \mathbf{y}_{1,2} - \mathbf{x}_2 \qquad P_2 = -\mathbf{x}_1 + \mathbf{y}_{1,2} - \mathbf{x}_2 \ . \tag{A.6}$$

Then, in this simple case, $\mathcal{P}_{\mathcal{G}}^{(w)} \cap \mathcal{H}_1 \neq \varnothing$. The general case can be proven by induction. Without loss of generality, let us consider an arbitrary subset of sites of $\overline{\mathfrak{g}}$ such that one of them, namely $\overline{s}_1$, is connected to at least a site, namely $s_1$, of $\mathfrak{g}$. Let $\mathcal{H}_{k-1} := \mathcal{W}^{(\overline{s}_2 \cdots \overline{s}_k)}$. Then, by the inductive step, $\mathcal{P}_{\mathcal{G}}^{(w)} \cap \mathcal{W}^{(\mathfrak{g})} \cap \mathcal{H}_{k-1} \neq \varnothing$ is a convex weighted polytope which contains the point $P_1, P_2$ as defined in (A.6). This in turn implies that $\mathcal{P}_{\mathcal{G}}^{(w)} \cap \mathcal{W}^{(\mathfrak{g})} \cap \mathcal{H}_k \neq \emptyset$ concluding our proof.

Finally, recall that every face of the weighted cosmological polytope factorizes into a product of ordinary polytopes and weighted polytopes $\mathcal{P}_{\overline{\mathfrak{g}} \cup \mathcal{L}}^{(w)}$. For this reason, this proof can be straightforwardly generalized to any face $\mathcal{P}_{\mathcal{G}}^{(w)} \cap \mathcal{W}^{(\mathfrak{g}_1 \cdot \mathfrak{g}_k)}$.

### A.3 Examples of higher multiplicity zeros

In Section 4.3 we discussed how $k + m$ facet hyperplanes containing the facets of a polytope that intersect each other on the polytope itself in codimension-$k$ identify a boundary living in codimension-$m$ subspace of the adjoint surface, and how this is reflected into the canonical form associated to the polytope via the existence of a zero of multiplicity $m$. ¡Here we would like to elucidate this feature by discussing three geometrically different ways in which a numerator can give rise to a multiplicity-2 zero for some simple examples in 2 dimensions.

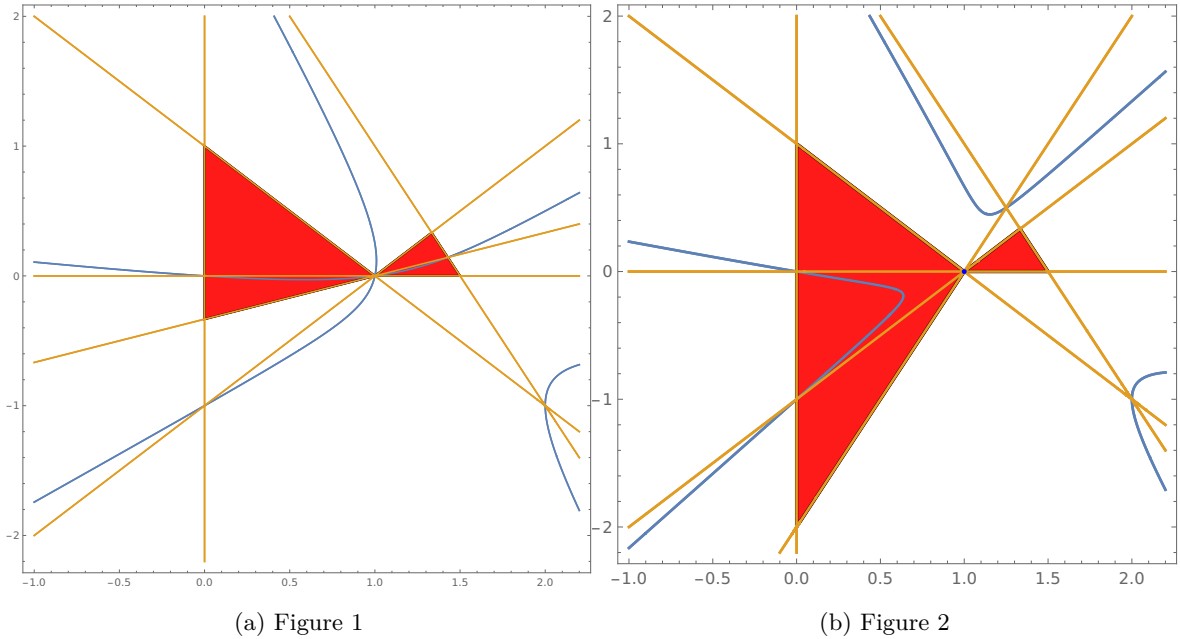

(a) Figure 1

(b) Figure 2

Figure 11: Two triangles with opposite weights. In both cases, no edges are on the same line. On the left side, the cubic curve self-intersects to generate a degree-2 zero. On the right side, the zero is an isolated point.

Consider two triangles with opposite weights that share a vertex and have both one edge and a common line. Working in a specific patch in projective space, it is possible to choose the following explicit equations for the two triangles

$$x > 0 \quad (y - x + 1) > 0 \quad y + x - 1 < 0 \,, y > 0 \quad (y - x + 1) < 0 \quad y + 2x - 3 < 0 \,. \quad \text{(A.7)}$$

The canonical form of this geometry explicitly reads

$$\omega(x,y) = \frac{x^2 + 5xy - x + 2y^2 - 6y}{xy(x - y - 1)(x + y - 1)(2x + y - 3)} dx dy \quad \text{(A.8)}$$

Note that the double residue along the lines $y - x + 1 = 0$ and $y + x - 1 = 0$ depends on the order in which it is taken:

$$\text{Res}_{y-x+1}\text{Res}_{y+x-1}(\omega) = 0 \,, \qquad \text{Res}_{y+x-1}\text{Res}_{y-x+1}(\omega) = 1 \,. \quad \text{(A.9)}$$

We have three boundary lines intersecting on the point $(1,0)$, given by the equation $y - x + 1 = 0, y + x - 1 = 0$. This means that the numerator must vanish at this point otherwise we have a double pole. However, the degree of the pole dictated by the geometry depends on the order in which the residues are taken. We can observe in figure (10) that the numerator is a hyperbola tangent to the common line. In this way, when approaching the point $(1,0)$ along the common line the numerator will have a degree-2 zero. By approaching the same point from any other direction the numerator will vanish linearly. Finally let us notice that the zero locus of the numerator doesn't have to be a connected codimension-1 variety, but can also involve an isolated point! Let's modify our previous

example such that the triangle share a vertex but no edges are aligned. Then we will have 4 lines intersecting on that edge. The numerator must therefore have a degree-2 zero on that point when approaching it from any edge. Depending on the angles at the shared vertex we can have different results. Either a degenerate quartic self-intersecting or an isolated zero. The two cases are represented in figure 11.

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
