# Peer review of "The Geometry of Cosmological Correlators"

_SciPost Physics, doi:SciPost Phys. 18, 105 (2025)_

## Round 1 · Referee Report · Anonymous (Referee 1) · 2024-4-30

Report

See attached pdf. I recommend the paper for publication and have some minor questions for the authors which they can choose to expand upon if they feel would enhance the presentation.

Attachment

Recommendation

Publish (surpasses expectations and criteria for this Journal; among top 10%)

  • validity: -
  • significance: -
  • originality: -
  • clarity: -
  • formatting: -
  • grammar: -

Author:  Gabriele Dian  on 2024-12-20  [id 5055]

(in reply to Report 1 on 2024-04-30)
Category:
answer to question

The authors would like to thank the referee for their thorough report  and valuable comments. Below, we address each of the points raised:

1. The dashing prescription directly follows from a diagrammatic rewriting of Eq. (2.11), after expanding the wavefunctions in terms of the coupling and summing over graphs.

  1. There are two perspectives one can take on generating loop integrands using a Feynman-like tree theorem:

a. our formula 2.36 constitutes an example of a Feynman-like tree theorem , even if it differs from the original formulation for Feynman diagrams in flat space as it relates a 1-loop n_s-site graph to a tree-level (n_s+1) graph -- so the trees are produced by opening up a site rather then cutting a loop-propagator. However, this formulation is valid for arbitrary loops and can be stated as: an L-loop n_s-site integrand can be expressed as an integral of an (L-1)-loop (n_s+1)-site integrand. This formula can be recursed to express the L-loop integrand as an L- fold integral of trees.

b. one can take the point of view of the cosmological tree theorem by Agüi Salcedo and Melville. Such tree-theorem has been proven for one- loop wavefunction integrands. At the moment, we are not able to state that the same structure goes through for correlators. One can note that one-loop correlators can be expressed in terms of a connected wavefunction term and a disconnected one, with the cosmological tree theorem valid for the connected part. One has to see whether it can be economical to check that, representing the connected term via the cosmological tree theorem, it can be re-organised with the disconnected part in a cosmological tree theorem fashion<

  1. The variable z_ss' can be interpreted as the energy of an additional additional external state: more precisely the integrand has the site s with an additional state with energy +z_ss', and the site s' with an additional with energy -z_ss'. This is similar to what happens in the more usual Feynman tree theorem where the integrand has two additional states with equal momentum but with opposite signs (in the forward limit). In our case, they have the same energy but opposite signs, and the energy is integrated over.

Furthermore, it acts as a deformation of the kinematics in the spirit of the BCFW recursion. However, unlike the BCFW approach, this deformation applies solely to the energy of the particle, without imposing energy conservation.

  1. Correlators share some of the wavefunction singularities and additionally have singularities in the loci where the energy of the internal states vanishes. Factorisation 4.94 corresponds to taking the residue of one such internal energy source. The most intuitive way to expect such a factorisation is considering the dashed-graph representation: each dash corresponds to removing an edge and replacing it with 1/y_e. Hence, whenever we consider the locus y_e=0, one has to consider just the contribution from this representation with a dash on the e-edge. For loop graphs this involves both connected and disconnected pieces, the singularity structure of all these pieces is the same as the correlators obtained from the original one by erasing the edge e, and reorganising the terms effectively leads to the upper line in 4.94. The second line is more characteristic of a tree graph, where the dashing produces just disconnected pieces, and the line of reasoning is the same.

However, we would like to point out how with this geometrical picture at hand, proving the existence of such a relation is simpler.

---

## Round 1 · Referee Report · Anonymous (Referee 2) · 2025-2-18

Report

This paper is a high-quality and an original contribution to the field. It meets the standards of SciPost and should be published.

Attachment

Recommendation

Publish (easily meets expectations and criteria for this Journal; among top 50%)

---

## Editorial Decision

published